# The transcription factor DDIT3 is a potential driver of dyserythropoiesis in myelodysplastic syndromes

Nerea Berastegui [1,2,14], Marina Ainciburu [1,2,14], Juan P. Romero[1,2], Paula Garcia-Olloqui [1,2], Ana Alfonso-Pierola[2,3], Céline Philippe [4], Amaia Vilas-Zornoza[1,2], Patxi San Martin-Uriz [1], Raquel Ruiz-Hernández [5], Ander Abarrategi[5,6], Raquel Ordoñez[7], Diego Alignani[1,2], Sarai Sarvide[1,2], Laura Castro-Labrador [1,2], José M. Lamo-Espinosa[8], Mikel San-Julian [8], Tamara Jimenez [9], Félix López-Cadenas[9], Sandra Muntion[9], Fermin Sanchez-Guijo [2,9], Antonieta Molero [10], Maria Julia Montoro[10], Bárbara Tazón[10], Guillermo Serrano [11], Aintzane Diaz-Mazkiaran[1,11], Mikel Hernaez [2,11], Sofía Huerga[3], Findlay Bewicke-Copley [12], Ana Rio-Machin [12], Matthew T. Maurano [7,13], María Díez-Campelo [2,9], David Valcarcel[10], Kevin Rouault-Pierre [4], David Lara-Astiaso [1,15], Teresa Ezponda [1,2,15] ✉ & Felipe Prosper [1,2,3,15] ✉

Myelodysplastic syndromes (MDS) are hematopoietic stem cell (HSC) malignancies characterized by ineffective hematopoiesis, with increased incidence in older individuals. Here we analyze the transcriptome of human HSCs purified from young and older healthy adults, as well as MDS patients, identifying transcriptional alterations following different patterns of expression. While aging-associated lesions seem to predispose HSCs to myeloid transformation, disease-specific alterations may trigger MDS development. Among MDS-specific lesions, we detect the upregulation of the transcription factor DNA Damage Inducible Transcript 3 (*DDIT3*). Overexpression of *DDIT3* in human healthy HSCs induces an MDS-like transcriptional state, and dyserythropoiesis, an effect associated with a failure in the activation of transcriptional programs required for normal erythroid differentiation. Moreover, *DDIT3* knockdown in CD34⁺ cells from MDS patients with anemia is able to restore erythropoiesis. These results identify *DDIT3* as a driver of dyserythropoiesis, and a potential therapeutic target to restore the inefficient erythroid differentiation characterizing MDS patients.

Hematopoiesis is regulated by different molecular mechanisms that precisely delineate the gene expression programs activated in distinct hematopoietic progenitor cells at specific moments, ultimately enabling the particular functions of each lineage[1]. Deregulation of such mechanisms can lead to the development of various hematological disorders, including myelodysplastic syndromes (MDS), which result from alterations in the first steps of hematopoietic differentiation[2,3]. Thus, MDS are characterized by ineffective hematopoiesis, and clinically manifested as cytopenias and increased risk of transformation to acute myeloid leukemia (AML)[3].

MDS prevalence is almost exclusive to older individuals, with a median age at diagnosis of 71 years and an increased incidence after the sixth decade of life[4], suggesting that alterations associated with aging predispose hematopoietic progenitors to the disease. In fact, aging is accompanied by a decline in the hematopoietic system that includes defects in both B cell and T cell lymphopoiesis, anemia, dysregulation of the innate immune system, and augmented risk of developing myeloid diseases[5,6]. Increasing evidence suggests that the mechanisms intrinsic to HSCs are critical for the adverse hematopoietic consequences seen with age. Some of these mechanisms have started to be elucidated and include changes in the methylome, epigenetic machinery, and histone modification profiles of HSCs[7–10], along with altered transcriptional programs that include higher expression of genes involved in myeloid differentiation[11,12]. Moreover, in mice, aging causes the expression of genes involved in leukemic transformation in long-term HSCs[11], while in human, developmental and cancer pathways are epigenetically reprogrammed in CD34[+] cells from older individuals[7], suggesting an aging-mediated predisposition towards the development of myeloid neoplasms. These previous findings suggest that aging-mediated gene expression alterations in HSCs may evolve towards more pathological profiles associated with MDS, although such lesions still need to be elucidated (Fig. 1a).

So far, many studies aiming to characterize the molecular pathogenesis of MDS have mainly focused on the analysis of mutational profiles associated with the disease, but the fundamental molecular bases of this pathology are still incompletely understood. Recent studies have demonstrated that, as in other hematological malignancies[13,14], transcriptional mechanisms may also play a relevant role. In this sense, several works aiming to elucidate gene expression aberrations in MDS patients have started to identify, in HSCs-enriched populations, specific lesions with prognostic value or a potential role in the phenotype of the disease[15–29]. Moreover, certain mutations have been associated with specific transcriptional profiles in patients with MDS[17,21–23], supporting the role of gene expression profiles as common integrators of different alterations, being affected by lesions such as mutations or aberrant signaling from the environment, among other factors.

Although different cell types are phenotypically altered in MDS, previous studies suggest that HSCs are key to understanding this disease, due to their privileged position at the apex of the differentiation process[2,3,30–32]. The low abundance of HSCs in the bone marrow makes their characterization very challenging, implying that their expression profile could be partially masked by other cells when HSCs-enriched populations (i.e: CD34[+]) are studied.

In this work, we focused on the analysis of the transcriptional profile of HSCs enriched subsets from a limited cohort of low-risk MDS patients. By comparing the transcriptional profile of HSCs from healthy young and older adults with that of patients with MDS, we identify specific transcriptional changes associated with aging and with MDS transformation. While aging-associated lesions seem to confer human HSCs with a transformation-prone state, alterations characterizing the progression to MDS may directly alter transcriptional programs leading to inefficient hematopoiesis. Among MDS-specific lesions, we identify *DDIT3*, a transcription factor upregulated in MDS HSCs, as a key regulator of erythropoiesis in MDS. Using gain and loss of function approaches in CD34[+] cells from healthy donors and MDS cases, we demonstrate the role of this TF in promoting inefficient erythroid differentiation, indicating that *DDIT3* may be a therapeutic target for patients with the disease.

## Results

### Profiling of HSCs reveals diverse transcriptional dynamics in aging and MDS

FACS-sorted HSCs (CD34[+] CD38[-] CD90[+] CD45RA[-]) obtained from young (*n* = 17, average=20.53 y/o, range = 18–22 y/o) (YHA_1-17) and older (*n* = 8, average = 67.5 y/o, range = 58–81) healthy adults (OHA_1–8), as well as from untreated MDS patients (*n* = 12, average = 70 y/o, range = 51–87 y/o) (MDS_1-MDS_12) were analyzed using low-input RNA-seq (Fig. 1b and S1a; details of healthy donors and patients are shown in Tables S1 and S2). MDS patients included only low- or very low-risk patients with MDS-MLD and MDS-SLD, excluding cases with del(5q), ring sideroblasts or excess blasts, to reduce the heterogeneity associated with MDS.

Principal component analysis (PCA) showed transcriptional differences between HSCs from young and older (both healthy and MDS) individuals, which were encoded by PC1 (Fig. 1c), and additional differences between healthy older adults and MDS patients, which were encoded by PC2. Differential expression analysis demonstrated 733 genes deregulated between young and older healthy samples, and 907 genes between HSCs from healthy older individuals and MDS patients (|Fold change (FC)| > 2; FDR < 0.05) (Fig. S1b and Supplementary Data file S1). However, a limited overlap was detected between differentially expressed genes (DEG) in both comparisons (Fig. 1d), indicating that specific transcriptional alterations, distinct from those taking place in aging HSCs, occur in MDS. Gene ontology (GO) analyses demonstrated that, among other processes, aging was characterized by an enrichment in inflammatory signaling, and a decrease in cell proliferation and DNA repair signatures; while MDS transcriptional lesions were enriched in RNA metabolism and showed underrepresentation of processes such as migration, or extracellular matrix organization (Fig. 1e). Thus, these results suggest that transcriptional lesions taking place during development of the disease are not a simple continuum of those found in aging, but instead, suggest more complex patterns of expression.

Next, we used MaSigPro[33] to ascertain the transcriptional dynamics characterizing HSCs in aging and MDS, and identified 8 different patterns of expression: two patterns with genes specifically upregulated or downregulated in aging, (clusters C1, C2); two patterns containing genes with augmented or decreased expression specifically in MDS compared to healthy HSCs (C3, C4); two linear trends harboring genes with either increased or decreased expression in aging and with an exacerbation of such changes in MDS (C5, C6); and finally, two patterns in which genes showed deregulation in aging, and alteration in the opposite direction between HSCs from healthy older adults and MDS patients (C7, C8) (Fig. 1f and Supplementary Data file S2). Using GO analysis, we identified the main functional categories and their associated biological processes enriched for each cluster (Fig. 1g and Supplementary Data file S3). Although some processes such as gene regulation and metabolism were common to different clusters, others were cluster-specific, including DNA replication, cell cycle, telomere maintenance or cell adhesion, among others.

Among other relevant findings, some of which are detailed in the supplementary data (Fig. S2), these analyses identified alterations that could be related to the predisposition to the development of hematological malignancies. Interestingly, some of such alterations were detected in HSCs from healthy older adults, suggesting that transcriptional lesions naturally occurring in aging could predispose HSCs for malignant transformation. For example, processes related to cell proliferation were mainly enriched in C2 but also in C6 (Fig. S2a, e, i), suggesting that an important loss of proliferation activity of HSCs takes place during aging, and it is further exacerbated in MDS, an event that has been previously associated with accumulation of DNA damage[34]. Furthermore, DNA sensing and repair processes were also enriched in C2 and C6 (Fig. S2a, e, i), suggesting a progressive loss of the ability to overcome DNA insults and an increased genomic instability of HSCs during aging and MDS, with a potential direct role in disease development. In order to further explore the potential role of transcriptional lesions occurring with age in the promotion of a myeloid disease/cancer susceptibility, we performed gene set enrichment analyses (GSEA) on clusters altered in aging, with or without

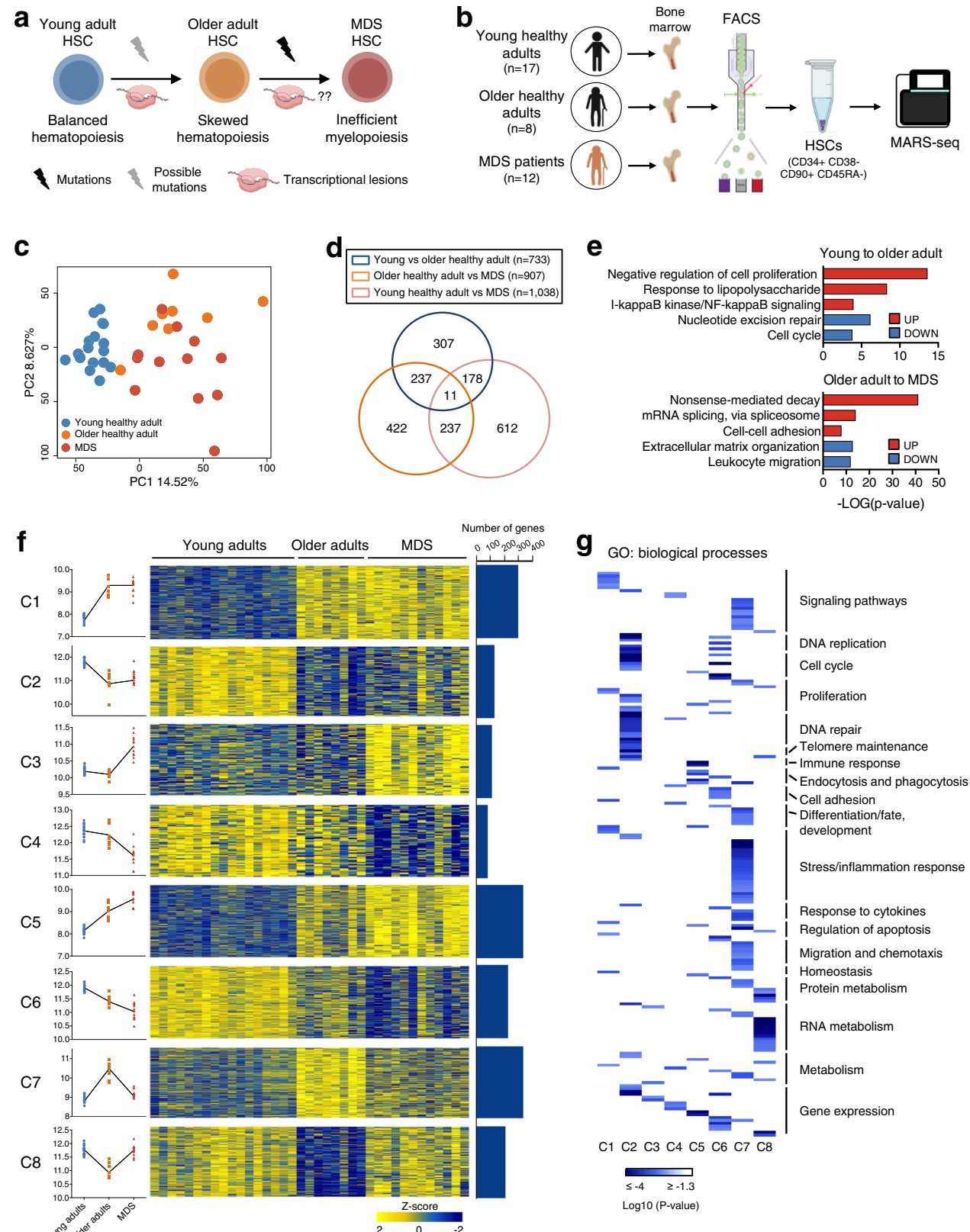

exacerbation of expression in MDS (C1, C2, C5 and C6). These analyses demonstrated that genes contained in such clusters were enriched in cancer-related signatures (Fig. 2a); furthermore, among such genes, we identified factors with known roles in the development or maintenance of myeloid malignancies (Fig. 2b). Some examples included the upregulation of *TRIB1*, a transforming gene for myeloid cells[35], and the matrix metallopeptidase *MMP2*, which presents high secretion levels in AML[36]; as well as the downregulation of genes such as adenosine deaminase *ADA2*, and the mini-chromosome maintenance proteins *MCM7*[37] and *MCM4*, whose repression has been involved in myeloid leukemias (Fig. 2c). These results suggest, in line with a previous work performed in mice, that aging-derived transcriptional changes

**Fig. 1 | Altered transcriptional profiles of HSCs in aging and MDS. a** Schematic representation of lesions taking place in aging and in MDS. Aging is characterized by transcriptional lesions of HSCs and can also present mutations in specific genes. MDS HSCs usually present genetic lesions but the transcriptional profile of these cells remains unexplored. Part of this figure was made in ©BioRender - biorender.com. **b** Schematic representation of the experimental approach: bone marrow specimens from young and older healthy adults as well as from MDS patients were obtained, and HSCs (CD34$^+$ CD38$^-$ CD90$^+$ CD45RA$^-$) were isolated using FACS. The transcriptome of these cells was characterized using MARS-seq. Part of this figure was made in ©BioRender - biorender.com. **c** Principal component analysis (PCA) of the transcriptome of cells isolated in B. Healthy young (blue), older (yellow) adults, and MDS cells (red) are plotted. The percentage of variance explained by PC1 and PC2 principal component is indicated in each axis. **d** Venn diagram representing a partial overlap of genes deregulated in aging (young vs older healthy adults), in the transition to the disease (healthy older adults vs MDS) or between the more distant states (healthy young vs MDS). The number of common or exclusive DEGs is indicated in each area. **e** Bar-plot showing enriched biological processes determined by GO analysis of genes deregulated in aging or between HSCs from healthy older adults and MDS patients. The -log(p-value) for several statistically significant processes is depicted. Analysis was performed using DavidOntology. Modified Fisher exact test was used to calculate p-values. **f** Transcriptional dynamisms identified in HSCs in the aging–MDS axis (C1–C8). Left: dot plot depicting the median expression of genes of each cluster in each sample. Each color represents the different states: healthy young adults: blue, healthy older adults: yellow, and MDS: red. $N = 37$ biologically independent samples ($n = 17$ young adults, 8 older adults, and 12 MDS patients). The trend of each cluster is indicated with a line linking the median of each group. Center: heatmap showing z-scores for the expression profile of each cluster of genes in healthy young and older adults and in MDS samples. Right: Bar-plot indicating the number of genes per cluster. **g** Heatmap showing the statistical significance ($\log_{10} p$ value) of enrichment of the genes of each cluster in different biological processes, as determined by GO analysis. The different processes have been manually grouped into more general biological functions (right). Modified Fisher exact test was used to calculate $p$ values.

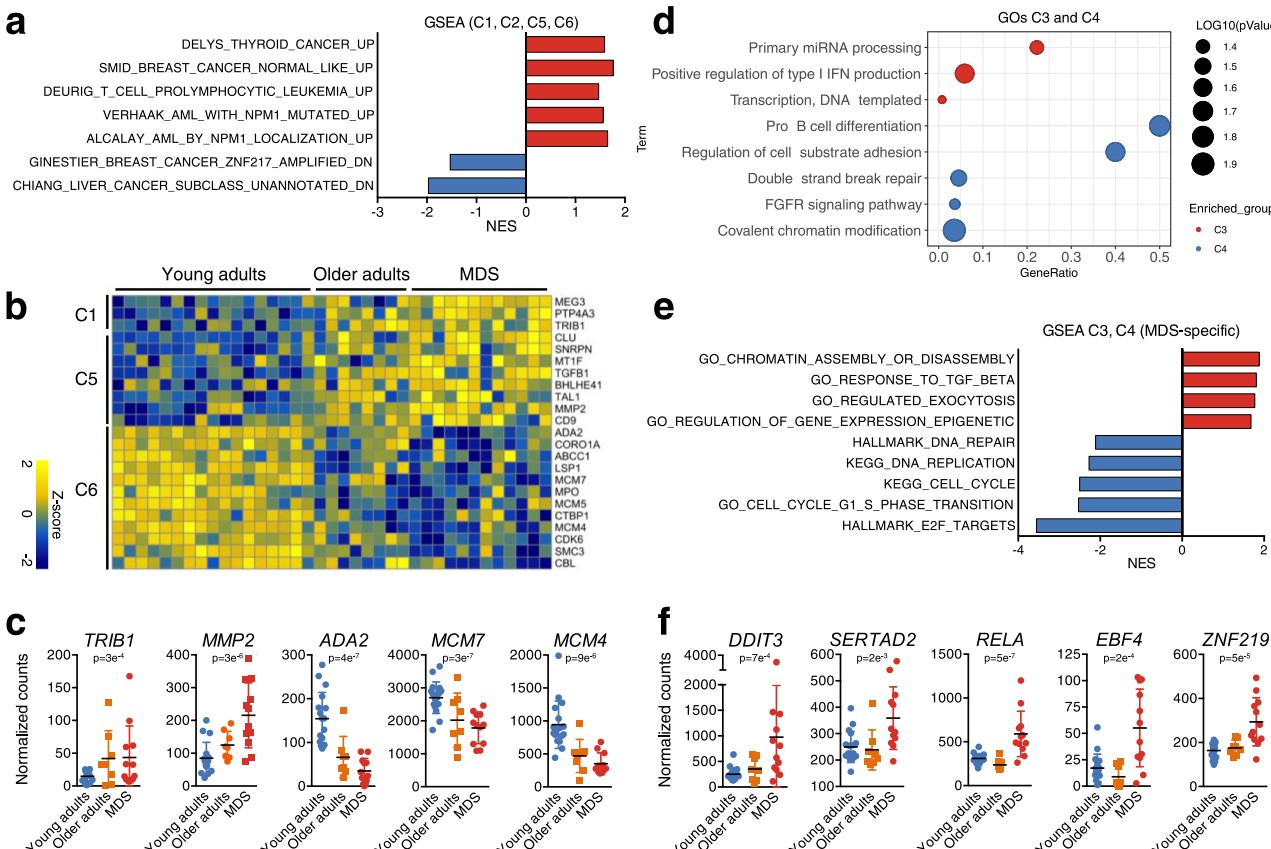

**Fig. 2 | Different transcriptional dynamisms of HSCs in the aging-MDS axis associate with specific biological functions. a** GSEA analysis of genes from C1, C2, C5 and C6. The normalized enrichment score (NES) for several cancer-related signatures is depicted. Only signatures in which age-matched healthy controls were used as normal counterparts of tumor cells were considered. **b** Heatmap of z-scores of genes with known roles in the development of myeloid malignancies. The cluster to which they correspond, and the gene names are indicated on the left and right side of the heatmap, respectively. **c** Normalized expression of several genes from **b** in healthy young adults (blue), healthy older adults (orange) and MDS (red) samples. $N = 37$ biologically independent samples ($n = 17$ young adults, 8 older adults, and 12 MDS patients). Each point represents an individual and the mean +/− standard deviation (SD) is depicted. **d** Bubble plot representing statistically significant biological processes and pathways enriched in genes being specifically altered in MDS (C3: red dots; C4: blue dots). Bubble size depicts −log10(p value) and x axis represents GeneRatio. Analysis was performed using DavidOntology. Modified Fisher exact test was used to calculate $p$ values. **e** GSEA analysis of genes from C3 and C4. The NES for several enriched signatures is shown. **f** Normalized expression of several genes from C3 involved in transcriptional regulation in healthy young adults (blue), healthy older adults (orange) and MDS (red) samples. $N = 37$ biologically independent samples ($n = 17$ young adults, 8 older adults, and 12 MDS patients). Each point represents a donor or patient and the mean +/− standard deviation (SD) is shown.

predispose HSCs for transformation. Such lesions may not be sufficient to promote disease development, and may need additional alterations to lead to an MDS phenotype, which may include enhanced alteration of the lesions found in aging (C5 and C6), genetic insults, or transcriptional alterations that occur exclusively in MDS HSCs.

Thus, we next focused on clusters showing exclusive deregulation in MDS (C3, C4), as they represented lesions with a potential direct role in MDS development. GSEA and GO analyses (Fig. 2d, e) demonstrated diminished expression of genes controlling cell division and DNA repair (Fig. S2k), further suggesting an accumulation of genetic lesions

in MDS-associated HSCs. Decreased levels of genes regulating cell-substrate adhesion and B cell differentiation as well as increased expression of genes involved in processes such as miRNA processing, exocytosis, type I IFN production, or response to TGF-ß were also detected in MDS-associated HSCs (Fig. S2l). Finally, increased expression of chromatin and transcriptional regulators, such as *DDIT3*, *SERTAD2*, *RELA*, *EBF4* or *ZNF219*, among others (Fig. 2f), suggested an MDS-specific epigenetic and transcriptomic reprogramming that could potentially guide the aberrant phenotype of these cells. Collectively, these transcriptional changes are consistent with an aging-mediated predisposition towards myeloid transformation, and with MDS-specific alterations that may contribute to the development of the disease.

## *DDIT3* overexpression produces an MDS-like transcriptional state and alters erythroid differentiation

To determine the potential functional relevance of the detected transcriptional alterations, we next focused on the transcriptional regulators specifically upregulated in MDS (C3), as they could represent key factors for MDS development. Among them, we selected the TF *DDIT3* (DNA Damage Inducible Transcript *3*/CHOP/C/EBPζ) as one of the top-ranking upregulated transcriptional regulators by FC, showing a range of 2.7–10.5 FC over samples from healthy older adults in several MDS-derived HSC specimens (Fig. 2f, S3a). Interestingly, increased *DDIT3* expression was only detected in a small percentage of patients when total CD34+ cells were analyzed, suggesting its upregulation mainly occurs at the most immature state of differentiation (Fig. S3b). In addition, *DDIT3* has been described to be involved in hematopoietic differentiation in mice, specifically promoting abnormal erythroid and myeloid differentiation[38]. To determine the role of its upregulation in HSCs from MDS patients, *DDIT3* was overexpressed in HSCs from healthy donors (YH_18, YH_19) (validation of the overexpression system at protein level by immunofluorescence in primary cells, and by immunoblot in cell lines can be found in figures Fig. S3c, d; and HSCs gating strategy in Fig. S1a), and sorted transduced cells were analyzed by MARS-seq after 2 days in culture in the absence of differentiation stimuli (Fig. 3a). *DDIT3*-overexpression induced transcriptional changes, with an up- and downregulation of 427 and 128 genes, respectively ($|FC| > 2$, $FDR < 0.05$) (Fig. 3b). Consistent with the alterations observed in MDS patients, GSEA demonstrated activation of chromatin remodelers, and decrease in DNA repair pathways and cell-substrate adhesion signatures upon *DDIT3* overexpression (Fig. 3c and S3e, f). In line with the *DDIT3*-induced erythroid bias previously described in mice, we observed an upregulation of genes associated with heme-metabolism or erythroid differentiation, such as *FN3K*, *GCLM* or *HBB* (Fig. S3g). Furthermore, *DDIT3* overexpression promoted an enrichment in cancer-related signatures, indicating a potential oncogenic role of this TF (Fig. 3d). More importantly, using our own gene signatures generated from the DEGs detected in our cohort of MDS samples, we observed that *DDIT3*-overexpression induced a "MDS-like" transcriptional state, activating genes upregulated in MDS patients, and repressing factors downregulated in the disease (Figs. 3e, f and S3h). All together, these data suggested that *DDIT3* upregulation plays a key role in inducing a pathological transcriptional state observed in MDS patients.

We next examined the effect of *DDIT3* overexpression on the function of HSCs using an ex vivo myeloid differentiation system starting from primary human healthy CD34+ cells (donors YH_20, YH_21, YH_22) (Fig. 3a) (gating strategy for CD34+ cells can be found in Fig. S1a). We observed a statistically significant decrease in the number of both erythroid burst-forming units (BFU-E) and monocytic colony forming units after *DDIT3* overexpression (Fig. 4a). Erythroid colonies did not only present a lower total number upon overexpression of the factor, but also showed a modified morphology, being smaller and less compact that those formed by cells transduced with the control

plasmid (Fig. S4a). Liquid culture erythroid differentiation assays demonstrated a delay in the differentiation of *DDIT3*-overexpressing cells, with a statistically significant decrease in later stages of erythroid differentiation (stage IV, CD71− CD235a+) at days 10 and 14, and an increase in stage II and III erythroid progenitors (CD71+ CD235a− and CD71+ CD235a+ erythroblasts) (Fig. 4b, c). No significant alterations were detected in myeloid liquid culture differentiation assays upon *DDIT3* overexpression (donors YH_20, YH_21, YH_23) (Fig. S4b). Moreover, *DDIT3*-overexpression did not produce alterations in the proliferation, cell cycle status, or in the percentage of apoptotic cells (donors YH_24, YH_25, YH_26) (Fig. S4c–e), suggesting that the abnormal erythropoiesis observed upon *DDIT3* overexpression is not due to an expansion of erythroid progenitors, but to a delayed differentiation of progenitor cells.

As previous experiments were performed in CD34+ cells, and in order to prove that the observed effect on erythropoiesis could be driven by the *DDIT3* upregulation observed in HSCs from MDS patients, we also induced *DDIT3*-overexpression in primary HSCs isolated from healthy donors (YH_27, YH_28, YH_29). Overexpression of this factor also produced a delay in erythroid differentiation that was very similar to that observed for CD34+ cells, with a 30% decrease in the percentage of cells in stage III, and an accumulation of cells at stage II at 14 days of differentiation (Fig. S4f).

To further characterize the effect of *DDIT3* overexpression in erythropoiesis, we performed scRNA-seq in human healthy CD34+ cells (donor YH_30) overexpressing *DDIT3* or a mock control after 14 days of ex vivo liquid culture differentiation (Fig. 3a). Clusters associated with myeloid differentiation were excluded from further analyses, and we focused on clusters representing erythroid progenitors (Fig. S4g). Cluster 4 was annotated as reticulocytes, due to its low counts, high ribosomal RNA content and the expression of genes characterizing this stage, such as genes from the ATG family (Fig. S4g, h). The remaining clusters were annotated as different stages of erythroid differentiation according to published transcriptomic data[39], and included CD34+ cells, burst forming unit/primitive erythroid progenitor cells (BFU), colony formation unit/later-stage erythroid progenitor cells (CFU), proerythroblasts, and different stages of erythroblasts (early-basophilic, late-basophilic, polychromatic and orthochromatic) (Fig. 4d, S4i). Expression of *DDIT3* was upregulated in every erythroid differentiation stage in comparison with control cells (Fig. S4j), and such overexpression was associated with an increase in the percentage of early erythroid progenitors, and a 34% and 46% decrease in the orthochromatic and reticulocyte stages, respectively (Fig. 4e). Analysis of differentiation trajectories showed differences in RNA velocity between control and *DDIT3*-overexpressing cells that were consistent with the observed decrease in late erythroid progenitors (Fig. 4f). *DDIT3*-upregulated cells showed a statistically significant decrease in RNA velocity from late basophilic to reticulocyte states, which could be clearly seen when length of velocity, which represents the speed or rate of differentiation was quantified (Fig. 4g, h). These results suggest an impairment of the correct differentiation of the later stages of erythroblasts. Altogether, our data indicate that *DDIT3* overexpression in normal progenitor cells leads to inefficient erythropoiesis.

## *DDIT3*-overexpression leads to a failure in the activation of key erythroid transcriptional programs

To define the molecular mechanisms underlying the erythropoiesis defect induced by *DDIT3* overexpression, we analyzed the transcriptional profile of healthy CD34+ cells transduced with this TF and subjected to erythroid differentiation (donors YH_21, YH_22) (Fig. 3a). GSEA of DEGs (Fig. S5a) demonstrated an enrichment in transcriptional signatures of stem and early progenitor cells and decreased features of erythroid differentiated cells, such as oxygen transport or erythrocyte development, upon *DDIT3* overexpression (Fig. 5a). Expression of

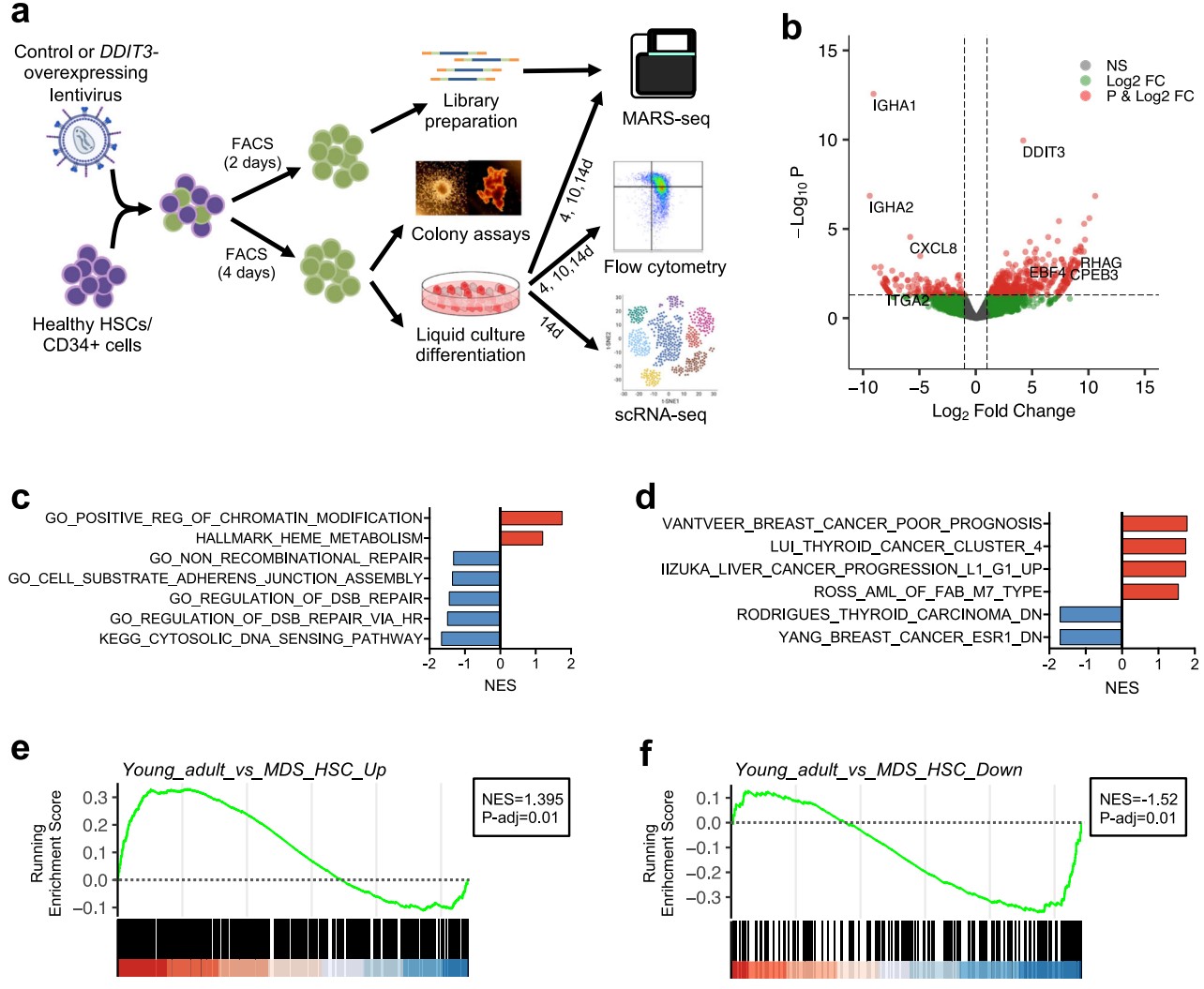

**Fig. 3 | *DDIT3* overexpression promotes and MDS-like transcriptional state.**
**a** Schematic representation of the experimental procedure: primary HSCs/CD34+ cells from healthy young donors were FACS-sorted and infected with a lentiviral plasmid harboring *DDIT3*, or a control plasmid. Two days after infection, transduced cells were sorted and their transcriptome was analyzed using MARS-seq. After 4 days of infection, transduced cells were also isolated and used to perform colony and liquid culture differentiation assays. The later were evaluated by flow cytometry and MARS-seq at different time points and by scRNA-seq at day 14 of differentiation. Part of this figure was made in ©BioRender - biorender.com. **b** Volcano plot of statistical significance ($-\log_{10}$ ($p$ value)) against fold-change ($\log_2$ Fold-change) of gene expression between cells overexpressing *DDIT3* and control

HSCs. Green points represent genes with |FC| > 2, red points depict genes with |FC| > 2 and FDR < 0.05 and grey points indicate genes with no relevant changes in expression. Several genes with significant up- and down-regulation are indicated. Statistical values of differential expression were calculated with DEseq2, and plotted using EnhancedVolcano library. **c**, **d** GSEA of cells overexpressing *DDIT3* and control HSCs. The NES for several signatures related to general biological processes (**c**) and cancer-related signatures (**d**) are depicted. **e**, **f** GSEA plots depicting the enrichment upon *DDIT3* overexpression in gene signatures representing genes up- and down-regulated in HSCs from MDS patients when compared to those from young healthy adults. GSEA was performed with the GseaPreranked tool. The NES and adjusted *p* values are indicated for each signature.

hemoglobin genes and enzymes involved in heme biosynthesis (i.e: *PPOX*, *FECH*, *ALAS2*, *HMBS*) showed increased expression during differentiation in control cells but not in *DDIT3*-transduced cells (Figs. 5b and S5b). On the contrary, factors characteristic of stem cells, such as *HOXB* genes, *NDN* or *TNIK*, which were progressively repressed in control cells, showed aberrant high expression in *DDIT3*-overexpressing cells (Figs. 5b and S5c). We next used the generated scRNA-seq data to analyze *DDIT3*-induced transcriptional lesions at different stages of erythroid differentiation, from CFU to orthochromatic erythroblast. *DDIT3*-overexpression induced an enrichment in hematopoietic stem and progenitor cell signatures in comparison with control cells (Fig. 5c), as well as a decrease in the expression of genes related to heme metabolism or oxygen transport at every stage of differentiation. This was also supported by pseudotime analyses showing early hematopoietic progenitor genes downregulated during normal erythroid differentiation (*SEC61A1*, *CBFB*, *WDR18*) to be upregulated in

*DDIT3*-overexpressing cells (Figs. 5d and S5d); while demonstrating decreased expression of genes involved in erythroid differentiation and heme synthesis (*FAM210B*, *FOXO3*, hemoglobin genes) upon *DDIT3* overexpression (Figs. 5d and S5e). These data demonstrate that *DDIT3* overexpression impairs the physiological transcriptional regulation of erythroid differentiation, leading to a failure in the repression of immature genes, and in the activation of factors that are required for proper erythrocyte formation.

To dwell into the transcriptional programs involved in the abnormal erythroid differentiation induced by *DDIT3* overexpression, we applied a recently described algorithm, SimiC, that infers the differential activity regulons (TFs and their target genes) between two different conditions[40]. Whereas most regulons behaved similarly between control and *DDIT3*-overexpressing cells (Fig. 5e, low dissimilarity score, purple), we identified a small number of regulons with differential activity (high dissimilarity score, orange-yellow). Regulons

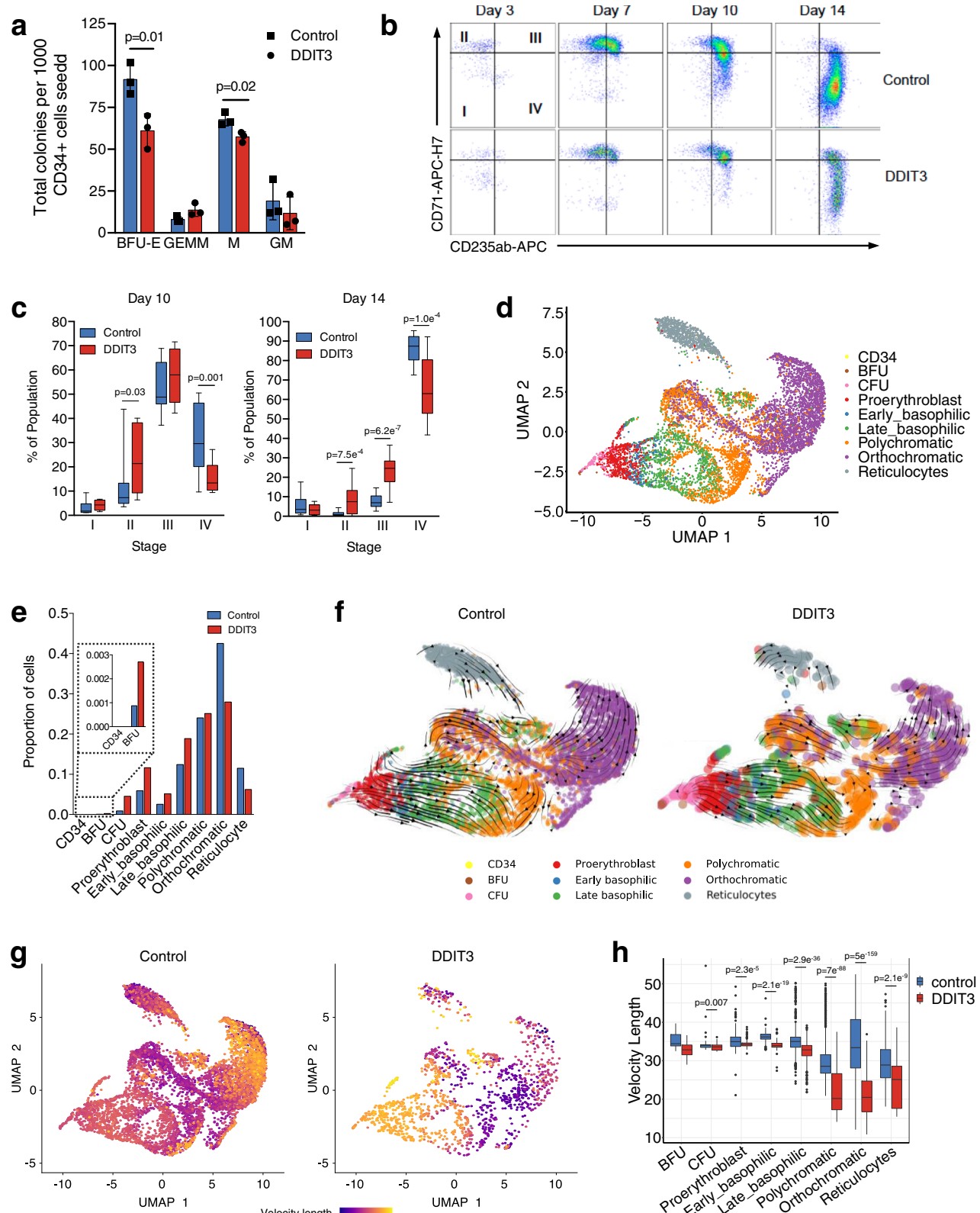

with higher activity in control cells were guided by TFs that positively regulate erythroid differentiation, such as *KLF1*, *ARID4B*, *SOX6*, *TAL1* or *HES6* (Fig. 5e, details for *KLF1*, *ARID4B* and *SOX6* in Fig. 5f), whereas regulons that gained activity upon *DDIT3* overexpression were driven by TFs that negatively impact erythropoiesis, such as *ARID3A*, *ZFPM1*, *JUND*, *ARID1A*, or *HMGA1* (Fig. 5e, details for *JUND*, *ARID3A* and *ARID1A* in Fig. S5f). In most cases, dissimilarities were more evident at earlier stages of differentiation (Figs. 5e, f and S5f), suggesting that aberrant activity of these regulons in immature erythroid progenitors may trigger inefficient erythropoiesis. Differential expression and pseudotime analyses showed that *DDIT3* overexpression was associated with diminished expression of TFs that positively regulate erythropoiesis, such as *SOX6*, *KLF1*, *TAL1* or *ARID4B* (Figs. 5g and S5g), potentially leading to lower activity of such regulons. Furthermore, the

**Fig. 4 | DDIT3 overexpression promotes a defect in erythroid differentiation.**
**a** Bar and scatter plot indicating the number of colonies (BFU-E and GEMM, M, GM) obtained for control cells or cells overexpressing *DDIT3* in three independent biological replicates. Data are presented as mean values +/− SD. *p* values were calculated with multiple *t* test corrected for multiple comparisons using the Holm−Sidak method. **b** Flow cytometry charts representing advanced erythroid differentiation (CD71 and CD235a markers; stages I–IV) for control cells and cell overexpressing *DDIT3*, at the indicated time points. **c** Box-plots (center line, median; box limits, 25th and 75th percentiles; whiskers, min to max) representing the percentage of cells observed in **c** in stages I–IV at 10 and 14 days of differentiation. Three independent biological replicates are depicted. Statistically significant differences are indicated, and were calculated with multiple *t* test corrected for multiple comparisons using the Holm−Sidak method. **d** UMAP plot of the transcriptome of cells subjected to ex vivo differentiation for 14 days. Cells have been clustered in different groups representing erythroid differentiation: CD34+ cells, burst forming unit/primitive erythroid progenitor cells (BFU), colony formation unit/later-stage erythroid progenitor cells (CFU), proerythroblast, early-basophilic erythroblast, late-basophilic erythroblast, polychromatic erythroblast,

orthochromatic erythroblast, and reticulocyte. **e** Bar-plot representing the proportions of cells in each of the clusters described in **f** for control cells or cells overexpressing *DDIT3* upon ex vivo differentiation for 14 days. **f** RNA velocity plotted in UMAP space for control or *DDIT3*-overexpressing cells in clusters defined in **e**. Streamlines and arrows indicate the location of the estimated future cell state. **g** UMAPs representing the length of velocity (from low-purple to high-yellow velocity length) for control and *DDIT3*-overexpressing cells. **h** Box plots representing the length of velocity for each erythroid differentiation state for control and *DDIT3*-ovexpressing cells. Box boundaries are the 25th and 75th quartiles, and the line within the box is the 50th quartile (median). Whiskers limits correspond to the largest and smallest values within 1.5 times the interquartile range (IQR). Dots represent values >1.5 times the IQR. *p* values were calculated using the Wilcoxon test. Biologically independent replicates (cells) for each differentiation state were: BFU: *n* = 6 (Ctrl), *n* = 4 (DDIT3); CFU: *n* = 62 (Ctrl), *n* = 67 (DDIT3); proerythroblast: *n* = 407 (Ctrl), *n* = 172 (DDIT3); early_basophilic: *n* = 176 (Ctrl), *n* = 76 (DDIT3); late_basophilic: *n* = 853 (Ctrl), *n* = 279 (DDIT3); polychromatic: *n* = 1,649 (Ctrl), *n* = 371 (DDIT3); orthochromatic: *n* = 2,918 (Ctrl), *n* = 413 (DDIT3); reticulocytes: *n* = 791 (Ctrl), *n* = 92 (DDIT3).

promoters of these TFs were enriched in binding sites for *CEBPB* and *CEBPG* (Fig. 5h), which are known to be sequestered and inhibited by *DDIT3*. Immunoprecipitation experiments confirmed DDIT3 binding to CEBPB and CEBPG, and how such binding was enhanced after overexpression of *DDIT3* (Figs. 5i and S5h, i). Accordingly, regulons guided by *CEBPG* and *CEBPB* showed decreased activity upon *DDIT3* overexpression (Figs. 5e and S5j). Thus, these results suggested that abnormally high levels of *DDIT3* in MDS sequester *CEBPB* and *CEBPG*, impeding the physiological transcriptional activity of these factors, and leading to decreased expression of their target genes, including TFs with key roles in erythropoiesis. Therefore, *DDIT3*-overexpression ultimately hampers the activity of transcriptional programs that are necessary for proper terminal erythroid differentiation.

## DDIT3 knockdown restores erythroid differentiation of primary MDS samples

Based on our results, we hypothesized that inhibition of *DDIT3* in MDS hematopoietic progenitor cells could restore normal erythropoiesis. *DDIT3* was knocked-down in CD34+ cells from patients with MDS using shRNAs, and cells were induced to differentiate using the OP-9 differentiation system (Fig. 6a). Validation of the knockdown system at the protein level was demonstrated by immunofluorescence in primary MDS cells, and by immunoblot in cell lines (Fig. S6a, b). When submitted to the differentiation system, CD34+ cells from MDS patients show a clear delay in erythropoiesis, barely reaching stage IV or reaching at a much lower percentage at day 14 than healthy cells (Fig. S6c). *DDIT3* knockdown in patient 13 (RNA levels after knockdown shown in Fig. S6d), a male with MDS-MLD and anemia (hemoglobin (Hb) 11.8 g/dL), led to an increase in the percentage of cells in stage IV (CD235+ CD71-) at day 7 of differentiation, an effect that was increased at day 13, where cells expressing the control shRNA were partially blocked at stage III (Fig. 6b, c). We also detected improved erythroid differentiation upon *DDIT3* knockdown in CD34+ cells of four additional MDS cases showing anemia. In two MDS-MLD cases, patient 14 (male, Hb 7.9 g/dL) (RNA levels after knockdown shown in Fig. S6e); and patient 15 (male, Hb 9.3 g/dL), *DDIT3* knockdown promoted higher levels of CD71 expression and improved transition to stage III (Fig. S6f, g). Furthermore, in the other two cases, one MDS-MLD (patient 16, male, Hb 11.9 g/dL) and one MDS-EB2 (patient 17, male, Hb 9.9 g/dL), knockdown of *DDIT3* enhanced the transition to stage IV (Fig. S7a, b), indicating that inhibition of this factor boosts terminal erythropoiesis. Notably, although the patients tested presented variable *DDIT3* basal levels of expression in their HSCs (Fig. S6h), all of them showed similar effects upon knockdown of the transcription factor. Interestingly, overexpression of *DDIT3* in CD34+ cells from patients 16 and 17, which showed low basal levels of this factor, promoted a slight delay in

erythroid differentiation, with a decrease of cells in stages III and IV, and a concomitant increase in stage I for MDS17, and stages I and II for specimen MDS16 (Fig. S7c). The opposite effects observed on erythroid differentiation upon knockdown and overexpression of *DDIT3* in the same specimens further supported the model of *DDIT3*-driven dyserythropoiesis in MDS. Finally, the promotion of erythroid differentiation by *DDIT3* knockdown was validated by transcriptional profiling of cells from two of the patients analyzed (patients 13 and 14) at day 7 of differentiation. This analysis revealed that *DDIT3* knockdown induced increased expression of hemoglobin and heme biosynthetic enzymes (Figs. 6d and S7d), and diminished levels of genes characteristic of immature progenitors, such as HOX genes (Figs. 6d and S7e), compared to cells transduced with a control shRNA. Moreover, using published data of gene expression profiling at different erythroid differentiation stages[41], we observed that *DDIT3* knockdown decreased the expression of genes that are mainly expressed in proerythroblasts, and early and late basophilic erythroblasts, while promoting the activation of genes defining poly- and orthochromatic stages (Figs. 6e and S7f). All together, these results suggest that inhibition of *DDIT3* in patients with MDS presenting anemia restores proper terminal erythroid differentiation.

## Discussion

In this work, we have taken two approaches in the study of transcriptional alterations in MDS that are different from other approximations previously used in the study of the disease, and that we believe are relevant for understanding its molecular pathogenesis. Firstly, as MDS is considered an HSC disease, we focused our analysis on HSC enriched subsets, which may explain the identification of several genes deregulated in MDS not previously described[15-24]. Secondly, we have analyzed the molecular lesions of these cells in the context of aging, allowing for the identification of complex alterations in aging and MDS development that include not only specific lesions of aging and MDS, but also continuous alterations, and even reversal of the aging transcriptome. Our findings suggest a profound transcriptional regulation of HSCs in aging and MDS development and points to the advantage of using age-matched controls for the study of aging-associated diseases.

One of the main characteristics of MDS is the high heterogeneity of patients regarding the type and severity of hematopoietic dysfunction, and the molecular lesions that hematopoietic progenitors harbor[42]. In our work, we aimed to partially overcome such heterogeneity by focusing on low or very low-risk MDS-MLD and MDS-SLD, excluding other types of MDS (MDS-RS, MDS-EB, MDS-del(5q), MDS-U). Thus, although the number of cases of study was limited, they represented a relatively homogeneous cohort, which we believe is key

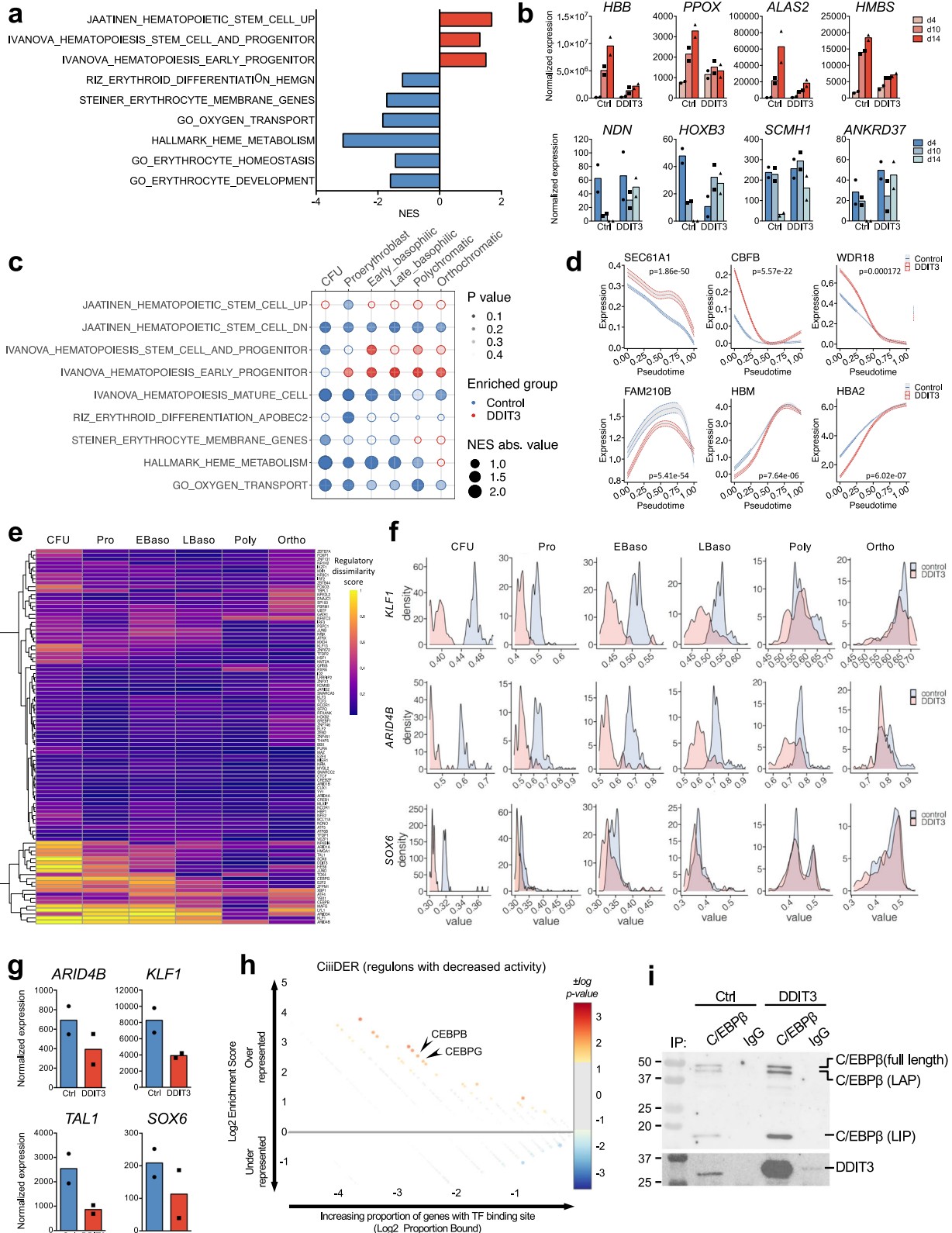

in order to find relevant transcriptional alterations. So far, some studies have described an association between genetic and transcriptomic alterations, showing specific gene expression profiles of different FAB and cytogenic groups[22,23], differential isoform usage of genes in MDS patients carrying mutations in spliceosome and non-spliceosome factors[17], and alteration of particular subsets of genes with specific splicing mutations[21]. The focus of the present work was not to establish such associations, but to demonstrate how the transcriptomic analysis of purified HSCs can unveil transcriptional lesions that may be relevant for MDS development, nevertheless, future works using larger cohorts of HSCs from MDS cases may provide such correlations.

**Fig. 5 | *DDIT3* upregulation leads to a failure in the activation of erythroid differentiation programs. a** Bar-plot depicting GSEA analysis upon *DDIT3* over-expression at day 14 of cells overexpressing *DDIT3* and control HSCs at day 14 of erythroid differentiation. The NES for several signatures related to erythropoiesis and stem and progenitor profiles are shown. **b** Normalized expression of erythroid differentiation (top) and stem cell genes (bottom) of CD34+ cells at different time points of erythroid differentiation. *N* = 2 biologically independent samples. **c** Plot representing GSEA after differential expression analysis between cells over-expressing *DDIT3* and control cells in scRNA-seq, done for the different stages of erythroid differentiation detected. We used the average logFC to rank all expressed genes. We tested for enrichment of signatures related to erythropoiesis and stem and progenitor profiles from MSigDB. GSEA was performed with the fgsea R package. The size of the dots represents NES absolute value, the color indicates the group in which processes are enriched (blue in control cells, red in *DDIT3*-over-expressing cells), and the intensity of the color depicts the p-value obtained for each geneset. **d** Gene expression trends of early hematopoietic progenitor genes (top) and erythroid differentiation factors (bottom) calculated by pseudotime are represented as a smooth fit (prediction of the linear model) with the SD of the fit

shown in a lighter shade for control (blue) and *DDIT3*-overexpressing cells (red). *p* values showing the statistical differences between both trends of expression are indicated, and were calculated using the Wilcoxon test. **e** Heatmap showing the regulatory dissimilarity score between control and *DDIT3*-overexpressing cells at day 14 of ex vivo liquid culture differentiation in different stages of erythroid differentiation defined by scRNA-seq. **f** Ridge plot showing AUC scores for regulons of *KLF1*, *ARID4B* and *SOX6* in control (blue) and *DDIT3*-overexpressing cells (pink) at different stages of erythroid differentiation. **g** Normalized expression of TFs guiding regulons showing decrease activity in *DDIT3*-overexpressing cells, at different day 14 of ex vivo erythroid differentiation. *N* = 2 biologically independent samples. **h** Analysis of putative transcription factor site enrichment in TFs guiding regulons with decreased activity in *DDIT3*-overexpressing cells using CiiiDER. Color and size of circles reflect *p* value of enrichment. Over-represented transcription factors of potential interest are depicted. **i** Immunoblot for CEBPB (different iso-forms indicated) and DDIT3 after the immunoprecipitation with an anti-CEBPB antibody or an IgG control in cells transduced with a control or a *DDIT3*-over-expressing plasmid. These experiments were repeated three independent times with similar results.

GO analyses unveiled distinct potential functions of the groups of genes identified, undercovering putative transformation mechanisms. Aging-derived lesions suggested a transformation-prone state of HSCs from healthy older adults. These alterations included the decreased expression of genes involved in DNA damage sensing and repair pathways, which goes in line with the accumulation of DNA damage observed in human CD34+CD38− cells from older adults[43]. Diminished levels of such genes might facilitate the accumulation of mutations in these cells, an event that is associated with MDS. Although contradictory data has been published suggesting either increased or diminished proliferation rate of human HSCs during aging, our data support the latest, in line with studies showing that the aging-phenotype of HSCs only emerges after HSCs have reached their maximum number of divisions[44]. Moreover, previous observations showing that DNA repair takes place when HSCs enter the cell cycle[34] suggest that the diminished proliferative activity of HSCs from healthy older individuals could contribute to the accumulation of mutations that takes place in these cells with age[43]. Furthermore, HSCs from healthy older adults also showed an enrichment in cancer-related signatures and factors with known roles in the development of myeloid malignancies. Accordingly, murine aged long-term HSCs overexpress genes involved in leukemic transformation[11], while in humans, a profound epigenetic reprogramming of enhancers has been suggested to promote an aging-driven leukemic transformation-prone state of HSC-enriched populations[7]. Collectively, these data support that aging-dependent alterations in gene expression at the HSC level increase the risk of malignant transformation into myeloid neoplasms, explaining at least in part the higher incidence of these diseases in older population.

Our data also identified MDS-associated transcriptional alterations that may trigger the development of the disease, such as upregulation of *DDIT3*, a member of the C/EBP family of TFs, which is involved in functions such as cellular differentiation and proliferation, control of apoptosis and inflammatory processes. *DDIT3* is altered in numerous tumors, with lesions ranging from altered expression (up- and down-regulation) to structural abnormalities, including deletions and amplifications (cBioPortal), and translocations that lead to fusion proteins and oncogenic variants[45,46], suggesting that its role in cancer may be context-specific. Intriguingly, a couple of previous works reported downregulation of *DDIT3*[47,48] and hypermethylation of its promoter[49] in MDS. These studies analyzed BM mononuclear cells, from which HSCs represent a very small percentage, and thus, the populations analyzed were different from our work. It is possible that in MDS, *DDIT3* is specifically upregulated in early phases of progenitor cell differentiation and commitment which is consistent with our results, showing that *DDIT3* upregulation was more evident in HSCs than in total CD34+ cells. In fact, *DDIT3* has been described as a central

regulator of erythro-myeloid lineage specification in mice, where its overexpression, in the absence of pro-differentiative cytokines, enables erythroid programs and impairs myeloid differentiation[38]. Similarly, we observed that, in the absence of differentiation stimuli, *DDIT3* overexpression in human HSCs promoted an erythroid-prone state. Nevertheless, it also promoted a failure in proper erythroid differentiation, with cells being blocked at early maturation stages, indicating that although high *DDIT3* levels seem to prime HSCs for erythropoiesis, they promote an inefficient erythroid differentiation, which is not terminal. Our data suggest that the defect in the activation of differentiation regulons at early stages of erythropoiesis may be due to decreased activity of *DDIT3* binding partners, *CEBPB* and *CEBPG*, which we show are sequestered by abnormally high *DDIT3* levels. Future studies characterizing specific subpopulations will determine whether downregulation of this TF takes place at more mature stages, the nature of its binding partners in those cells, and the phenotypic effect that such lesions imply.

The fact that knockdown of *DDIT3* in CD34+ cells from MDS cases restores erythroid differentiation, suggests that *DDIT3* could represent a potential therapeutic target for the treatment of patients, particularly those with anemia. Intriguingly, our experiments demonstrate that *DDIT3* knockdown may not only be beneficial for patients showing upregulation of this factor, as cases presenting levels similar to those observed in healthy older adults also demonstrated improved erythroid differentiation upon its knockdown. These results suggest that, independently of its levels of expression, *DDIT3* may acquire a key role in the control of erythropoiesis in MDS patients. Finally, we would like to pinpoint that, besides *DDIT3*, other factors identified in this study showing altered expression in MDS could also have a potential role in the promoting an aberrant differentiation, and their functional role will be evaluated in future studies.

In summary, by analyzing the transcriptome of HSCs we have characterized different transcriptional programs altered during aging and MDS development. Moreover, we have identified *DDIT3* as a key TF involved in the pathogenesis of abnormal erythropoiesis in MDS, thus, representing a potential therapeutic target to restore the inefficient erythroid differentiation of these patients. Similarly, future transcriptional analysis of HSCs in other MDS subtypes may also uncover transcriptional lesions with relevant roles in MDS development and progression.

## Methods

The research performed in this work complies with all relevant ethical regulations: the study was approved by the research ethics committee of University of Navarra, and informed consent was obtained from all patients and healthy donors.

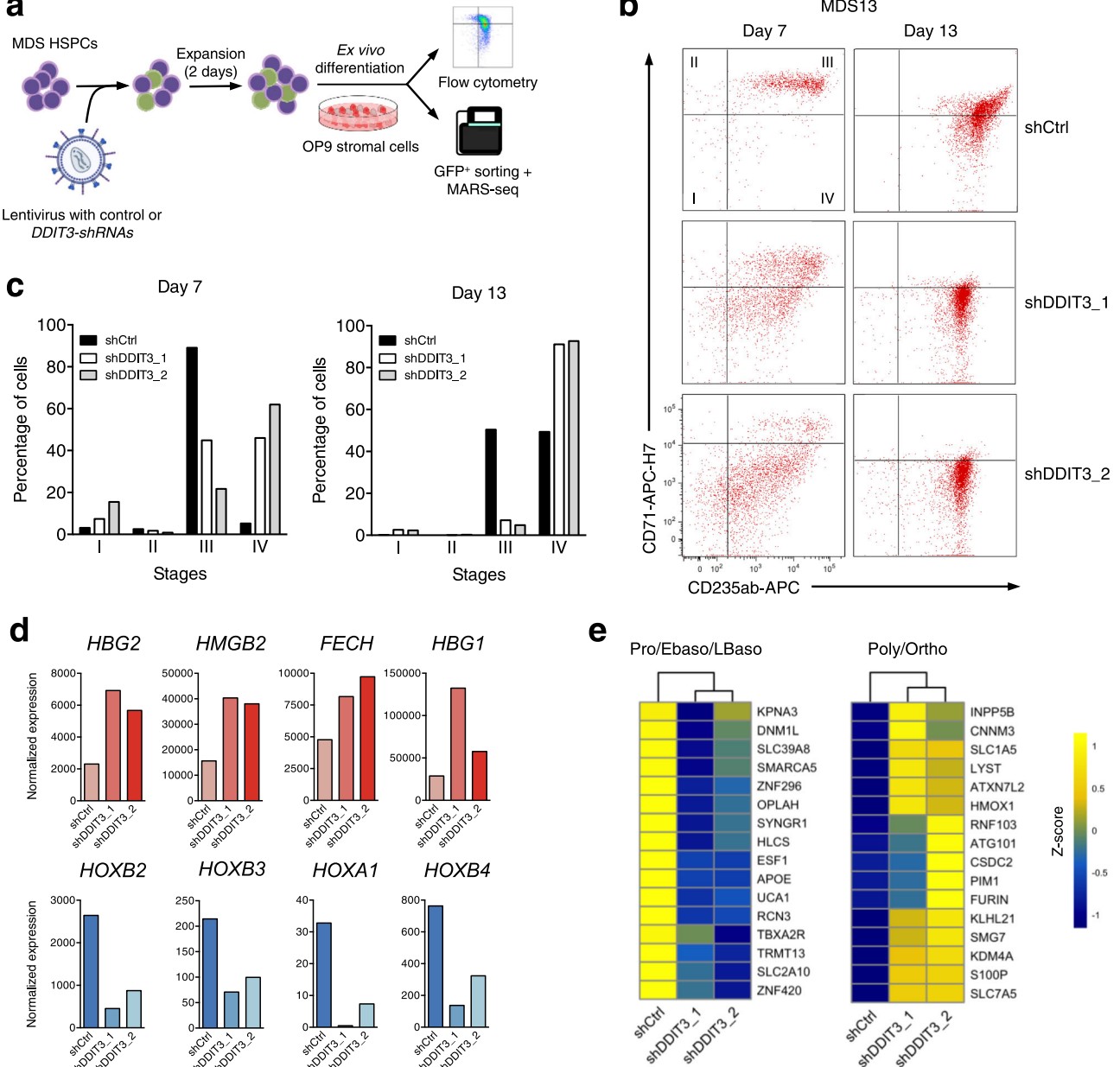

**Fig. 6 | *DDIT3* knockdown in MDS patients with anemia restores erythroid differentiation. a** Schematic representation of *DDIT3* knockdown experiments in CD34⁺ cells from patients with MDS showing anemia. Cells were transduced with a control or a *DDIT3*-targeting shRNA, and after 2 days of infection, cells were subjected to ex vivo liquid culture differentiation over OP9 stromal cells. The differentiation state was evaluated by flow cytometry and MARS-seq analyses. Part of this figure was made in ©BioRender - biorender.com. **b** Flow cytometry charts representing advanced erythroid differentiation (CD71 and CD235a markers; stages I–IV) for cells from patient #13 harboring a control shRNA (shCtrl) or shRNAs targeting *DDIT3*, at the indicated time points. **c** Bar-plots representing the percentage of cells observed in **b** in stages I–IV at 7 and 13 days of differentiation. **d** Normalized expression of hemoglobin (top) and stem cell (bottom) genes of CD34⁺ cells from patient MDS13 transduced with a shRNA control or an shRNA targeting *DDIT3* and subjected to 7 days of ex vivo differentiation. **e** Heatmap of z-scores of genes characteristic of proerythroblasts, early and late basophilic erythroblasts (left), and of genes expressed in poly- and orthochromatic stages (right), for cells from patient MDS13 transduced with a shRNA control, or an shRNA targeting *DDIT3*, and subjected to 7 of ex vivo differentiation.

## Sample collection and cell isolation

Bone marrow aspirates were obtained from healthy young (*n* = 17, average = 20.53 y/o, range = 18–22 y/o), or older adults (*n* = 8, average = 67.5 y/o, range = 58–81) and untreated MDS patients (*n* = 12, average = 70 y/o, range = 51–87 y/o) after the study was approved by the research ethics committee of University of Navarra, and informed consent was obtained. MDS patients and healthy donors were not economically compensated for the samples donated. Patient's data were fully anonymized, and all patients provided informed written consent to have data from their medical records such as age, gender,

and diagnosis to be used for research purpose. Healthy controls were volunteers (healthy young adults) or patients undergoing orthopedic surgery (healthy older adults) (Supplementary Table S1). MDS samples were obtained from several sources in Spain: Clinica Universidad de Navarra, Hospital Universitario de Salamanca and Hospital Universitari Vall d'Hebron. Only non-treated MDS patients were included in this study, and their classification was performed according to the WHO criteria. Furthermore, in order to homogenize the group of patients, only MDS cases with multilineage (MDS-MLD) and single-lineage dysplasia (SLD-MDS) were included in this study. Patient clinical data

are shown in Supplementary Table S2. Ficoll-Paque (GE Healthcare #17-1440-093) density gradient centrifugation was performed to enrich for the mononuclear cell population of our samples after prior red blood cells lysis. HSCs were isolated by multiparameter FACS using a BD FACSAria™ IIu sorter. The gating strategy and antibodies used are detailed in Fig. S1A and Supplementary Table S3. CD34+ cells for ex-vivo differentiation assays were isolated using FACS as above, or using a CD34 Positive Selection Kit II (StemCell technologies, #17856), according to manufacturer instructions. In the last case, the purity of CD34+ cells was assessed by flow cytometry. Sample information and cell isolation strategy can be found in Tables S1, S2, and S3.

## Bulk RNA-seq

Bulk RNA-seq was performed following MARS-seq protocol adapted for bulk RNA-seq[50,51], with minor modifications. Briefly, 300–10,000 cells were directly sorted in 100 ul of Lysis/Binding Buffer (Invitrogen, A33562), vortexed and stored at −80 °C until further processing. Poly-A RNA was selected with Dynabeads Oligo (dT) (Ambion) and reverse-transcribed with AffinityScript Multiple Temperature Reverse Transcriptase (Agilent) using poly-dT oligos carrying a 7 bp-index. Up to 8 samples with similar overall RNA content were pooled together and subjected to linear amplification by IVT using HiScribe T7 High Yield RNA Synthesis Kit (New England Biolabs). Next, aRNA was fragmented into 250–350 bp fragments with RNA Fragmentation Reagents (Ambion) and dephosphorylated with FastAP (Thermo) following the manufacturer's instructions. Partial Illumina adaptor sequences[50] were ligated with T4 RNA Ligase 1 (New England Biolabs), followed by a second RT reaction. Full Illumina adaptor sequences were added during final library amplification with KAPA HiFi DNA Polymerase (Kapa Biosystems). RNA-seq libraries quality controls consisted in quantification with Qubit 3.0 Fluorometer (Life Technologies) and size profiles examination with Agilent's 4200 TapeStation System. Libraries were sequenced in an Illumina NextSeq 500 at a sequence depth of 10 million reads per sample.

## RNA-seq analysis

Demultiplexing of raw base call files (BCL) was done with the Illumina software bcl2fastq (v2.20). Reads were then mapped to the GRCh38 version of the human genome using STAR (v2.6) and the option -outSAMattributes NH HI AS nM XS. Quantification of sorted BAM files was performed with the featureCounts function, inside the R package RSubread (v1.32.4). The R/Bioconductor DeSeq2 package was used to normalize raw data and to identify statistically significant differentially expressed genes (FDR < 0.05 and |FC| > 2). Before differential gene expression analysis, genes without at least one sample presenting more than 50 reads were filtered out. The R/Bioconductor package MaSigPro[33] was applied to normalized data to identify gene expression dynamics in the aging−MDS axis. An alpha of 0.05 for multiple hypothesis testing was used. A k-cluster of 14 was selected, and resulting clusters were manually curated into 8 final clusters based on their similar profile of the median expression in each group. Gene ontology enrichments were determined for each cluster using DAVID ontology 48. Gene expression heatmaps and volcano plots were generated using the CRAN pheatmap and the R/Bioconductor EnhancedVolcano packages, respectively. Genes from several clusters and from ex vivo experiments were imported to Gene Set Enrichment Analysis tool. Curated gene sets were obtained from MSigDB and were tested using GseaPreranked module for enrichment in the list of genes.

## Mutational analysis

Myeloid mutational panels were performed in three different hospitals. Samples were analyzed either with a custom pan-myeloid panel targeting 56 myeloid genes (CUN), with the panel Oncomine Myeloid Research Assay (ThermoFisher Scientific) (Hospital Val d´Hebron) or

with an in-house custom capture-enrichment panel of 92 genes (Hospital Universitario de Salamanca). At CUN Universidad de Navarra, DNA was isolated from 400 μL of total buffy coat using QIAamp DNA Blood Mini Kit (Qiagen). DNA samples were quantified using Qubit dsDNA BR Assay Kit on a Qubit 3.0 Fluorometer (Life Technologies), and quality was assessed by DNA genomic kit on a Tape Station 4100 (Agilent Technologies). Samples were analyzed with a custom pan-myeloid panel targeting 56 myeloid genes described by Aguilera-Diaz et al.[52]. Libraries were carried out following manufacturer's instructions, quantified using the Qubit dsDNA HS Assay Kit on a Qubit 3.0 Fluorometer (Life Technologies), and quality was assessed using the D1000 Kit on the 4100 Tape Station (Agilent Technologies). Libraries were sequenced on a MiSeq Sequencer (Illumina). In the case of samples from healthy older adults, poly-Adenosine-depleted cell lysates from MARS-seq initial steps were stored at −80 °C for further isolation of gDNA. Briefly, cell lysates were incubated sequentially with 20 μg of RNase A (AM2271, Ambion) and 1.6 U of Proteinase K (P8107S, New England Biolabs). Upon purification, gDNA was quantified with Qubit 3.0 Fluorometer (Life Technologies) and its integrity examined with Agilent's 4200 TapeStation System. At Hospital Val d´Hebron, 40 genes and 29 fusion gene drivers associated with myeloid malignancies were studied by NGS using the Oncomine Myeloid Research Assay (ThermoFisher Scientific) and the ThermoFisher automated sequencing platform (Ion Chef and Ion S5 XL systems). DNA and RNA were extracted from mononuclear cells isolated from bone marrow aspirates and library preparation was performed using 20 ng of DNA and 100 ng of RNA. The mean sequencing depth of coverage was 2500×. Pathogenic/likely pathogenic variants and variants of unknown significance were selected based on a minimum variant coverage of 25 reads and a minimum allele frequency of 1%. The limit of detection for fusion genes was transcript level ≥0.1%. At Hospital Universitario de Salamanca, targeted-deep sequencing was performed using an in-house custom capture-enrichment panel (Nextera Rapid Capture Enrichment, Illumina) of 92 genes previously related to the pathogenesis of myeloid malignancies, according to a Nextera sequencing design using Illumina DesignStudio. Genomic DNA was obtained from all samples from mononuclear cells using a QIAamp DNA Mini Kit (Qiagen) according to the manufacturer's standard protocol, and libraries were prepared according to the manufacturer's instructions. Libraries were sequenced on Illumina NextSeq 500 and MiSeq sequencers. The mean coverage was 665× (range, 251–1198) where 99.5% of target regions were captured at a level greater than 100×. For true oncogenic somatic variant calling, a severe criterion for variant filtering was applied. Synonymous, noncoding variants and polymorphisms, present at a population frequency (MAF) ≥ 1% in dbSNP138, 1000 G, EXAC, ESP6500 and our in-house databases, were excluded. Similarly, those variants recurrently observed and, from visual inspection on the IGV browser, suspected of being sequencing errors were removed. The remaining variants were considered candidate somatic mutations based on the following criteria: (i) variants with ≥10 mutated reads; (ii) described in COSMIC and/or ClinVar as being cancer-associated and known hotspot mutations; and (iii) classified as deleterious and/or probably damaging by PolyPhen-2 and SIFT web-based platforms. In addition, VARSOME was also consulted.

## Cell line cell culture

The human 293T cell line was grown in DMEM medium supplemented with 10% FBS, 100 U/mL penicillin, and 100 μg/mL streptomycin. The human K-562 erythroleukemia and MM-1S multiple myeloma human cell lines were cultured in RPMI supplemented with 10% FBS, 100 U/mL penicillin, and 100 μg/mL streptomycin. The murine stromal cell line OP9 was cultured in Minimum Essential Medium-Alpha (α-MEM) supplemented with 16% FBS, 100 U/mL penicillin, and 100 μg/mL streptomycin, and 0.12 mM β-mercaptoethanol (Merck, #M3148). All cell

lines were maintained at 37 °C in a humid atmosphere containing 5% $CO_2$. Cell lines were obtained from the ATCC (293T: CRL-3216; K-562: CCL-243; MM.1S: CRL-2974; OP9: CRL-2749) and authenticated by short tandem repeat (STR) profiling 1 month before their use (Genomics department at CIMAlab).

### *DDIT3* overexpression and knockdown systems
*DDIT3* cDNA was amplified from the MM.1S cell line using the following primers: AATTGAATTCATGGCAGCTGAGTCATTGCCTTTC, and AATT GCGGCCGCTGCTTGGTGCAGATTCACCATTCG, and the Platinum™ Taq DNA polymerase High fidelity (Invitrogen, #11304011). Upon digestion with EcoRI (Takara, #1040A) and NotI (Takara, 1166A), and phosphorylation of the inserts using the T4 PNK (New England Biolabs, #M0201), the cDNA was ligated to the pCDH-MCS-T2A-copGFP-MSCV lentiviral vector (Systems Biosciences #CD523A-1). The empty vector was used as a negative control for overexpression experiments. For *DDIT3* knockdown, two different short hairpin RNAs (shRNAs) were selected from the TCR2 shRNA library (Broad institute): shRNA 1 (CTGCACCAAGCATGAACAATTCTCGAGAATTGTTCATGCTTGGTGCA GTTTTT) and shRNA 2 (AGGTCCTGTCTTCAGATGAAACTCGAGTTTCA TCTGAAGACAGGACCTTTTTTT). The genescript software tool (https://www.genscript.com/tools/create-scrambled-sequence) was used to generate a scramble shRNA: shScramble (AGAGACGAACAAG AGGAGGCTCGAGCCTCCTCTTGTTCGTCTCTTTTTT). Each shRNA was annealed and cloned into pSIH1-H1-copGFP shRNA lentiviral vector (System Biosciences #SI501A-A), which allowed for the expression of the shRNA and GFP. Lentiviruses were generated using the second-generation packaging vectors psPAX2 and pMD2.G (Addgene) and Lipofectamine 2000 (Thermofisher, #11668019) following the manufacturer's protocol[53]. Lentiviruses were collected 3 days after transfection and concentrated by ultracentrifugation at 26,000 rpm for 2.5 h.

### Immunoblotting
For immunoblot analyses, equal protein amounts were run on 12% SDS-PAGE gels and transferred to nitrocellulose membranes with 0.45 μm pore size. Membranes were fixed with glutaraldehyde 0.5% for 5 min. Then, after blocking in 5% non-fat milk in PBS with 0.1% Tween-20 (PBS-T), membranes were probed with primary antibodies, DDIT3 (L63F7 clone, Cell Signaling Technology, #2895S, 1:1000) and β-actin (Ac-15 clone, Sigma Aldrich, #A5441, 1:5000) overnight at 4 °C. Membranes were washed with PBS-T, and incubated with a secondary antibody (anti-mouse IgG, HRP-linked whole Ab Merck, #NA931V, 1:2000) for 1 h at room temperature. Chemiluminescence was detected using ECL reagents and a Biorad ChemiDoc imager. For immunoblot visualization and analysis ImageLab software was used.

### Immunofluorescence
CD34+ cells were transduced with the overexpression or knockdown systems as described above. Two days after transduction, cells were subjected to the differentiation system (see scRNA-seq experiment upon *DDIT3* overexpression) for two additional days, and then were transduced cells were sorted based on GFP. Sorted cells were seeded on coverslips previously treated with poly-l-lysine (Sigma P4707), and incubated in serum-free medium for 30 min at 37 °C. Afterwards, cells were fixed with 4% paraformaldehyde (Sigma 28908) for 10 min, blocked with 10% BSA 0.5% Tween20 PBS for 30 min at RT, and incubated with anti-DDIT3 antibody (Cell Signaling, #2895S) at 1:400 in incubation solution (5% BSA 0.1% Tween20 PBS) overnight in a humidified dark chamber. After washes with 0.1% Tween20 PBS cells were incubated with alexa568 DK-anti mouse antibody (Thermofisher A-10037) at 1:500 in incubation solution for 1 h at RT. Cells were washed as above, treated with the VECTASHIELD® Antifade Mounting Medium with DAPI (Vector laboratories H-1200-10), and visualized under a fluorescence microscope (AX10 Zeiss).

### Transcriptome profiling of primary healthy HSCs upon *DDIT3* overexpression
Lentiviruses were added to freshly sorted HSCs isolated from healthy young adults in 96-well plates containing StemSpan™ SFEM media (Stem Cell Technologies, #09600), 50 ng/mL SCF (Peprotech, #300-07), 10 ng/mL Flt3-L (Peprotech, #300-19), 10 ng/mL Tpo (Peprotech, #300-18), and 6 μg/mL polybrene (Millipore). Cells were spinoculated at 2500 rpm and 32 °C for 90 min. Two days after transduction, GFP+ cells were FACS-sorted using a BD FACSAria™ IIu in Lysis/Binding Buffer for Dynabeads™ mRNA Purification Kits (Invitrogen), and samples were snap-frozen and kept at −80 C until MARS-seq was performed.

### Colony assays and ex vivo lineage differentiation assays
CD34+ cells were stimulated for 3 h prior transduction in StemSpan (Stem cell Tech., # 09655) supplemented with 150 ng/mL SCF, 150 ng/mL Flt-3, 10 ng/mL IL-6, 25 ng/mL G-CSF, 20 ng/mL TPO, and 1% HEPES. Cells were then transduced with the lentiviral particles overnight at multiplicity of infection of 30[54], and washed with StemSpan before switching for an expansion medium, composed of StemSpan supplemented with 150 ng/mL SCF, 150 ng/mL FLT3-L and 20 ng/mL TPO. After 4 days, cells were sorted based on GFP and CD34 (BD Biosciences, #347213) expression on a FACSAria Fusion (BD Biosciences) to perform colony, ex vivo lineage differentiation assays and transcriptomic analyses. For colony assays, cells were seeded in methylcellulose (Stem Cell Technologies, #4434), and incubated at 3% oxygen. Colonies were scored at 14 days. For erythroid lineage differentiation experiments, freshly FACS sorted CD34+ GFP+ cells were incubated in erythroid differentiation medium, consisting of Stem Span SFEMII (Stem Cell Technologies, #09605) supplemented with 25 ng/mL SCF, 3IU/mL EPO (Peprotech, #100-64), and 50 ng/mL IGF1 (Peprotech, #100-11). Erythroid differentiation markers (Biolegend CD71 #113807 1:100 and CD235a #349115 1:100) were assessed by flow cytometry at 7, 10 and 14 days. For transcriptomic analyses, cells were harvested in Lysis/Binding Buffer at days 10 and 14 and kept at −80 °C until RNAseq was performed. For granulocytic differentiation assays, CD34+ GFP+ cells were cultured in granulocytic differentiation medium (SCF 25 ng/mL, PeproTech, Cat 300-07; GM-CSF 10 ng/mL, PeproTech, Cat 300-03) for 14 days. Cells were stained with antibodies specific for human antigens (CD11b APC RRID:AB_10561676 1:100; CD14 PE-Cy7, RRID:AB_1582277 1:100; CD15 BV786, RRID:AB_2740635 1:100; CD13 PE, RRID:AB_395795 1:100; CD45 APC eFluor780, RRID:AB_1944368 1:100) and DAPI. Cells were immunophenotyped by using Fortessa flow cytometer (BD Biosciences, Oxford, UK) at day 10 and 14.

### Apoptosis, proliferation, and cell cycle analysis
Apoptosis was assessed by staining cells with Annexin-V PE Cy7 (Biolegend, cat 640912, dilution 1:100) and DAPI in 1× Annexin V binding buffer (Biolegend, cat422201). Proliferation was monitored using counting beads (Invitrogen, C36950). For cell cycle analysis, $10^4$–$10^5$ cells were harvested and washed in PBS. Fixation/Permeabilization solution (BD Cytofix/Cytoperm™ #554714) was added per sample (250 μL) and incubated for 20 min at 4 C. Cells were washed twice in 1× BD Perm/Wash™ buffer, and resuspended in 1 mL of BD Perm/Wash™ buffer + DAPI and incubated at 4 °C for 30 min in the dark. Analyses were performed on Fortessa flow cytometer (BD Biosciences).

### scRNA-seq of ex-vivo differentiated cells upon *DDIT3* overexpression
Freshly sorted CD34+ cells were transduced with lentiviruses as described above. Two days after transduction, cells were transferred into 6-well plates containing OP9 cells, in 2 mL of OP9 medium supplemented with cytokines to induce differentiation: 10 ng/mL SCF, 10 ng/mL Tpo, 2 ng/mL Flt3-L, 50 ng/mL EPO, 5 ng/mL IL-3 (Peprotech, #200-03), 10 ng/mL IL-6 (Peprotech, #200-06), 5 ng/mL G-CSF

(Peprotech, #300-23) and 5 ng/mL GM-CSF (Peprotech, #300-03). OP9 cells, media, and cytokines were renewed every 2–3 days. After 14 days of differentiation, transduced cells (GFP⁺) were sorted in iced cold PBS 1× and 0.05% BSA using a MoFlo AstriosEQ Sorter (Beckman Coulter), and scRNAseq was performed. Samples were processed according to 10× Genomics Chromium Single Cell 3′ Reagent Guidelines. Briefly, approximately 16,000 and 5400 (for control and *DDIT3*-overexpression, respectively) FACS-sorted cells were loaded on a Chromium Controller instrument (10x Genomics) to generate single-cell gel bead-in-emulsions (GEMs) at a concentration of 1000 cells/μl. In this step, each cell will be barcoded with primers containing an Illumina R1 sequence, a 16 bp 10× barcode, a 10 bp Unique Molecular Identifier (UMI) and a poly-dT primer sequence. After GEM generation, GEM-RT was performed with partitioned cells resulting in barcoded, full-length cDNA from poly-adenylated mRNA. After GEM-RT, GEMs were harvested and the cDNA was amplified and cleaned up with magnetic beads. Enzymatic Fragmentation and Size Selection was used to optimize cDNA amplicon size prior to library construction. P5, P7, a sample index (i7), and R2 (read 2 primer sequence) were added during library construction via End Repair, A-tailing, Adaptor Ligation and PCR. Libraries quality control and quantification was performed using Qubit and bioanalyzer. Sequencing was performed using NEXTseq500 (Illumina).

## scRNA-seq analysis

Raw data was demultiplexed using the 10X software cellranger mkfastq. Then, fastq files were aligned to the GRCh38 genome and count matrix created with cellranger counts. We removed cells with more than 15% of the counts on mitochondrial genes and established an upper threshold for the number of counts in order to delete doublets (control: 75,000, *DDIT3*: 60,000). Both samples were integrated following the Seurat pipeline. Counts were log normalized and scaled and integration was performed on the 2000 genes with highest variability, using 50 dimensions. Integrated data was rescaled, regressing cell cycle effect. Next, we obtained the UMAP coordinates based on the first 20 principal components and performed an initial clustering with resolution of 0.4. After exploration of the clusters, we removed those with myeloid progenitor markers and low quality (low counts and high ribosomal gene expression). We repeated the previous analysis on the subset data. Finally, we established cell identity using as reference published bulk RNAseq data from different stages of erythropoiesis[39]. Identities were predicted using the singleR algorithm. Then, we performed differential expression between conditions, for each identity separately. We used the MAST test implemented in Seurat and extracted the complete gene list to use as input for GSEA. We calculated RNA velocity using velocyto.py and the dynamical model implemented in the scVelo Python library, with the default options. Pseudotime analysis was carried out with Palantir (implemented in Python), based on the normalized data for the 2000 highly variable genes and for each sample independently. Starting cells were added as input based on their UMAP coordinates and terminal states were automatically determined by the algorithm. Next, we inferred gene expression trends along pseudotime using generalized additive models. Finally, we built gene regulatory networks using SimiC. This algorithm infers a joint network for multiple states and then allows for the detection of differential activity between states. We performed data imputation and selected the 100 most variable TFs and 1,000 more variable genes as targets for this analysis. After parameter tuning with cross-validation, we set lambda1 = 0.01, lambda2 = 0.1. We extracted regulons by calculating association weights for each TF and target, and filtering out small weight. We then computed the weighted area under the curve (wAUC) to measure regulon activity per cell. Lastly, we tested for differences in regulon activity between control and *DDIT3* overexpressing cells, performing a Kolmogorov–Smirnov test for the wAUC distributions.

## Transcription factor binding site analysis

The factors driving regulons with decreased activity in *DDIT3*-overexpressing cells were analyzed using CiiiDER, a TF binding site analysis software[55]. A background gene list was generated by selecting the TFs guiding regulons with low dissimilarity score. The selected deficit threshold value in the analysis was 0.15 and the used matrix of TF binding sites was Jaspar_2020_core_vertebrates. The TF enrichment analysis was performed using the Fisher´s exact test included in the algorithm to obtain the significantly over- and under-represented TF binding sites when *DDIT3* was overexpressed.

## Immunoprecipitation

Control or *DDIT3*-overexpressing K562 cells were lysed in cell lysis buffer (Sodium Orthovanadate 5 mM (New England Biolabs #P0758S), cell lysis buffer (Cell signaling #9893), complete protease inhibitor cocktail (Merck #11836153001)) for 30 min on ice and subsequently centrifuged at 4 °C 13,200 RPM for 10 min. One mg of protein per IP was taken and the supernatants were precleared by incubation with Protein-G Dynabeads (ThermoFisher scientific #10004D) for 2 h at 4 °C. Antibody-coated beads were prepared by incubating mouse anti-human C/EBPβ antibody (10 μg per sample, Santa Cruz Biotechnology, (H-7): sc-7962), mouse anti-human C/EBPγ (10 μg per sample, Santa Cruz Biotechnology, (S2): sc-517003), CHOP (1:100 per sample Cell Signaling #L63F7) or control rabbit IgG (10 μg per sample; Abcam, #ab37415) with pre-washed Protein-G Dynabeads 40 min at room temperature. The antibodies were crosslinked to the dynabeads using in dimethyl pimelimidate × 2HCl 20 mM (ThermoFisher #21667) in triethanolamine 0.2 M at 20 °C for 30 min. IP was carried out overnight at 4 °C. The Dynabeads were washed 4 times with cell lysis buffer. Washed beads were resuspended in 1× Laemmli buffer and boiled at 70 °C for 10 min before resolving on an 12% SDS-PAGE, and immunoblotting with the previously indicated antibodies. The immunoblot detection of primary antibodies was done using Clean Blot IP Detection reagent (HRP) (ThermoFisher #21230) and ECL anti mouse IgG, HRP linked whole Ab (Merck #NA931).

## Statistics and reproducibility

Data are shown as mean, and error bars correspond to standard deviation (SD) unless otherwise indicated. Statistical significance was determined using non-parametric Mann–Whitney test, Multiple t test corrected for multiple comparisons using the Holm-Sidak method, or Wilcoxon signed rank test. Graphs were made using GraphPad Prism software. Each statistical method used is specified in the corresponding figure legend. All methods were analyzed using two-tailed analyses, and a *p* value < 0.05 was considered statistically significant. All measurements were taken from biologically independent samples. Assays, including those shown as representative (immunoblots, immunofluorescence and colony formation micrographs), were repeated at least three independent times unless otherwise stated in the figure legend. The number of human samples, biological experiments or cells (for scRNA-seq data) analyzed in each subpanel is indicated in the figure legends. No statistical method was used to predetermine sample size, and no data were excluded from the analyses performed. When possible, investigators were blinded during the analysis of several of the ex vivo experiments performed (i.e., colony formation, apoptosis, ex vivo differentiation assays).

## Reporting summary

Further information on research design is available in the Nature Portfolio Reporting Summary linked to this article.

## Data availability

The sequencing data generated in this study have been deposited in the Gene Expression Omnibus (GEO) database under accession code GSE183328. The in vitro data generated in this study are provided in

the Supplementary Information/Source Data file. The previous publicly available datasets data used in this study are available in the Gene Expression Omnibus (GEO) database under accession codes: GSE114922, and GSE19429. Finally, to classify scRNA-seq cells, bulk RNAseq data from erythroid differentiation were obtained from GSE107218. The GRCh38 assembly of the human genome used is available at NCBI, under the accession code NCBI:GCA_000001405.27. Source data are provided with this paper.

## Code availability

Bioinformatic code regarding the single-cell RNA-seq analyses performed in this work can be found at https://github.com/mainciburu/DDIT3_2022; https://doi.org/10.5281/zenodo.7299427[56].

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

## Acknowledgements

This work was supported by the Instituto de Salud Carlos III and co-financed by ERDF A way of making Europe (PI17/00701, and PI20/01308) (F.P.) and (PI19/00726) (T.E.), CIBERONC (CB16/12/00489) (F.P.); Gobierno de Navarra (AGATA 0011-1411-2020-000010/0011-1411-2020-000011 and DIANA 0011-1411-2017-000028/0011-1411-2017-000029/0011-1411-2017-000030) (F.P.); Fundación La Caixa (GR-NET NORMAL-HIT HR20-00871) (F.P.); and Cancer Research UK [C355/A26819], FC AECC and AIRC under the Accelerator Award Program, and MCIN/AEI/10.13039/501100011033 and by ERDF A way of making Europe [RTI2018-101708-A-I00] (M.H.). Moreover, this work was supported by PhD fellowships from Gobierno de Navarra (0011-0537-2019-000001) (N.B.), and (0011-0537-2020-000022) (A.D.-M.); a PhD fellowship from Ministerio de Ciencia, Innovación y Universidades (FPU18/05488) (M.A.); an Investigador AECC award from the Fundación AECC (INVES19059EZPO) (T.E.), H2020 Marie S. Curie IF Action, European Commission, Grant Agreement No. 898356 (M.H.); and by grants RYC2018-025502-I (A.A.) and PRE2018-084542 (R.R.) funded by MCIN/AEI/10.13039/501100011033 and by ESF Investing in your future. We particularly acknowledge the patients and healthy donors for their participation in this study, and the Biobank of the University of Navarra for its collaboration.

## Author contributions

T.E. and F.P. conceived and supervised the study. A.A., D.A., J.M.L.-E., M.S.-J., T.J., F.L.-C., S.M., F.S.-G., A.M., M.J.M., B.T., S.H., M.D.-C., D.V., B.P., and M.S.J. collected clinical samples and data. A.V.-Z. and P.S.M.-U. generated sequencing data. T.E., N.B., P.G.-O., C.P., S.S., R.R.-H., A.A., and K.R.-P. performed ex vivo and in vitro experiments. N.B., M.A., J.P.R., R.O., L.C.-L., G.S., A.D.-M., M.H., F.B.-C., A.R.-M., M.T.M, and D.L.-A. performed the data analysis. T.E. and F.P. wrote the manuscript.

## Competing interests

The authors declare no competing interests.

## Additional information

[1]Department of Hematology-Oncology, CIMA Universidad de Navarra, Instituto de Investigación Sanitaria de Navarra (IDISNA), Pamplona, Spain. [2]Centro de Investigación Biomédica en Red de Cáncer, CIBERONC, Instituto de Salud Carlos III, Madrid, Spain. [3]Department of Hematology, Clínica Universidad de Navarra, Universidad de Navarra and CCUN, 31008 Pamplona, Spain. [4]Department of Haemato-Oncology, Barts Cancer Institute, Queen Mary University of London, London, England, UK. [5]Center for Cooperative Research in Biomaterials (CIC biomaGUNE), Basque Research and Technology Alliance (BRTA), San Sebastian, Spain. [6]Ikerbasque, Basque Foundation for Science, Bilbao, Spain. [7]Institute for Systems Genetics, NYU School of Medicine, New York, NY, USA. [8]Department of Orthopedic Surgery and Traumatology, Clínica Universidad de Navarra, Universidad de Navarra and CCUN, 31008 Pamplona, Spain. [9]Department of Hematology, Hospital Universitario de Salamanca-IBSAL, Universidad de Salamanca, Salamanca, Spain. [10]Department of Hematology, Experimental Hematology, Vall d'Hebron Institute of Oncology (VHIO), Hospital Universitari Vall d'Hebron, Barcelona, Spain. [11]Computational Biology Program, Institute for data science and artificial intelligence (datai), CIMA Universidad de Navarra, Instituto de Investigación Sanitaria de Navarra (IDISNA), Navarra, Spain. [12]Centre for Genomics and Computational Biology, Barts Cancer Institute, Queen Mary University of London, London, UK. [13]Department of Pathology, NYU School of Medicine, New York, NY, USA. [14]These authors contributed equally: Nerea Berastegui, Marina Ainciburu. [15]These authors jointly supervised this work: David Lara-Astiaso, Teresa Ezponda, Felipe Prosper. ✉e-mail: tezponda@unav.es; fprosper@unav.es

