## [Peer Review File · Nature Communications]

The transcription factor DDIT3 is a Potential Driver of Dyserythropoiesis in Myelodysplastic SyndromesREVIEWER COMMENTS

Reviewer #1 (Remarks to the Author): with expertise in HSC, haematopoiesis

In their manuscript Transcriptional regulation of HSCs in Aging and MDS reveals DDIT3 as a Potential Driver of Dyserythropoiesis, Berastegui et al propose a role for DDIT3 in myelodysplastic syndrome pathogenesis through interference with normal progression of erythroid differentiation and relative enrichment of early erythroid differentiation states. The authors identify DDIT3 as being specifically overexpressed in MDS samples, beyond effects of patient age. They employ a combination of overexpression and knockdown approaches in healthy and MDS CD34+ cells respectively, and bulk and single-cell transcriptional profiling of in vitro differentiation systems, to associate DDIT3 with a relative delay in erythroid lineage progression that captures aspects of MDS-associated transcriptional signatures. The authors suggest that DDIT3 may act through interference with CEBPB and CEBPG-driven erythroid transcriptional programmes, a pattern reminiscent of the putative dominant negative effect of DDIT3 on CEBPA transcriptional activity, which results in myeloid differentiation block, transient self-renewal and erythroid lineage bias of mouse GMP. While some of the data presented suggest a possible contribution of DDIT3 to MDS pathogenesis, the results are preliminary and incomplete at present, and require additional experiments and careful validation to positively affirm this conclusion.

MAJOR POINTS

1. The authors should include validation of DDIT3 overexpression and knockdown at the protein level to confirm the biological relevance of the transcriptional and functional phenotypes observed.

2. The authors evaluate differentiation of healthy and MDS CD34+ cells with respective enforced and knockdown DDIT3 expression using various in vitro differentiation systems, namely liquid cultures for erythroid-specific or myeloid-specific differentiation (healthy cells / overexpression), OP-9 co-culture in the presence of multilineage-supporting cytokines (DDIT3 overexpression for scRNA-seq analysis), and likely (this is not explicitly stated) the same OP-9 co-culture system for DDIT3 knockdown in MDS samples. Additionally, they use CFC assays for analysis of DDIT3-overexpressing CD34+ cells only, and phenotypic analysis relies almost exclusively on flow cytometry analysis of erythroid progression.

2.1 The authors should include more detailed data to validate and compare the various differentiation systems employed using healthy untransduced cells – including cell expansion, cell death and progression of erythroid / myeloid differentiation and loss/preservation of HSPC at different timepoints.

2.2 Given the centrality of the opposing effects of DDIT3 overexpression and knockdown to the conclusions, it would be important to demonstrate that opposing effects are achieved in the same culture systems – can MDS samples be assessed and how do they behave in erythroid liquid culture? is the erythroid delay observed in multilineage OP-9 cultures of CD34+ cells with DDIT3 overexpression? How do MDS CD34+ behave in CFC assays upon DDIT3 knockdown? And have the authors attempted to perform CFC re-plating of healthy or MDS cells with or without DDIT3 as a means to assess DDIT3 contribution to in vitro transformation? Given the data in mouse GMP, this could also be a contributing factor to MDS pathogenesis.

2.3 The authors state the importance of analysing MDS signatures in purified HSC, yet perform functional analyses in total CD34+ cells (see also MINOR POINT 1). While I understand that LV transduction of purified HSC produces limiting numbers of cells for the various analyses – namely transcriptomic – performed, possible effects of cell selection, death or biased expansion are not explored. Given the specific association of DDIT3 with a relative expansion or differentiation delay of early erythroblasts, but without evidence of an absolute defect in terminal differentiation, the authors should include timecourse assessments of differentiation status, cell cycle and apoptosis, as well as more detailed analysis of single-cell initiated cultures. This level of detail will help eliminate potential confounder effects of cell selection, and may reconcile different and some contradictory effects on erythroid programmes observed in this and other studies.

3. The authors report DDIT3 overexpression in only a subset of MDS samples. Can the authors please include more detail on the levels of baseline expression and knockdown achieved in the individual samples #13-17 reported in this study? Were there other samples with similar DDIT3 expression that responded differently / not responded to DDIT3 knockdown? Have the authors

attempted to overexpress DDIT3 in low-expressing samples?

MINOR POINTS

1. MARS-seq analysis of MDS vs Young / Old healthy HSC shows that DDIT3 to be mildly albeit significantly over-expressed in MDS samples. While the mild overexpression may be due to sample heterogeneity with only a subset of MDS HSC showing detectable expression of DDIT3, other publicly available datasets do not seem to capture DDIT3 overexpression. The authors suggest that this may be due to a specific effect in highly purified HSC, which is not captured by analysis of bulk CD34+, but (i) provide no additional characterisation of the MDS samples in which DDIT3 was overexpressed, and (ii) perform overexpression / knockdown experiments in total CD34+ cells which may be confounded by cell heterogeneity. On this point, it would be helpful if the authors could give some insight into the perceived specificity of DDIT3 pathogenic effects, for example in terms of characteristic genetic alterations or transcriptional contexts in which DDIT3 may be relevant to MDS.
2. The CFC results presented in this study are of technical replication for individual samples, with 3 different experiments analysed separately and shown in the main and supplementary figures. The authors should instead analyse the statistical significance of biological replicates for robustness of their conclusions.
3. DDIT3 has also been implicated in HSC maintenance through regulation of ER stress. Have the authors investigated ER stress signatures in the context of DDIT3 overexpression / knockdown and their contribution to MDS pathogenesis?
4. In Supplementary Table S1 YH should be Young_Healthy, not Young_Elderly.

Reviewer #2 (Remarks to the Author): with expertise in aging, HSC

Review of NCOMMS-21-47839

In this study the group of Felipe Prosper and Teresa Ezponda aims at elucidating the transcriptional changes occurring in human HSCs during aging and in MDS, a hematopoietic malignancy mainly affecting elderly individuals. The biological processes extrapolated by their transcriptional analysis indicate that the transcriptional profile of MDS samples differs from the one reported in aged HSC. The authors next identify a Transcription Factor, DDIT3, specifically up-regulated in MDS samples and show that when overexpressed in healthy HSC DDIT3 leads to ineffective erythropoiesis, a condition also observed in MDS patients. The study is of translational relevance and the transcriptomic data are presented in a clear manner with conclusions drawn from experiments that are largely coherent with data. However, there are some open points, mainly regarding the mechanistic part of the study and outlined below, that preclude publication in its current form.

Major revisions

DDIT3 overexpression in healthy HSPCs and its downregulation in MDS give interesting and consistent results. However, the mechanisms behind the rescue of phenotype remains elusive. It would be intriguing to analyze DNA damage markers (53BP1/γH2AX/pRPA) in a time-course from day 1 to day 8 or beyond upon DDIT3 overexpression in healthy donors or following its downregulation in MDS as DNA replication stress and accumulation of DNA damage may play a key role in MDS pathogenesis as well as during aging. I would expect an upregulation of replication stress and accumulation of double strand breaks as genes involved in the regulation of DSB repair are downregulated upon DDIT3 overexpression. Interestingly, the authors should describe and comment the impact of DDIT3 on inflammation and upregulation of inflammatory gene categories, given the prominent role of inflammatory pathways in aging and age-associated diseases,

including MDS.

The authors discuss that upregulated levels of DDIT3 in MDS could sequester CEBPB and CEBPG, resulting in a diminished expression of their target genes, including TFs with key roles in erythropoiesis. In this context, further experiments would be necessary to corroborate the mechanistic part of the manuscript. It would be interesting to dissect association of CEBPB and CEBPG in promoters of TFs involved in erythropoiesis by ChIP or to verify a direct association of DDIT3 and CEBPB and CEBPG by Immunoprecipitation upon DDIT3 overexpression and downregulation.

(lines 132-137): Authors claim that HSC aging and transition from elderly to MDS HSCs is driven by different mechanisms. This conclusion derives only from a part transcriptome data and from a small cohort of MDS patient with a low risk profile. However, later on (lines 165-170) they detect some factors known to be involved in the development of myeloid malignancies both in aged and MSD HSC and they conclude that "aging-derived transcriptional changes predispose HSCs for transformation", which seems to be in contrast with the sentence above. This should be better explained or better characterized by experiments with samples from younger MDS patient or by separating dataset of MSD patients by age (they put together data of 50 years old patients and 80 years old and I wondered if there is any differential response).

To further validate the transcriptome data and broaden the main conclusions of the paper the authors should try to overexpress in normal HSCs 1/2 additional transcription factors found upregulated in MDS listed in Figure 2F and see what happens in terms of biological phenotype and response. I would like to see if there is a similar effect in colonies and erythropoiesis upon overexpressing other genes with a similar function such as SERTAD2, RELA, EBF4 and ZNF219 in HSCs derived from healthy donors or if this effect is specific of DDIT3. Conversely, experiments testing the role of genes detected in figure 2C such as MCM4 and MCM7 in aged or MDS samples should be performed to observe if there is a rescue of hematopoietic dysfunctions (including ineffective erythropoiesis) in aged or MDS cells. In the same line, having established a nice differentiation protocol in liquid culture coupled with clonogenic assays (Fig. 3A and Fig. 4), it would be important to see how aged and MDS samples behave in response to these culture-induced activating stimuli.

Finally, although challenging, experiments with in vivo modeling of the role of DDIT3 in humanized mouse models for MDS (<https://doi.org/10.1038/s41467-018-08166-x>) or in ossicle growth of cancer cells (doi: 10.3324/haematol.2019.233320) will further strengthen the translational relevance of the manuscript.

Minor points:

As Clonal Hematopoiesis of indeterminate potential (CHIP) has emerged as key regulator of HSC aging and age-associated malignancies, the authors should also report whether the aged samples under study have clonal expansion and/or CHIP mutations that could explain the transcriptional phenotypes observed.

Line 101. Given the phenotypic markers used (CD34+, CD38-, CD90+ CD45RA-) I would substitute "highly purified HSCs" with HSCs enriched subsets here and throughout the manuscript.

Line 149. A Gene ontology analysis is presented in Figure 1G and the related GO terms are reported in Table 6. Adding the manual curated list in the same table would help data reproducibility.

Line 265. The authors present some results regarding the pseudotime analysis (Fig 5D), it would be interesting to visualize the pseudotime trends for each cell in the UMAP.

In the scRNA analysis the myeloid clusters are eliminated because DDIT3 is a transcription factor for erythropoiesis. Can the authors add something or discuss more in depth about changes in differentiation within myeloid clusters?

Color code for panel MMP2 in Fig. 2C across young/elderly/MDS is not consistent with the other panels

Please specify the source of CD34+ cells used in the study across all ex-vivo experiments (especially in Fig. 3 and 4).

Reviewer #3 (Remarks to the Author): with expertise in MDS, erythropoiesis

In this paper, the authors study highly purified normal and malignant stem cells based on surface markers from young and old subjects and lower-risk MDS patients. They examine the transcriptomes from these subjects and grouped these into 8 different expression patterns. They suggest that aging results in a transcriptomic state that predisposes to developing a myeloid malignancy, upon which additional changes result in MDS. In particular, they focus on the transcription factor DDIT3, which when overexpressed resulted in an MDS-like transcriptional state and suppression of erythropoiesis in an in vitro model.

This is an interesting paper and the experiments are generally well conducted. The selection of purified cells for these studies is a strength, and the finding that DDIT3 may be responsible for some cases of MDS-related anemia is convincing.

Less convincing is the idea proposed that the transcriptional state of old stem cells is more like that of MDS stem cells than young stem cells. Several questions arise.

First, given that the authors are flow-sorting cells based on an immunophenotype, it is not clear how they are separating normal from malignant stem/progenitor cells for their transcriptomic studies. Some detail should be provided.

Second, while the PCA plot does seem to separate young stem cells from old and MDS (Fig 1c), the hierarchical clustering results (Fig. S1B) are not nearly as convincing. 4 MDS patients are in the most distant cluster from the rest of the samples, all on their own (left-most group). 4 MDS samples are in a cluster with elderly patients only (2nd group from the left). 1 MDS sample is in a mixed cluster with elderly and young patients (3rd group from the left). 3 MDS samples are in a cluster with young patients only (the rest of the groups). For any branch in the tree, it can be rotated 180 degrees around its axis. If any of the higher-level branches were rotated in this way, then the apparent separation of young and old samples, and the apparent clustering of MDS with old, would not be evident any more. Further Differential expression analysis showed more genes deregulated between MDS and elderly healthy samples than between healthy elderly and young HSCs.

At the bottom of p. 6, the authors refer to Figs. 2I and J, but there is no Fig. 2I or J. Similarly other labels are incorrect, and should be checked through the manuscript (e.g. Fig. 2K).

It is not clear why stress inflammation goes down with MDS compared to old in Fig. 1G when MDS is associated with an inflammatory state?

The choice of clusters that the authors choose to compare to make there is inconsistent and lacks a clear rationale. For example, In Fig. 2A and B, they mainly examine and compared clusters C1, C2, 5 and 6, but 1 and 2 show no difference between aged and MDS, so it is unclear where the Cancer associated genesets are coming from? It appears that clusters C3 and C4 are perhaps the most important clusters to examine closely as this is where unique MDS-related expression patterns reside.

I have some concern about the use of the MaSigPro R package for comparison of the healthy young, healthy old, and MDS groups (Fig. 1F, G). The package is specifically designed for time series data, which would imply that the young, old, and MDS samples would have to be harvested from the same group of individuals across time (this was not the case). The authors to explain their statistical reasoning that MaSigPro is appropriate for comparing 3 groups of distinct

individuals as if it were a time series. Particularly concerning is the notion that the MDS group naturally comes after the Elderly group in time. Are the MDS patients actually older than the Elderly healthy controls? I understand why it seems logical that MDS comes after Elderly, based on what we know about MDS as a disease of the elderly, but it's a big assumption for a time series analysis.

A propos the above statements, the authors should show the GSEA enrichment plots for Elderly vs MDS as this would be more relevant to their thesis than comparing Young to MDS.

In Fig. 4, it is not clear what the expression of DDIT3 in stages of erythroid differentiation means given it is overexpressed as a transgene. Does this mean that the transgene under an exogenous promoter is upregulated or are the authors looking specifically at endogenous DDIT3?

As well, DDIT3 is not seen to be expressed in the CD34 cells - it is possible that the transduction is mainly happening in erythroid progenitors hence the effect seen. In other words, are the authors sure that they are able to transduce HSC rather than progenitors, hence observing a lineage-specific effect rather than an effect that is actually reflective of what is happening in the HSC?

In Fig. 4F it is not evident that there is "shorter, thinner and less dense streamlines" in the DDIT3 populations. In which trajectories are they seeing this? The authors should quantify this effect and analyze it statistically.

In Fig 5F, the authors provide a list of differentially expressed factors on p. 10 of the text, but Not all the data are shown. They should either limit the list of factors or show data for all.

RESPONSE TO REVIEWER COMMENTS

Please, find below a detailed point-by-point answer to all the comments raised in the review process and an explanation on how we have addressed each of the criticisms and suggestions in the new version of the manuscript and supplemental material. Our responses are written in blue font, while changes in the manuscript are indicated in red font.

Reviewer #1 (Remarks to the Author): with expertise in HSC, haematopoiesis

In their manuscript Transcriptional regulation of HSCs in Aging and MDS reveals DDIT3 as a Potential Driver of Dyserythropoiesis, Berastegui et al propose a role for DDIT3 in myelodysplastic syndrome pathogenesis through interference with normal progression of erythroid differentiation and relative enrichment of early erythroid differentiation states. The authors identify DDIT3 as being specifically overexpressed in MDS samples, beyond effects of patient age. They employ a combination of overexpression and knockdown approaches in healthy and MDS CD34+ cells respectively, and bulk and single-cell transcriptional profiling of in vitro differentiation systems, to associate DDIT3 with a relative delay in erythroid lineage progression that captures aspects of MDS-associated transcriptional signatures. The authors suggest that DDIT3 may act through interference with CEBPB and CEBPG-driven erythroid transcriptional programmes, a pattern reminiscent of the putative dominant negative effect of DDIT3 on CEBPA transcriptional activity, which results in myeloid differentiation block, transient self-renewal and erythroid lineage bias of mouse GMP. While some of the data presented suggest a possible contribution of DDIT3 to MDS pathogenesis, the results are preliminary and incomplete at present, and require additional experiments and careful validation to positively affirm this conclusion.

MAJOR POINTS

1. The authors should include validation of DDIT3 overexpression and knockdown at the protein level to confirm the biological relevance of the transcriptional and functional phenotypes observed.

We thank the reviewer for the comment, and agree that a demonstration that the system works at the protein level is important. We have taken two different approaches to answer this point:

1. Immunoblot in cell lines

Firstly, and due to the limited cell number of primary CD34+ cells, and therefore, the impossibility of performing immunoblot on such samples, we have transduced cell lines with the overexpression and knockdown plasmids to prove that the systems work at the protein level. The overexpression and knockdown of *DDIT3* can be clearly seen in **Fig. 1A**.

2. Immunofluorescence in primary CD34+ cells

Secondly, to prove that the systems also work in primary cells, we have FACS-sorted healthy CD34+ cells transduced with the overexpression system, and MDS CD34+ cells infected with the knockdown system to demonstrate, by immunofluorescence, the change in DDIT3 protein levels. As it can be observed, levels of *DDIT3* overexpression in primary cells are modest but they can be clearly seen by this technique (**Fig. 1B**), which is in accordance with the 3-4 fold-change upregulation detected by RNA-seq in this system (**Fig. 1C**), and with the endogenous upregulation observed in MDS patients (**Fig. 1D** or **Fig. 2F of the revised manuscript**). Thus, these results demonstrate that the overexpression system in primary cells recapitulates the upregulation detected in patients. The results of immunofluorescence of DDIT3 after transduction with the overexpression system have been included in **figure S3C** and in the text of the revised version of the manuscript:

Line 202: “*DDIT3* was overexpressed in CD34+ cells from healthy donors (YH_18, YH_19) (Fig. S3C)”

Regarding *DDIT3* knockdown, we detected a decrease of the protein upon transduction with the targeting shRNAs (**Fig. 1E**), an effect also detected at mRNA level (**Fig. 1F**), which was sufficient to enhance erythroid differentiation in MDS patients.

Figure 1. Validation of overexpression and knockdown systems at protein level. **(A)** K562 cells were transduced with the overexpression (left) or knockdown (right) systems, and extracts were obtained at 8 (overexpression) or 2 (knockdown) days and immunoblotted with the indicated antibodies. **(B)** Healthy CD34⁺ cells were transduced with the overexpression system, and after 3 days, infected cells (GFP⁺) were FACS-isolated and immunofluorescence was performed with an anti-DDIT3 antibody. **(C)** Healthy CD34⁺ cells were transduced and sorted as above and RNA-seq was performed 4 days after transduction. **(D)** Normalized expression of *DDIT3* in HSCs from healthy young (blue), healthy elderly (orange) and MDS (red) samples, analyzed by RNA-seq. Each point represents a donor or patient and the mean \pm standard deviation (SD) is shown. **(E)** MDS CD34⁺ cells were transduced with the knockdown system, and after 3 days, infected cells (GFP⁺) were FACS-isolated and immunofluorescence was performed with an anti-DDIT3 antibody. **(F)** MDS CD34⁺ cells were transduced and sorted as above and RNA-seq was performed 13 days after transduction with the knockdown system.

2. The authors evaluate differentiation of healthy and MDS CD34+ cells with respective enforced and knockdown DDIT3 expression using various in vitro differentiation systems, namely liquid cultures for erythroid-specific or myeloid-specific differentiation (healthy cells / overexpression), OP-9 co-culture in the presence of multilineage-supporting cytokines (DDIT3 overexpression for scRNA-seq analysis), and likely (this is not explicitly stated) the same OP-9 co-culture system for DDIT3 knockdown in MDS samples. Additionally, they use CFC assays for analysis of DDIT3-overexpressing CD34+ cells only, and phenotypic analysis relies almost exclusively on flow cytometry analysis of erythroid progression.

First of all, we apologize for not indicating properly which system was used for all the experiments performed in this work. In the case of *DDIT3* knockdown experiments, we have clarified in the revised version of the manuscript that the OP-9 co-culture differentiation system was used to assess the effect of *DDIT3* knockdown in MDS samples:

Line 316-317: “*DDIT3* was knocked down in CD34+ cells from patients with MDS using shRNAs, and cells were induced to differentiate using the OP-9 differentiation system (Fig. 6A).”

To address this question we have performed additional experiments using the 2 different culture systems to prove the consistency of the results between the systems. The details are provided below for each specific question.

2.1 The authors should include more detailed data to validate and compare the various differentiation systems employed using healthy untransduced cells – including cell expansion, cell death and progression of erythroid / myeloid differentiation and loss/preservation of HSPC at different timepoints.

As suggested by the reviewer, we have compared both differentiation systems utilized in this study using primary untransduced healthy CD34+ cells (**Fig. 2**).

Regarding **differentiation**:

- In the liquid erythroid-specific differentiation system cells reach stages II (CD71+ CD235a-) and III (CD71+ CD235a+) of erythropoiesis by day 3. By day 7, there are barely any cells in stages I and II, with most cells (85%) reaching stage III (CD71+ CD235a+), and some of them (13%) stage IV (CD71- CD235a-). Days 10 and 14 of differentiation show a progressive decrease in the percentage of cells in stage III and a concomitant increase of cells in stage IV, that represent about 73% of the population at day 14 (**Fig. 2A**).
- In the myeloid-erythroid differentiation system over OP-9 cells, erythroid differentiation progression looks very similar, although at day 7 less cells (57%) have reached stage III and there are still some cells in stages I and II. The percentage of cells reaching stage IV at day 14 of differentiation is similar to the other system (77%) (**Fig. 2B**). In the myeloid-erythroid differentiation system over OP-9 cells, some cells also undergo myeloid differentiation (measured as CD13+ cells), although they represent a minor percentage (around 15%) compared to those undergoing erythropoiesis, specially from day 7 to day 14 (**Fig. 2C**).

Both systems show a progressive loss of HSPCs (measured as percentage of CD34+ cells), with 45- 55% at day 3, 10-13% at day 7, and almost no HSPCs at days 10 and 14, when most cells have reached more mature stages (**Fig. 2D, 2E**).

Regarding **cell death**: the liquid erythroid-specific differentiation system shows a rate of apoptotic cells of 10-20% (**Fig. 2F**). On the other hand, the myeloid-erythroid differentiation system over OP-9 cells shows increased cell survival, with 1.7-5% of the cells being annexin V+ (**Fig. 2G**)

Finally, the **proliferation** of the cells in both systems is different. Figures **2H** and **2I** represent the cumulative cell counts obtained from 16,000 cells at different time points over a 14-day differentiation period. As it can be observed, the proliferation in the liquid erythroid-specific differentiation system

rendered about 60,000 cells at day 14, while the myeloid-erythroid differentiation system over OP-9 cells presented an increased rate of proliferation, resulting in 1.7×10^6 cells over the same period of time.

Altogether, the comparison of the two systems indicates that they are very similar systems to model erythroid differentiation, although the myeloid-erythroid differentiation system over OP-9 cells presents less cell death and increased proliferation. In our hands, this system has rendered better results when starting from samples with limiting cell numbers such as those obtained from MDS specimens or healthy elderly individuals.

These results have not been included in the manuscript due to space constraints.

Figure 2. Comparison of differentiation systems: liquid erythroid-specific differentiation system (left) and myeloid-erythroid differentiation system over OP-9 cells (right). **(A-B)** Bar-plots representing the percentage of cells observed in stages I-IV of erythropoiesis at different times of differentiation for the two systems. **(C)** Percentage of CD13+ cells in the myeloid-erythroid differentiation system over OP-9 cells at the indicated time points. **(D, E)** Percentage of HSPCs measured as CD34+ cells at the indicated time points for the two systems. **(F, G)** Percentage of annexin V+ cells at the indicated time points for the two systems. **(H, I)** Cumulative number of cells at each indicated time point. For all panels (A-I), the average of three independent experiments +/- SD is shown.

2.2 Given the centrality of the opposing effects of *DDIT3* overexpression and knockdown to the conclusions, it would be important to demonstrate that opposing effects are achieved in the same culture systems – can MDS samples be assessed and how do they behave in erythroid liquid culture? is the erythroid delay observed in multilineage OP-9 cultures of CD34+ cells with *DDIT3* overexpression? How do MDS CD34+ behave in CFC assays upon *DDIT3* knockdown? And have the authors attempted to perform CFC re-plating of healthy or MDS cells with or without *DDIT3* as a means to assess *DDIT3* contribution to *in vitro* transformation? Given the data in mouse GMP, this could also be a contributing factor to MDS pathogenesis.

We thank the reviewer for the comment and agree that a demonstration of the opposing effects of *DDIT3* overexpression and knockdown in healthy and MDS samples using the same system is important. To demonstrate such opposing effects, we have overexpressed *DDIT3* in healthy CD34+ cells from three different healthy donors and have submitted the cells to the multilineage OP-9 differentiation system, which, as indicated above, was the model used to analyze the effect of the knockdown in MDS specimens. Similarly to the liquid erythroid differentiation system (**Fig. 4B, 4C of the revised manuscript**), we have observed how *DDIT3* overexpression promotes a delay in erythroid differentiation, with about 22% decrease of cells in stage IV, and an accumulation of cells at more immature stages at day 14 of differentiation (**Fig. 3A**), demonstrating that the delayed erythroid differentiation is not assay-dependent. Furthermore, the effect of *DDIT3* overexpression in the erythroid differentiation of healthy CD34+ cells using this system can also be seen in more detail in figures **4D-F of the revised manuscript**, as this was the system used for the scRNA-seq experiment performed upon overexpression of the factor. Altogether, this data demonstrate that, using the same system, *DDIT3* overexpression and knockdown in healthy and MDS CD34+ cells, respectively, have opposing effects.

Regarding CFC formation studies in the knockdown system, we believe this may not be the best system to analyze differentiation of MDS cases, as it has been previously described, and also observed in the experiments performed for this revision (see **Fig. 12, reviewer 2**), that cells from these patients barely form any erythroid or myeloid CFCs. Thus, the sensitivity to identify abnormalities during differentiation may be lower than that of liquid differentiation systems. Nevertheless, we attempted to perform these assays upon *DDIT3* knockdown in CD34+ cells from 3 different MDS patients. Patient MDS17 barely formed any BFU-E CFCs, and no significant changes in the number of colonies obtained were detected upon knockdown of the factor. Patients MDS18 and MDS19 did not form any BFU-E or CFU-G/M/GM colonies at baseline or upon knockdown of *DDIT3* (**Fig. 3B**). These results suggested that this system may not be sensitive enough to study alterations in the differentiation abilities of MDS cells.

Finally, as suggested, we have performed re-plating experiments upon *DDIT3* overexpression in healthy CD34+ cells in order to assess *DDIT3* contribution to *in vitro* transformation. Replating of 500,000 cells from primary colony formation experiments in order to form secondary colonies rendered very few CFCs. In particular, only M and GM colonies were observed, and no significant differences were detected between control and *DDIT3*-overexpressing cells (**Fig. 3C**), suggesting *DDIT3* overexpression on its own may not be sufficient to promote *in vitro* transformation.

Collectively, these results demonstrate that, independently of the liquid differentiation system used, *DDIT3* overexpression in healthy CD34+ cells renders opposing effects than its knockdown in MDS patients, promoting a delay in erythroid differentiation. Nevertheless, its overexpression in healthy cells may not be sufficient to promote *in vitro* transformation.

Figure 3. (A) Left: representative flow cytometry charts representing advanced erythroid differentiation (CD71 and CD235a markers; stages I-IV) of CD34+ cells for control cells and cell overexpressing *DDIT3* at day 14 of differentiation, using the myeloid-erythroid system over OP-9 cells. Right: bar-plots representing the percentage of cells observed in stages I-IV in three biological replicates (average +/- SD is depicted). Statistical significant differences are indicated (*:p<0.05; **: p<0.01). (B) Graphs indicating the number of colonies (BFU-E and G/M/GM) obtained for CD34+ cells obtained from 3 different patients and transduced with a control or a *DDIT3*-targeting shRNA. Average +/- SD of technical replicates is shown. (C) Graph indicating the number of colonies observed for cells obtained in a secondary colony formation experiment for control and *DDIT3*-overexpressing cells. The total number of colonies detected per 500,000 cells seeded from a primary colony formation experiment is depicted. Average +/- SD of three independent experiments is shown.

2.3 The authors state the importance of analysing MDS signatures in purified HSC, yet perform functional analyses in total CD34+ cells (see also MINOR POINT 1). While I understand that LV transduction of purified HSC produces limiting numbers of cells for the various analyses – namely transcriptomic – performed, possible effects of cell selection, death or biased expansion are not explored. Given the specific association of *DDIT3* with a relative expansion or differentiation delay of early erythroblasts, but without evidence of an absolute defect in terminal differentiation, the authors should include timecourse assessments of differentiation status, cell cycle and apoptosis, as well as more detailed analysis of single-cell initiated cultures. This level of detail will help eliminate potential confounder effects of cell selection,

and may reconcile different and some contradictory effects on erythroid programmes observed in this and other studies.

We thank the reviewer for this relevant comment. To address it, we have taken different approaches. On the one hand, as suggested, we have included assessments of differentiation, proliferation, cell cycle, and apoptosis in control and *DDIT3*-overexpressing CD34+ cells. On the other hand, we have performed differentiation studies with HSCs enriched subsets upon *DDIT3* overexpression.

1. Overexpression experiments in CD34+ cells

Regarding erythroid differentiation, at day 3 a delay in erythroid differentiation can already be observed, with less cells in stage II and an accumulation of cells in stage I upon *DDIT3* overexpression. At day 7, while almost 50% of control cells have reached stage III, *DDIT3*-overexpressing cells show an accumulation at earlier stages of differentiation, with only 20% of such cells reaching stage III. Finally, at days 10 and 14, there is a significant decrease of *DDIT3*-overexpressing cells in stage IV, with an accumulation of erythroid progenitors at stages II and III (**Fig. 4A, 4B, and Fig. 4B-C of the revised manuscript**).

When proliferation was assessed, we did not detect any statistically significant differences between control and *DDIT3*-overexpressing cells (**Fig. 4C**), which was also reflected by no changes in cell cycle status (**Fig. 4D**). Moreover, apoptosis measured by annexin V analyses did not show any significant alterations in the percentage of apoptotic cells upon *DDIT3* overexpression (**Fig. 4E**). Altogether, these results suggest that *DDIT3* overexpression does not affect the proliferation or apoptosis of healthy CD34+ cells undergoing erythroid differentiation. Thus, abnormal erythropoiesis observed upon *DDIT3* overexpression does not seem to be due to an expansion of erythroid progenitors but to a delayed differentiation of progenitor cells.

The delayed differentiation observed in these experiments is supported by the velocity analysis performed in the scRNA-seq data obtained upon 14 days of differentiation. This analysis showed delayed differentiation of *DDIT3*-overexpressing cells from late basophilic to reticulocyte states (**Fig. 4F-H of the revised manuscript**). Furthermore, the gene regulatory network analyses performed in these data also indicated that at specific differentiation states, *DDIT3* overexpression leads to decreased activation of key erythroid differentiation transcriptional programs, supporting the idea of delayed erythroid differentiation.

Regarding single-cell initiated cultures, we observed a significant decrease in CFU-M as well as in BFU-E colonies upon *DDIT3* overexpression (**Fig. 4F**). Interestingly, erythroid colonies did not only present a lower total number upon overexpression of the factor, but also showed a modified morphology, being smaller and less compact than those formed by cells transduced with the control plasmid (**Fig. 4G**). It is worth mentioning that these results support our hypothesis of the role of *DDIT3* in the erythroid maturation at least in healthy cells. We acknowledge that in MDS patients, the limited growth of erythroid colonies has precluded us from using the system to address the impact of *DDIT3* knockdown on the formation of BFU-E.

2. Overexpression experiments in HSCs

Besides the above detailed experiments, we also have performed overexpression experiments starting from primary healthy HSCs to analyze the effect on erythropoiesis of *DDIT3* upregulation. As it can be seen in **figure 5**, *DDIT3* overexpression produces a delay in erythroid differentiation that is very similar to that observed for CD34+ cells, with a 30% decrease in the percentage of cells in stage III, and an accumulation of cells at stage II at 14 days of differentiation. Furthermore, at a later time point of 17 days, 40-45% of control cells have progressed to stage IV whereas only 12% of *DDIT3*-overexpressing cells reach such state.

In our perspective, collectively these results would indicate that it is unlikely that the observed results are due to a bias related to the use of CD34+ cells instead of an enriched population of HSCs.

Several of the results explained in this point have been incorporated in the revised version of the manuscript:

- The proliferation, cell cycle, and apoptosis results have been incorporated in figure S4 and in the revised text:
Lines 228-232: “Moreover, DDIT3-overexpression did not produce alterations in the proliferation, cell cycle status, or in the percentage of apoptotic cells (donors YH 20, YH 21, YH 23) (Fig. S4B-D), suggesting that the abnormal erythropoiesis observed upon DDIT3 overexpression is not due to an expansion of erythroid progenitors, but to a delayed differentiation of progenitor cells.”
- The CFC results have been included in figure 4A and in the revised text:
Lines 221-223: “We observed a statistically significant decrease in the number of both erythroid burst-forming units (BFU-E) and monocytic colony forming units after DDIT3 overexpression (Fig. 4A).”
- Finally, we believe that the results obtained upon overexpression of *DDIT3* in HSCs are relevant for the validation of our hypothesis and thus, they have been incorporated in figure S4E and in the revised manuscript:
Lines 233-238: “As previous experiments were performed in CD34+ cells, and in order to prove that the observed effect on erythropoiesis could be driven by the DDIT3 upregulation observed in HSCs from MDS patients, we also induced DDIT3-overexpression in primary healthy HSCs. Overexpression of this factor also produced a delay in erythroid differentiation that was very similar to that observed for CD34+ cells, with a 30% decrease in the percentage of cells in stage III, and an accumulation of cells at stage II at 14 days of differentiation (Fig. S4E).”

Figure 4. Phenotypic effects of *DDIT3* overexpression. **(A)** Flow cytometry charts representing advanced erythroid differentiation (CD71 and CD235a markers; stages I-IV) for control cells and cell overexpressing *DDIT3*, at the indicated time points. **(B)** Bar-plots representing the percentage of cells observed in A in stages I-IV. Statistically significant differences are indicated (*p-value<0.05; **p-value<0.01; ***p-value<0.001). **(C)** Cumulative number of cells at 0 and 14 days for control and *DDIT3*-overexpressing cells. **(D)** Percentage of cells in different phases of the cell cycle, detected by propidium iodide for control and *DDIT3*-overexpressing cells. **(E)** Percentage of cells stained with Annexin V and/or DAPI for control and *DDIT3*-overexpressing cells. **(F)** Graph indicating the number of colonies obtained for healthy CD34+ cells with a control or a *DDIT3*-overexpressing plasmid. For panels B-F the average of three biological replicates +/- SD is depicted. **(G)** Micrographs of representative erythroid colonies formed by cells detailed in F.

Figure 5. Effect on erythroid differentiation of *DDIT3* overexpression in primary healthy HSCs. Left: flow cytometry charts representing advanced erythroid differentiation (CD71 and CD235a markers; stages I-IV) of HSCs transduced with a control or a *DDIT3*-overexpressing plasmid at days 14 and 17 of differentiation. Right: bar-plots representing the percentage of cells observed for stages I-IV. Average \pm SD is depicted, and statistically significant differences are indicated ($*p < 0.05$).

3. The authors report *DDIT3* overexpression in only a subset of MDS samples. Can the authors please include more detail on the levels of baseline expression and knockdown achieved in the individual samples #13-17 reported in this study? Were there other samples with similar *DDIT3* expression that responded differently / not responded to *DDIT3* knockdown? Have the authors attempted to overexpress *DDIT3* in low-expressing samples?

We thank the reviewer for this thoughtful question. The baseline expression of *DDIT3* in HSCs of the patients included in the knockdown experiments was variable: while specimen MDS15 showed modest *DDIT3* overexpression, the rest of the samples presented levels similar to those detected in elderly healthy individuals (**Fig. 6A**). Interestingly, all the patients tested responded to *DDIT3* knockdown regardless of the baseline expression of this factor, suggesting that *DDIT3* may acquire a key role in the impairment of erythroid differentiation in MDS patients. Moreover, these results suggest that its inhibition may be beneficial even for those patients not showing high levels of this factor.

These results have been included in Fig. S6G of the revised manuscript and in the text:

Lines 330-332: “Notably, although the patients tested presented variable *DDIT3* basal levels of expression in their HSCs (Fig. S6C), all of them showed similar effects upon knockdown of the transcription factor.”

Moreover, we have commented these observations in the discussion section:

Lines 422-426: “Intriguingly, our experiments demonstrate that *DDIT3* knockdown may not only be beneficial for patients showing upregulation of this factor, as cases presenting levels similar to those observed in healthy elderly donors also demonstrated improved erythroid differentiation upon its knockdown. These results suggest that, independently of its levels of expression, *DDIT3* may acquire a key role in the control of erythropoiesis in MDS patients.”

Regarding the level of knockdown achieved, we analyzed *DDIT3* expression in transduced and GFP-sorted MDS samples for which we had sufficient material. As it can be seen in **Fig. 6**, the knockdown was modest at day 7 for patients MDS13 (10% and 43% of knockdown) (**Fig. 6B**), and MDS14 (42% and 30% of knockdown) (**Fig. 6C**), and was more evident at day 13 (65% and 76% of knockdown) (**Fig. 6B**). The levels

of knockdown achieved have been included in figures S6A and S6C of the revised manuscript and in the text:

Line 320-321: “knockdown in patient 13 (RNA levels after knockdown shown in Fig. 6SA),”.

Line 325-326: “patient 14 (male, Hb 7.9 g/dL) (RNA levels after knockdown shown in Fig. 6SC);”.

As suggested by the reviewer, we have also evaluated the effect on erythroid differentiation of overexpressing *DDIT3* in low-expressing samples. Specifically, we selected patients MDS16 and MDS17, for which we had already performed knockdown experiments, so we could compare overexpression and knockdown of *DDIT3* within the same MDS specimens. Interestingly, overexpression of this factor promoted a slight delay in erythroid differentiation, with a decrease of cells in stages III and IV, and a concomitant increase in stage I for MDS17, and stages I and II for specimen MDS16 (**Fig. 6D**). Interestingly, knockdown of *DDIT3* in the same MDS samples enhanced the transition to stage IV (**Fig. 6E, 6F and Fig. S6E, S6F of the revised manuscript**), indicating that inhibition of this factor boosts terminal erythropoiesis. Thus, these results suggest that *DDIT3* overexpression in low-expressing samples promotes an effect opposite to that observed upon its knockdown, promoting a delay in erythroid differentiation. In our perspective, these results reinforced the effect of *DDIT3* on regulating erythroid differentiation.

Finally, we also performed colony formation assays in patients MDS6 and MDS17 upon *DDIT3* overexpression and observed no statistically significant differences in the numbers of CFU-G/M/GM colonies formed (**Fig. 6G**). Nevertheless, as indicated above, these patients barely formed any erythroid colonies, what precluded us from using the system to address the impact of *DDIT3* on the formation of BFU-E.

Figure 6. (A) Normalized expression of *DDIT3* in healthy young (blue), healthy elderly (orange) and MDS (red) samples used for the knockdown experiments. Each point represents a donor or patient and the mean \pm standard deviation (SD) is shown for each group. Each MDS patient is indicated with its sample ID (Table S2 of the manuscript). (B-C) Normalized expression of *DDIT3* in patients MDS13 (B) and MDS14 (C) upon transduction of CD34+ cells with the indicated shRNAs, culture in the myeloid-erythroid differentiation system for 7 (MDS13 and MDS14) and 14 (MDS14) days, and FACS-isolation of transduced cells. (D) Left: flow cytometry charts representing advanced erythroid differentiation (CD71 and CD235a markers; stages I-IV) of CD34+ cells from two patients (MDS16 and MDS17) showing low basal levels of *DDIT3*, transduced with a control or *DDIT3*-overexpression plasmid and differentiated for 14 days. Right: bar-plots representing the percentage of cells observed on the left. (E-F) Left: flow cytometry charts representing advanced erythroid differentiation (CD71 and CD235a markers; stages I-IV) for cells from patients MDS16 (E) and MDS17 (F) harboring a control shRNA (shCtrl) or shRNAs targeting *DDIT3*, after *ex vivo* myeloid differentiation in liquid culture. Right: bar-plots representing the percentage of cells observed in stages I-IV. (G) Graphs indicating the number of colonies (BFU-E and G/M/GM) obtained for CD34+ cells isolated from 2 different patients (MDS16 and MDS17) with low basal expression of *DDIT3*, and transduced with a control or a *DDIT3*-overexpression plasmid.

MINOR POINTS

1. MARS-seq analysis of MDS vs Young / Old healthy HSC shows that *DDIT3* to be mildly albeit significantly over-expressed in MDS samples. While the mild overexpression may be due to sample heterogeneity with only a subset of MDS HSC showing detectable expression of *DDIT3*, other publicly available datasets do not seem to capture *DDIT3* overexpression. The authors suggest that this may be due to a specific effect in highly purified HSC, which is not captured by analysis of bulk CD34+, but (i) provide no additional characterisation of the MDS samples in which *DDIT3* was overexpressed, and (ii) perform overexpression / knockdown experiments in total CD34+ cells which may be confounded by cell heterogeneity. On this point, it would be helpful if the authors could give some insight into the perceived specificity of *DDIT3* pathogenic effects, for example in terms of characteristic genetic alterations or transcriptional contexts in which *DDIT3* may be relevant to MDS.

We thank the reviewer for this relevant comment. As the sample set analyzed in this work was limited in number, associations between *DDIT3* upregulation and specific characteristics of the patients cannot be clearly established. Taking this into consideration, we observed that MDS samples showing *DDIT3* upregulation showed a variety of genetic lesions: two of the patients had no detectable myeloid mutations, while others had variable genetic lesions: *GATA2*, *JAK2*, *BCOR* (MDS3), *ASXL1* and *ZRSR2* (MDS4), *TET2*, *SRSF2*, and *ZRSR2* (MDS9), and *DNMT3A*, *PPM1D*, *SF3B1*, and *NF1* (MDS12). Thus, the only genetic commonality was *ZRSR2* mutation, which showed its higher VAF (78%) in patient MDS4, which was the one showing higher levels of *DDIT3* mRNA. Future studies using larger sets of data are needed to establish associations between *DDIT3* upregulation and genetic alterations. The information regarding the genetic context of the patients can be found in Table S2 of the revised manuscript.

As the reviewer indicated, the overexpression/knockdown experiments shown (with the exception of the transcriptional studies carried out upon *DDIT3* overexpression shown in figure 3 of the manuscript), were performed in total CD34+ cells due to cell number limitations of isolated HSCs. Nevertheless, we agree that in such experiments, effects could be confounded by cell heterogeneity. As indicated above (Fig. 5), and in order to prove that upregulation of this factor in HSCs may be responsible for aberrant erythroid differentiation, we have performed overexpression experiments starting from healthy primary HSCs. As explained above, *DDIT3* overexpression produced a delay in erythroid differentiation that was very similar to that observed for CD34+ cells, with a 30% decrease in the percentage of cells in stage III, and an accumulation of cells at stage II at 14 days of differentiation. Furthermore, at a later time point of 17 days, 40-45% of control cells had progressed to stage IV whereas only 12% of *DDIT3*-overexpressing cells reached such state. Thus, these results demonstrate that *DDIT3* overexpression specifically in HSCs is able to promote aberrant erythroid differentiation.

2. The CFC results presented in this study are of technical replication for individual samples, with 3 different experiments analysed separately and shown in the main and supplementary figures. The authors should instead analyse the statistical significance of biological replicates for robustness of their conclusions.

Following the comment from the reviewer, we have now plotted in figure 4A of the manuscript the average +/- SD of three independent experiments. Moreover, we have provided more detail of the myeloid colonies observed in these assays. These changes have been included in Fig. 4A of the revised version of the manuscript and in the text:

Lines 221-223: “We observed a statistically significant decrease in the number of both erythroid burst-forming units (BFU-E) and monocytic colony forming units after *DDIT3* overexpression (Fig. 4A).”

3. *DDIT3* has also been implicated in HSC maintenance through regulation of ER stress. Have the authors investigated ER stress signatures in the context of *DDIT3* overexpression / knockdown and their contribution to MDS pathogenesis?

We believe this is a very interesting comment, due to the involvement of *DDIT3* in ER stress and its association with HSC maintenance. As suggested, we have investigated ER stress signatures by performing GO and GSEA analyses in healthy CD34+ with overexpression of *DDIT3* and in MDS CD34+ cells after its knockdown. Although we have not detected statistically significant enrichment in these signatures in neither of the systems, we have observed a tendency in the enrichment of *DDIT3*-overexpressing cells after 10 days of erythroid differentiation in signatures such as:

- GO positive regulation of transcription from rna polymerase ii promoter in response to endoplasmic reticulum stress (NES=1.28, FDR=0.54).
- GO regulation of endoplasmic reticulum stress induced intrinsic apoptotic signaling pathway (NES=1.26, FDR=0.52).

These signatures only presented core enrichment in 6 and 4 genes, respectively. These results suggest that *DDIT3* overexpression in healthy CD34+ cells does not produce a significant effect on ER stress pathways, at least in our models, although it may slightly affect the expression of few genes involved.

4. In Supplementary Table S1 YH should be Young_Healthy, not Young_Elderly.

Thank you for noticing this typo, we have changed Young_Elderly for Young_Healthy in Table S1.

Reviewer #2 (Remarks to the Author): with expertise in aging, HSC

Review of NCOMMS-21-47839

In this study the group of Felipe Prosper and Teresa Ezponda aims at elucidating the transcriptional changes occurring in human HSCs during aging and in MDS, a hematopoietic malignancy mainly affecting elderly individuals. The biological processes extrapolated by their transcriptional analysis indicate that the transcriptional profile of MDS samples differs from the one reported in aged HSC. The authors next identify a Transcription Factor, DDIT3, specifically up-regulated in MDS samples and show that when overexpressed in healthy HSC DDIT3 leads to ineffective erythropoiesis, a condition also observed in MDS patients. The study is of translational relevance and the transcriptomic data are presented in a clear manner with conclusions drawn from experiments that are largely coherent with data. However, there are some open points, mainly regarding the mechanistic part of the study and outlined below, that preclude publication in its current form.

Major revisions

DDIT3 overexpression in healthy HSPCs and its downregulation in MDS give interesting and consistent results. However, the mechanisms behind the rescue of phenotype remains elusive. It would be intriguing to analyze DNA damage markers (53BP1/γH2AX/pRPA) in a time-course from day 1 to day 8 or beyond upon DDIT3 overexpression in healthy donors or following its downregulation in MDS as DNA replication stress and accumulation of DNA damage may play a key role in MDS pathogenesis as well as during aging. I would expect an upregulation of replication stress and accumulation of double strand breaks as genes involved in the regulation of DSB repair are downregulated upon DDIT3 overexpression. Interestingly, the authors should describe and comment the impact of DDIT3 on inflammation and upregulation of inflammatory gene categories, given the prominent role of inflammatory pathways in aging and age-associated diseases, including MDS.

We thank the reviewer for this interesting comment. In order to assess the question we have taken two different approaches: on the one hand, we have performed *in vitro* experiments to study DNA damage upon *DDIT3* manipulation; and on the other hand, we have dwelled into DNA damage and inflammation gene signatures upon *DDIT3* overexpression or knockdown in primary healthy and MDS cells.

1. DNA damage

As suggested, we have performed experiments to explore DNA damage at different time points after *DDIT3* overexpression in healthy donors, and after its knockdown in MDS samples. As shown in figure 7, manipulation of *DDIT3* levels did not alter the percentage of cells showing DNA damage, measured as p-γH2AX, at any of the time points tested (**Fig. 7A, B**).

These results were intriguing as, as the reviewer indicated, we observed decreased expression of DSB repair genes after *DDIT3* overexpression (with no differentiation stimuli) (**Fig. 3C of the revised manuscript**). To investigate this in more depth, we evaluated DNA repair signatures at different time points of differentiation (days 4, 10 and 14 of erythroid differentiation). While no enrichment in gene signatures related to DNA damage were observed at 4 and 10 days, an enrichment in signatures related to DNA repair was observed at day 14. In particular, we observed how genes upregulated upon overexpression were enriched in several signatures related to response to DNA damage, and DNA repair, including double-strand break repair (**Fig. 7C**), whereas genes downregulated were enriched in different signatures related to DNA damage response and repair (**Fig. 7D**).

Altogether, these data suggest that at baseline, with no differentiation stimuli, overexpression of *DDIT3* seems to repress DSB repair signatures, but upon induction of erythroid differentiation, this effect is lost. Although at day 14 of differentiation the overexpression of this factor alters expression of genes regulating DNA damage and repair, such alteration is observed in both directions, as it promotes the overexpression

and repression of specific genes. Nevertheless, these transcriptional alterations are not manifested as increased or decreased DSB upon manipulation of *DDIT3* levels. Therefore, although the effect of *DDIT3* on erythroid differentiation is very clear, with the observed data we cannot conclude that regulation of DSBs by *DDIT3* mediates such effect.

Figure 7: Effect of *DDIT3* on DNA damage. **(A-B)** percentage of p-γH2AX+ cells upon *DDIT3* overexpression (A) or knockdown (B) in healthy or MDS CD34+ cells, respectively, at the indicated time points. The average +/- SD of two independent experiments is represented. ns: no statistically significant differences. **(C-D)** Signatures related to DNA damage and repair obtained from the GO analysis of genes up- (C) or down- (D) regulated upon *DDIT3* overexpression after 14 days of liquid erythroid differentiation. p-value (-Log2) for each signature is depicted.

2. Impact of *DDIT3* on inflammation response

To study the impact of *DDIT3* overexpression on inflammation response, we analyzed the GSEA performed upon overexpression of the factor at baseline (without differentiation stimuli; **figure 3 of the revised manuscript**), and upon differentiation towards the erythroid lineage (**figure 5 of the revised manuscript**).

- *At baseline (no differentiation stimuli)*, *DDIT3* overexpression was performed on HSCs. Such overexpression seemed to promote a less inflammatory transcriptional profile, with signatures such as positive regulation of inflammatory response, response to bacterium, NFKB1 targets, inflammatory response upon LPS treatment being less enriched upon overexpression of the factor, and the signature negative regulation of inflammatory response being more enriched (**Fig. 8A**).

- For liquid erythroid differentiation experiments, *DDIT3* overexpression was carried out on CD34+ cells. At day 10 of differentiation, it promoted a more inflammatory transcriptional profile, with several signatures related to inflammatory response being significantly enriched (**Fig. 8B**).

Figure 8. GSEA of *DDIT3*-overexpressing and control cells, at baseline (with no differentiation stimuli) (**A**), or upon differentiation for 10 days (**B**). The NES for several signatures related inflammation/immune response is depicted.

Although the analysis at baseline and upon differentiation was performed with different populations (HSC enriched subpopulations and CD34+ cells), we believe that the differences in the inflammatory response might be related to the culture of the cells in very different conditions (the differentiation culture contains cytokines and growth factors more akin with the physiological situation, thus probably with a better resemblance of the situation *in vivo*). These results would be consistent with studies describing that CD34+ cells from MDS cases show augmented inflammation/stress response.

The authors discuss that upregulated levels of *DDIT3* in MDS could sequester CEBPB and CEBPG, resulting in a diminished expression of their target genes, including TFs with key roles in erythropoiesis. In this context, further experiments would be necessary to corroborate the mechanistic part of the manuscript. It would be interesting to dissect association of CEBPB and CEBPG in promoters of TFs involved in erythropoiesis by ChIP or to verify a direct association of *DDIT3* and CEBPB and CEBPG by Immunoprecipitation upon *DDIT3* overexpression and downregulation.

We thank the reviewer for the question and agree that corroborating the mechanism of action of *DDIT3*-overexpression would strengthen our conclusions. In order to verify the direct association of *DDIT3* with CEBPB and CEBPG we performed immunoprecipitation experiments upon *DDIT3* overexpression in the K562 cell line. When immunoprecipitation was performed with an anti-CEBPB antibody, an association of this factor with *DDIT3* was detected in cells transduced with the control plasmid. Moreover, upon overexpression of *DDIT3*, more *DDIT3* was co-immunoprecipitated by CEBPB, demonstrating that when this factor is overexpressed, it is able to sequester more CEBPB, potentially impeding the normal activity

of this transcription factor (**Fig. 9A**). Likewise, immunoprecipitation with an anti-DDIT3 antibody co-immunoprecipitated CEBPB, a binding that was enhanced upon *DDIT3* overexpression (**Fig. 9B**). Similarly to CEBPB, immunoprecipitation of CEBPG also demonstrated a binding to DDIT3. In this case, this binding was exclusively detected upon *DDIT3* overexpression (**Fig. 9C**).

These results demonstrate the binding of DDIT3 with CEBPG and CEBPB, and how *DDIT3* overexpression leads to enhanced binding and sequestering of these factors. Supporting these data, the analysis of regulon activity performed (**figure 5E of the revised manuscript**) demonstrated how the regulons guided by CEBPB and CEBPG show decreased activity upon *DDIT3* overexpression (**Fig. 9D**).

Collectively, these results demonstrate that *DDIT3* overexpression leads to increased binding and sequestering of CEBPB and CEBPG, which in turn, results in decreased activity of these factors and diminished expression of their target genes, including erythroid differentiation genes.

The results have been incorporated in Figure 5I and S5H-S5J of the revised manuscript and in the text:

Lines 305-312: *“Immunoprecipitation experiments confirmed DDIT3 binding to CEBPB and CEBPG, and how such binding was enhanced after the overexpression of DDIT3 (Fig. 5I, S5H-I). Accordingly, regulons guided by CEBPG and CEBPB showed decreased activity upon DDIT3 overexpression (Fig. 5E, S5J). Thus, these results suggested that abnormally high levels of DDIT3 in MDS sequester CEBPB and CEBPG, impeding the physiological transcriptional activity of these factors, and leading to decreased expression of their target genes, including TFs with key roles in erythropoiesis. Therefore, DDIT3-overexpression ultimately hampers the activity of transcriptional programs that are necessary for proper terminal erythroid differentiation.”*

Figure 9. (A) Immunoblot for CEBPB (different isoforms indicated) and DDIT3 after the immunoprecipitation with an anti-CEBPB antibody or an IgG control in cells transduced with a control or a *DDIT3*-overexpressing plasmid. (B) Immunoblot for CEBPB (different isoforms indicated) and DDIT3 after the immunoprecipitation with an anti-DDIT3 antibody or an IgG control in cells transduced with a control or a *DDIT3*-overexpressing plasmid. (C) Immunoblot for CEBPγ and DDIT3 after the immunoprecipitation with an anti-CEBPγ antibody or an IgG control in cells transduced with a control or a *DDIT3*-overexpressing plasmid. (D) Ridge plot showing AUC scores for regulons of *CEBPG* and *CEBPB* in control (blue) and *DDIT3*-overexpressing cells (pink) at different stages of erythroid differentiation.

(lines 132-137): Authors claim that HSC aging and transition from elderly to MDS HSCs is driven by different mechanisms. This conclusion derives only from a part transcriptome data and from a small cohort of MDS patient with a low risk profile. However, later on (lines 165-170) they detect some factors known to be involved in the development of myeloid malignancies both in aged and MSD HSC and they conclude that “aging-derived transcriptional changes predispose HSCs for transformation”, which seems to be in contrast with the sentence above. This should be better explained or better characterized by experiments with samples from younger MDS patient or by separating dataset of MSD patients by age (they put together data of 50 years old patients and 80 years old and I wondered if there is any differential response).

We thank the reviewer for the comment, and apologize for the lack of clarity trying to explain our results. In order to address the issue raised by the reviewer we have undertaken two approaches: firstly, to rephrase and explain our observations, also trying to reflect some caveats, and secondly, to perform an additional analysis associating the transcriptional profile of MDS patients with their age.

Before getting into the details, it is important to stress, as mentioned in the manuscript, that the number of cases analyzed was limited; nevertheless, they represented a relatively homogeneous cohort (low-risk MDS cases categorized as MDS-MLD or MDS-SLD).

We would like to clarify the concepts that we aimed to make, based on our analyses:

- Our results indicate that the transcriptional lesions found between healthy elderly and MDS HSCs are, in most part, different from the transcriptional alterations observed between healthy young and elderly HSCs. Although for obvious reasons the samples utilized in this work are not longitudinal, and thus cautiousness needs to be taken when interpreting the results, these observations suggest that, at the transcriptional level, MDS development is not a simple continuum of aging, but instead, suggest more complex patterns of expression. In fact, we detected different transcriptional dynamics among the three types of HSCs (**Fig. 1F of the revised manuscript**), showing how some transcriptional alterations found in aging persist in MDS, some are exacerbated, and some are reverted when compared to MDS. Moreover, some transcriptional alterations were specifically found in MDS HSCs.
- We observed how transcriptional lesions found between healthy young and elderly HSCs, with or without exacerbation of such lesions in MDS, are enriched in signatures and genes with known roles in the development of several types of cancer including myeloid malignancies. This observation is in line with previous data showing how murine aged long term-HSCs overexpress genes involved in leukemic transformation¹, while in humans, a profound epigenetic reprogramming of enhancers has been suggested to promote an age-driven leukemic transformation-prone state of HSC-enriched populations². Collectively, these data support that aging-dependent alterations in gene expression at the HSC level increase the risk of malignant transformation into myeloid neoplasms. Nevertheless, the role of other lesions, transcriptional, epigenetic and/or genetic, may be needed in order to fully develop MDS.
- Finally, we detected transcriptional alterations that were exclusively in HSCs from MDS patients (C3 and C4). Due to their specificity, such alterations could be involved, along with other insults (genetic, epigenetic), in the development of an MDS phenotype. Among them, we detected the overexpression of *DDIT3*, which we show in following experiments has a role in promoting aberrant erythroid differentiation.

Altogether, our data suggest that transcriptional alterations found in elderly HSCs may predispose these cells for the development of myeloid malignancies. Nevertheless, additional transcriptional lesions and/or the acquisition of genetic insults may be needed in order to fully develop MDS.

The manuscript has been modified in order to clarify the above mentioned concepts:

- We have eliminated the following sentence, as it may lead to confusion: *“All together, these data indicate that the transition from healthy elderly to MDS HSCs is characterized by novel transcriptional alterations that lead to deregulation of specific biological processes. Thus, not only deregulated expression of different genes but also distinct biological processes characterize HSCs in aging and the transition to MDS, suggesting that transcriptional lesions taking place during development of the disease are not a continuous evolution of those found in aging.”*; and have replaced it with:
Line: 132-134: *“Thus, these results suggest that transcriptional lesions taking place during development of the disease are not a simple continuum of those found in aging, but instead, suggest more complex patterns of expression.”*
- We have also modified the following paragraph in order to make our main points clearer:
Lines 150-174: *“Among other relevant findings, some of which are detailed in the supplementary data (Fig. S2), these analyses identified alterations that could be related to the predisposition to the development of hematological malignancies. Interestingly, some of such alterations were*

detected in healthy elderly HSCs, suggesting that transcriptional lesions naturally occurring in aging could predispose HSCs for malignant transformation. For example, processes related to cell proliferation were mainly enriched in C2 but also in C6 (Fig. S2A, S2E, S2I), suggesting that an important loss of proliferation activity of HSCs takes place during aging, and it is further exacerbated in MDS, an event that has been previously associated with accumulation of DNA damage³. Furthermore, DNA sensing and repair processes were also enriched in C2 and C6 (Fig. S2A, S2E, S2I), suggesting a progressive loss of the ability to overcome DNA insults and an increased genomic instability of HSCs during aging and MDS, with a potential direct role in disease development. In order to further explore the potential role of transcriptional lesions occurring with age in the promotion of a myeloid disease/cancer susceptibility, we performed gene set enrichment analyses (GSEA) on clusters altered in aging, with or without exacerbation of expression in MDS (C1, C2, C5 and C6). These analyses demonstrated that genes contained in such clusters were enriched in cancer-related signatures (Fig. 2A); furthermore, among such genes, we identified factors with known roles in the development or maintenance of myeloid malignancies (Fig. 2B). Some examples included the upregulation of TRIB1, a transforming gene for myeloid cells⁴, and the matrix metalloproteinase MMP2, which presents high secretion levels in AML⁵; as well as the downregulation of genes such as adenosine deaminase ADA2, and the mini-chromosome maintenance proteins MCM7⁶ and MCM4, whose repression has been involved in myeloid leukemias (Fig. 2C). These results suggest, in line with a previous work performed in mice, that aging-derived transcriptional changes predispose HSCs for transformation. Such lesions may not be sufficient to promote disease development, and may need additional alterations to lead to an MDS phenotype, which may include enhanced alteration of the lesions found in aging (C5 and C6), genetic insults, or transcriptional alterations that occur exclusively in MDS HSCs.”

Regarding the age of the MDS patients included in this study, we performed two additional analyses: on the one hand, we analyzed the distribution of age in both healthy elderly and MDS groups and observed that both groups showed a similar heterogeneity in age, ranging from 58-81 (healthy elderly), and 45-87 (MDS) years old (**Fig. 10A**), with non significant differences between them. On the other hand, we analyzed the transcriptional profile of the MDS patients by PCA and observed that they did not group by age when any of the eight main principal components were analyzed (**Fig. 10B**), what suggested that the transcriptome of this group of patients does not substantially vary based on the age.

Collectively, these data suggest that the transcriptomic characterization performed in MDS is not affected by the heterogeneity in the age of the patients, and that the group of healthy elderly individuals analyzed represents an appropriate age-matched control group.

Figure 10. (A) Dot plot depicting the age of the individuals included in the healthy elderly and MDS patients. (NS: non-significant). (B) Principal component analysis (PCA) of the transcriptome of HSCs isolated from MDS patients. The two main principal components are shown. Color scale indicates age of the patients.

To further validate the transcriptome data and broaden the main conclusions of the paper the authors should try to overexpress in normal HSCs 1/2 additional transcription factors found upregulated in MDS listed in Figure 2F and see what happens in terms of biological phenotype and response. I would like to see if there is a similar effect in colonies and erythropoiesis upon overexpressing other genes with a similar function such as SERTAD2, RELA, EBF4 and ZNF219 in HSCs derived from healthy donors or if this effect is specific of DDIT3. Conversely, experiments testing the role of genes detected in figure 2C such as MCM4 and MCM7 in aged or MDS samples should be performed to observe if there is a rescue of hematopoietic dysfunctions (including ineffective erythropoiesis) in aged or MDS cells. In the same line, having established a nice differentiation protocol in liquid culture coupled with clonogenic assays (Fig. 3A and Fig. 4), it would be important to see how aged and MDS samples behave in response to these culture-induced activating stimuli.

We thank the reviewer for the comment. To answer this question we have performed differentiation experiments upon manipulation of the expression of additional factors, and have also evaluated how healthy elderly and MDS samples behave in the erythroid differentiation culture, and in clonogenic assays.

1. Role of other factors on erythroid differentiation

As suggested, we have selected two additional transcriptional regulators from C3 (exclusively overexpressed in MDS), *SMARCD3* and *MXD1*, to determine whether they could also have a functional role in the disease. *SMARCD3* is a member of the SWI/SNF family of proteins, and, in our RNA-seq analysis, HSCs from 50-60% of the patients studied showed higher expression than that observed in healthy elderly cells (Fig. 11A), while *MXD1* encodes a member of the MYC/MAX/MAD network of basic helix-loop-helix leucine zipper transcription factors that is also overexpressed in about 50% of the patients (Fig. 11B). To test the putative role of *SMARCD3* and *MXD1* on erythroid differentiation, these factors were overexpressed in healthy CD34+ cells, and liquid erythroid differentiation and colony formation experiments were performed. Overexpression of these factors in healthy primary CD34+ cells did not lead to significant effects on liquid erythroid differentiation (Fig. 11C-D). In the clonogenic assays, *SMARCD3* overexpression produced a decrease in the number of BFU-E colonies formed, whereas *MXD1* overexpression did not affect the colony formation abilities of healthy CD34+ cells (Fig. 11E-F).

We also selected a factor from figure 2C of the manuscript (progressively downregulated in aging and MDS) (**Fig. 11G**), *ADA2*, to determine whether its overexpression could rescue hematopoietic dysfunction in MDS cells. Interestingly, CD34+ cells from MDS patients harboring *ADA2* overexpression showed a decrease in the number of infected (GFP+) cells over time (**Fig. 11H**), what suggested that low levels of this factor are relevant for the survival/proliferation of the cells, and thus, preserving decreased expression of *ADA2* is important for the maintenance of MDS CD34+ cells. Erythroid differentiation in liquid culture could not be evaluated due to the low number of cells transduced with *ADA2* obtained at 7 and 14 days post-transduction. On the other hand, no significant changes in the number of colonies formed were observed upon re-expression of this factor in MDS cells (**Fig. 11I**).

Figure 11. (A-B) Normalized expression of *SMARCD3* (A) and *MXD1* (B) in healthy young (blue), healthy elderly (orange) and MDS (red) samples used for the knockdown experiments. Each point represents a donor or patient and the mean \pm standard deviation (SD) is shown for each group. (C-D) Bar-plots representing the percentage of cells observed in stages I-IV of erythropoiesis at different times of differentiation for control and *SMARCD3*-or *MXD1*-overexpressing healthy CD34+ cells. (E-F) Graph indicating the number of colonies (BFU-E and G/M/GM) obtained for healthy CD34+ cells transduced with a control or a *SMARCD3*-overexpression or a *MXD1*-overexpressing plasmid. (G) Normalized expression of *ADA2* in healthy young (blue), healthy elderly (orange) and MDS (red) samples used for the knockdown experiments. Each point represents a donor or patient and the mean \pm standard deviation (SD) is shown for each group. (H) Graph depicting the percentage of GFP+ cells over time relative to that observed at day 3 after transduction, for control and *ADA2*-overexpressing cells. (I) Graph indicating the number of colonies (BFU-E and G/M/GM) obtained from healthy CD34+ cells transduced with a control or a *ADA2*-overexpression plasmid. For panels C-F and H-I, the average of three independent biological replicates \pm SD is represented. Statistically significant differences are indicated (** $p < 0.01$; ns: non significant).

Altogether, these results demonstrate that, not all the factors showing aberrant dynamics of expression may have the same functional role in MDS cells as indicated by the different effects of alteration of expression of *DDIT3*, *SMARCD3*, and *ADA2*.

2. Behavior of healthy elderly and MDS cells in culture

Finally, as suggested by the reviewer, we have also determined how CD34+ cells from elderly and MDS individuals behave in the *ex vivo* cultures used in this study, in comparison with those from young healthy donors. **Figure 12** shows erythroid differentiation stages at days 7 and 14 of healthy young (3 biological replicates) (**Fig. 12A**), 3 healthy elderly (**Fig. 12B**), and 5 MDS cases (**Fig. 12C**). For healthy elderly and MDS cases, the differentiation of each individual is shown independently in order to showcase the heterogeneity of this type of samples. As it can be seen, healthy elderly CD34+ cells show a differentiation that is very similar to that observed in CD34+ cells isolated from young individuals, although one of the samples analyzed show a slight delay in differentiation, with less cells reaching stage IV of differentiation (CD71- CD235a+) at days 7 and 14. In the case of clonogenic assays, healthy elderly samples showed a statistically significant decrease in the number of both CFU-G/M/GM ($p < 0.05$) and BFU-E ($p < 0.01$) colonies, although the reduction in later colonies was much more pronounced (**Fig. 12D**). These results indicate that CD34+ from healthy elderly individuals show a liquid erythroid differentiation that is similar to that observed in healthy young individuals, although they present a marked decreased in the granulocytic and, specially, in the erythroid clonogenic potential.

Erythroid differentiation observed for CD34+ isolated from MDS was aberrant in all the cases tested. At day 7, cells showed a delay in differentiation compared to healthy individuals, with more cells at stages I and II, and with barely any cells in stage IV. At day 14, the delayed differentiation is more evident, while in healthy CD34+ cells around 70-80% of the cells reach stage IV at this time point, for MDS cases this was not the case: in some cases, cells barely reached stage IV, whereas in others they reached it to a lower percentage than healthy cells. At this time, cells accumulated in stage III (CD71+ CD235a+), and in some patients cells also accumulated in stage I (CD71- CD235a-). Regarding clonogenic assays, the patients tested demonstrated a very pronounced and statistically significant decrease in the number of CFU-G/M/GM ($p < 0.01$) and BFU-E ($p < 0.01$) colonies formed: MDS patients showed either one or none BFU-E colonies, and very few CFU-G/M/GM colonies per 1,000 cells seeded. Thus, all the MDS patients tested demonstrated a marked aberrant erythroid differentiation and colony formation capabilities.

The abnormal erythroid differentiation of untransduced CD34+ cells from MDS patients in the liquid differentiation system has been mentioned in the revised manuscript:

Lines 317-320: “When submitted to the differentiation system, CD34+ cells from MDS patients show a clear delay in erythropoiesis, barely reaching stage IV or reaching at a much lower percentage at day 14 than healthy cells (data not shown).”

Figure 12. (A-C) Left: flow cytometry charts representing advanced erythroid differentiation (CD71 and CD235a markers; stages I-IV) of CD34+ cells from healthy young (A) (biological replicates are depicted as average \pm SD), 3 healthy elderly (B), and 5 MDS cases (C) at days 7 and 14 of differentiation. Right: bar-plots representing the percentage of cells observed in A-C in stages I-IV. (D) Graph indicating the number of colonies (BFU-E and G/M/GM) obtained for CD34+ cells from young healthy donors (3 biological replicates) (left), from 3 healthy elderly (center) and 2 MDS patients (right). For healthy elderly and MDS patients the average \pm SD of technical replicates is represented.

Finally, although challenging, experiments with *in vivo* modeling of the role of *DDIT3* in humanized mouse models for MDS (<https://doi.org/10.1038/s41467-018-08166-x>) or in ossicle growth of cancer cells (doi: 10.3324/haematol.2019.233320) will further strengthen the translational relevance of the manuscript.

Following this question, we tried our overexpression system in an *in vivo* context, in a mouse model based on humanized microenvironment ossicle formation. This model has been recently reported as useful for MDS patient derived xenograft (PDX) engraftment³. Briefly, CD34+ cells from healthy young donors were FACS-isolated and transduced with lentiviruses harboring *DDIT3* or a control plasmid. Four days after infection, 1×10^5 bone marrow derived human mesenchymal stem cells were seeded along with 2.5×10^5 of the target hematopoietic cells, in each gelatin sponge. These cell-carrier structures were subcutaneously implanted in NOD-SCID mice. Six weeks later, animals were subjected to computed tomography scan to confirm ossicle formation (Fig. 13A), samples were retrieved, and cells were isolated and processed for flow cytometry analysis.

Using an anti-human CD45 antibody we were able to detect around 2% of human hematopoietic cells in the inserts, which represented a low engraftment (**Fig. 13B**). We hypothesized that the low engraftment observed may be related to the mouse strain selected for the study (NOD-SCID mouse). Although this ossicle formation model was designed to create humanized bone and bone marrow microenvironment, the original report assessed the MDS engraftment in more severely immunocompromised mouse strain models, as NSG, and/or complemented with human cytokine expression, as NSG-SGM3 mouse. Other successful MDS xenografts are also based on humanized immunocompromised mouse strains, as MISTRG model. Therefore, data indicate that, for proper high MDS engraftment in humanized ossicle implantation model, severe immunocompromised, or humanized, mouse strains are required.

In any case, among the engrafted human hematopoietic cells, we observed a low percentage of GFP+ cells in the samples harvested at 6 weeks (**Fig. 13C**). This percentage was specially low for *DDIT3*-overexpressing cells, suggesting a very low engraftment or a loss of these cells in this *in vivo* setting over time. Furthermore, we observed how the GFP positive cells did not seem to undergo erythroid differentiation in this system, as they expressed CD13, and did not acquire CD235a and barely showed CD71 expression (**Fig. 13B**). Thus, unfortunately, we were unable to draw any conclusions for *DDIT3* overexpression effects on erythroid differentiation using this *in vivo* system.

Figure 13. *DDIT3* overexpression system in a mouse model based on humanized microenvironment ossicle formation. **(A)** Computed tomography image of a mouse to confirm ossicle formation. The ossicle is indicated in the red circle and it is shown in a picture on the right. **(B)** Flow cytometry analysis strategy for infected cells: once singlets are selected based on FSC-A and FSC-H, dead cells are excluded by 7-AAD staining. Afterwards, debris is eliminated by FSC-A and SSC-A gating, and human hematopoietic cells are selected by human CD45 expression. From those cells, infected cells are selected by GFP expression, and erythroid differentiation is analyzed by CD71 and CD235a markers. **(C)** Graph depicting the percentage of GFP+ cells from human CD45 population for control and *DDIT3*-overexpressing cells after 6 weeks of ossicle implantation.

Minor points:

As Clonal Hematopoiesis of indeterminate potential (CHIP) has emerged as key regulator of HSC aging and age-associated malignancies, the authors should also report whether the aged samples under study have clonal expansion and/or CHIP mutations that could explain the transcriptional phenotypes observed.

We agree with the reviewer that this is an interesting aspect to consider. Due to limitations in the amount of sample obtained for elderly healthy donors (specimens obtained from orthopedic surgeries), we were only able to perform mutational analysis of 5 of the 8 individuals included in the study. Specifically, we used the leftover material generated in the RNA extraction from HSCs and other hematopoietic progenitors to analyze DNA. Thus, the mutational analysis was not performed on bone marrow mononuclear cells but on hematopoietic stem and progenitor cells, which have been described to be the origin of mutations in CHIP and MDS. In these specimens, we analyzed the set of genes associated with CHIP. We obtained an average coverage per gene of 1000x for 3 of the samples (EH_5, EH_7 and EH_8). For the other two, all of the genes reached the 250X and most of them the 500x coverage. The mean coverage of each gene in each individual is represented in **figure 14A**. Although some of the samples presented a lower coverage

than usual for these type of mutations, we believe that the fact that this analysis was performed on the specific cell types mentioned increases the chances of finding putative CHIP mutations.

From the specimens analyzed, only one of them, EH_8, showed a mutation. In particular, this individual presented a mutation categorized as likely pathogenic (ACMG categorization) in DNMT3A (DNMT3A:25458599-GA-G) at a VAF of 3% (Fig. 14B). The mutational information for the healthy elderly individuals included in this study has been incorporated in Table S1 of the revised manuscript.

B

Sample_ID	Categorization	gene	type	codingConsequence	chromosome	genome_position	depth	VAF(%)	c.DNA	protein
EH_8	Likely pathogenic	DNMT3A	INDEL	frameshift	2	25458599	3697	3.1	c.2573del	p.(Ile858Thrfs*23)

Figure 14. Mutational analysis of healthy elderly samples. (A) Bar plots depicting the average coverage per gene for each of the specimens analyzed. The percentage (average per gene) of base pairs covered at 250x, 500x, and 1000x is indicated per gene. (B) Characteristics of the mutation found in sample EH_8.

Line 101. Given the phenotypic markers used (CD34+, CD38-, CD90+ CD45RA-) I would substitute “highly purified HSCs” with HSCs enriched subsets here and throughout the manuscript.

We agree with the reviewer, and have now changed “highly purified HSCs” for “HSCs enriched subsets” throughout the manuscript.

Line 149. A Gene ontology analysis is presented in Figure 1G and the related GO terms are reported in Table 6. Adding the manual curated list in the same table would help data reproducibility.

We thank the reviewer for the comment, and agree that adding the manual curated list can help data interpretation and reproducibility. As suggested, the main functional categories represented in Figure 1G of the manuscript have been added to Table S6 of the revised manuscript.

Line 265. The authors present some results regarding the pseudotime analysis (Fig 5D), it would be interesting to visualize the pseudotime trends for each cell in the UMAP.

We thank the reviewer for the comment. In figure 15, we analyzed the distribution of pseudotime in both control and *DDIT3*-overexpressing cells. As expected, we found that pseudotime gradually increases as cells advance through the differentiation (from HSC to more differentiated cells) (Fig. 15).

Figure 15. UMAP plot of the pseudotime of control and *DDIT3*-overexpressing cells subjected to *ex vivo* differentiation for 14 days.

In the scRNA analysis the myeloid clusters are eliminated because *DDIT3* is a transcription factor for erythropoiesis. Can the authors add something or discuss more in depth about changes in differentiation within myeloid clusters?

As indicated by the reviewer, although some myeloid clusters were found in the scRNA-seq analysis (**Fig. S4F of the revised manuscript**), our analysis mainly focused on clusters representing erythropoiesis, as our experiments indicated that the effect of *DDIT3* overexpression was observed on this lineage. Nevertheless, following this question, we have studied in more depth the myeloid clusters detected in the scRNA-seq analysis. We detected 2 clusters out of 9 corresponding to myeloid cells. Cluster 7, with 442 cells (5% of total), expressed markers of progenitors at multiple steps of differentiation: LMPP, CMP, GMP

and neutrophils. Cluster 9, with 145 cells (1.6% of total), was composed by basophil progenitors (Fig. 16A, 16B). Cluster 7 showed very low expression of *DDIT3* (Fig. 16C-D), and the overexpression of this factor was barely visible in these cells, whereas cluster 9 showed no expression of *DDIT3* for either of the conditions (Fig. 16C-D). We included the myeloid clusters in the study of cell proportions and found that

Figure 16. (A) UMAP plot of the transcriptome of cells subjected to *ex vivo* differentiation for 14 days. Cells are colored by the clusters that resulted from unsupervised clustering. (B) Dot plot of canonical myeloid cluster markers for the different identified cellular subpopulations, combining the 2 myeloid clusters and the erythroid subpopulations. Dot size represents percentage of cells expressing each marker and color represents average expression for the corresponding gene and subpopulation. (C) UMAP plot of the cells showed in A, depicting the log normalized expression of *DDIT3*. (D) Violin plot representing the log normalized expression of *DDIT3* in each of the cell subpopulations, for control cells (blue) and *DDIT3*-overexpressing cells (red). (E) Barplot representing the proportions of cells in each of the cell subpopulations for control cells (blue) or cells overexpressing *DDIT3* (red) upon *ex vivo* differentiation for 14 days.

both of them account for a bigger proportion of the cells overexpressing *DDIT3* (**Fig. 16E**). In the liquid differentiation system we did not observe any effect on myeloid differentiation upon *DDIT3* overexpression. Taking into consideration that these clusters represent a minority and do not maintain *DDIT3* overexpression, we believe that it is not possible to conclude that the factor affects this hematopoietic lineage.

Color code for panel MMP2 in Fig. 2C across young/elderly/MDS is not consistent with the other panels

Thank you so much for noticing this error, we have changed the color palette of the MMP2 plot to match the other panels.

Please specify the source of CD34+ cells used in the study across all ex-vivo experiments (especially in Fig. 3 and 4).

We apologize for not including these data in our previous version. We have now indicated the source of the CD34+ cells used in each of the *ex vivo* experiments performed. Information of each healthy donor included in the study is now included in Table S1.

Reviewer #3 (Remarks to the Author): with expertise in MDS, erythropoiesis

In this paper, the authors study highly purified normal and malignant stem cells based on surface markers from young and old subjects and lower-risk MDS patients. They examine the transcriptomes from these subjects and grouped these into 8 different expression patterns. They suggest that aging results in a transcriptomic state that predisposes to developing a myeloid malignancy, upon which additional changes result in MDS. In particular, they focus on the transcription factor DDIT3, which when overexpressed resulted in an MDS-like transcriptional state and suppression of erythropoiesis in an in vitro model.

This is an interesting paper and the experiments are generally well conducted. The selection of purified cells for these studies is a strength, and the finding that DDIT3 may be responsible for some cases of MDS-related anemia is convincing.

Less convincing is the idea proposed that the transcriptional state of old stem cells is more like that of MDS stem cells than young stem cells. Several questions arise.

First, given that the authors are flow-sorting cells based on an immunophenotype, it is not clear how they are separating normal from malignant stem/progenitor cells for their transcriptomic studies. Some detail should be provided.

We thank the reviewer for the comment and apologize if this was not clear in the text. In our work, HSCs were isolated based on immunophenotype (CD34+ CD38- CD90+ CD45RA-), in healthy young individuals (healthy volunteers), healthy elderly donors (patients undergoing orthopedic surgery), and in MDS patients. In the last group, we did not separate normal from malignant HSCs as, to our knowledge, there are no markers that segregate such populations. Although we are aware that, in this case, a limitation of our bulk transcriptomic studies is that we are analyzing both populations, previous studies have shown that in MDS patients, normal hematopoiesis is minimal, with the majority of hematopoiesis being driven by malignant cells⁴. Supporting this idea, a recent work demonstrated that although MDS cells are less proliferative than healthy cells, they are able to suppress normal hematopoietic differentiation, becoming dominant against normal hematopoiesis⁵. Moreover, this idea is further supported by a recent work performed by the group of Dan Landau (BioRxiv: doi: <https://doi.org/10.1101/2022.06.08.495292>), in which CD34+ cells from MDS samples with ringed sideroblasts were analyzed using single-cell DNA and RNA technologies. Among other results, this work demonstrated a significantly higher number of mutant versus wild-type cells (wt: n=2,498; Mut: n=9,996 cells), suggesting that residual normal hematopoiesis is minimal in these patients. Furthermore, higher mutant cell frequency was found in several cell types, including HSPCs, suggesting that the majority of the compartment harboring HSCs was composed of malignant cells.

Collectively, these data suggest that although the analysis of MDS patients carried out in this work was not performed exclusively on malignant HSCs, the contamination due to healthy HSCs may be minimal.

Second, while the PCA plot does seem to separate young stem cells from old and MDS (Fig 1c), the hierarchical clustering results (Fig. S1B) are not nearly as convincing. 4 MDS patients are in the most distant cluster from the rest of the samples, all on their own (left-most group). 4 MDS samples are in a cluster with elderly patients only (2nd group from the left). 1 MDS sample is in a mixed cluster with elderly and young patients (3rd group from the left). 3 MDS samples are in cluster with young patients only (the rest of the groups). For any branch in the tree, it can be rotated 180 degrees around its axis. If any of the higher-level branches were rotated in this way, then the apparent separation of young and old samples, and the apparent clustering of MDS with old, would not be evident any more. Further Differential expression

analysis showed more genes deregulated between MDS and elderly healthy samples than between healthy elderly and young HSCs.

We thank the reviewer for this thoughtful comment, and after taking a closer look at the hierarchical clustering shown in Fig. S1B, we agree that it does not support the idea of greater transcriptional differences in aging than between healthy elderly and MDS samples. Furthermore, we agree with the reviewer that PCA cannot quantitatively analyze the data, and therefore, it is not demonstrated that transcriptional differences between healthy young and elderly HSCs are greater than those detected between healthy elderly and MDS samples. Nevertheless, PCA analysis does show that differences between HSCs from young and from elderly donors (both healthy and MDS), which are encoded in PC1, are driven by different genes (likely age-related) than differences between healthy elderly and MDS, which are encoded by PC2, and may be driven by disease-related genes. This is further validated by gene expression analyses showing different sets of differentially expressed genes with only partial overlap between young and elderly healthy samples, and healthy elderly and MDS HSCs (**Fig. 1D of the revised manuscript**). In order to clarify our observations regarding gene expression differences between the different types of HSCs, we have eliminated Fig. S1B and have changed the text of the revised manuscript to:

Lines 121-125: *“Principal component analysis (PCA) showed transcriptional differences between young and elderly (both healthy and MDS) HSCs, which were encoded by PC1 (Fig. 1C), and additional differences between elderly and MDS patients, which were encoded by PC2. Differential expression analysis demonstrated 733 genes deregulated between young and elderly healthy samples, and 907 genes between healthy elderly and MDS HSCs”.*

At the bottom of p. 6, the authors refer to Figs. 2I and J, but there is no Fig. 2I or J. Similarly other labels are incorrect, and should be checked through the manuscript (e.g. Fig. 2K).

We apologize for the typos in the figure numbers. All references to figures have been checked and corrected throughout the text.

It is not clear why stress inflammation goes down with MDS compared to old in Fig. 1G when MDS is associated with an inflammatory state?

We thank the reviewer for noticing these data. It was also very surprising to us to find that the stress/inflammation response gene expression profile of HSCs from MDS patients was similar to that of healthy young HSCs, and lower to that of healthy elderly HSCs. We expected to see the increase in the expression of these genes in healthy elderly samples, as aging is known to be associated with a more inflammatory bone marrow microenvironment, but also expected to detect augmented expression of these genes in MDS patients, due to the vast literature showing increase inflammation response in the disease.

In order to test whether the observations at the transcriptional level were translated into functional differences, we performed *ex vivo* functional analyses starting from primary HSCs. We isolated HSCs from healthy young, healthy elderly, and MDS patients, treated them with lipopolysaccharide (LPS) at 10 and 100 ng/mL for 6 and 24 hours, and evaluated their transcriptional profile by low-input RNA-seq. Our data showed that healthy elderly HSCs presented increased number of differentially expressed genes (DEGs) upon treatment (treatment vs vehicle) compared to healthy young cells, indicating an increased response to inflammation stimuli with age. Interestingly, HSCs from MDS patients showed a similar response to that observed for young cells (**Fig. 17**). These data suggest that MDS HSCs may be somehow more resistant to the increased inflammatory signaling present in the bone marrow microenvironment of these patients.

The discrepancy between our data and previously published data may be due to the fact that, to our knowledge, most previous studies on stress/inflammation in MDS have not been performed on HSCs but in CD34+ or even on bone marrow mononuclear cell populations. and thus, our data suggest that this minor

population may behave differently. Nevertheless, we believe that these unexpected observations require further validation and analyses, that will be performed in future studies.

Figure 17. Graphs depicting the number of differentially expressed genes (compared to cells treated with vehicle) for HSCs from healthy young, healthy elderly, and MDS specimens treated with LPS at 10 or 100 ng/mL for 6 (left) or 24 (right) hours.

The choice of clusters that the authors choose to compare to make there is inconsistent and lacks a clear rationale. For example, In Fig. 2A and B, they mainly examine and compared clusters C1, C2, 5 and 6, but 1 and 2 show no difference between aged and MDS, so it is unclear where the Cancer associated genesets are coming from? It appears that clusters C3 and C4 are perhaps the most important clusters to examine closely as this is where unique MDS-related expression patterns reside.

We apologize if the description of the potential biological role of the clusters of genes found in the transcriptional analysis was not clear in the manuscript. In order to avoid overloading the main text with a long descriptive section, many of the potential biological roles, as well as examples of specific genes from different clusters were included in supplementary figure 2. In the main text, we narrowed the information to make two main points (also addressed in response to a question from reviewer 2):

- We noted that transcriptional lesions naturally occurring in aging may predispose HSCs for transformation, which is in agreement with previous observations^{1,2}. We reasoned that such alterations include genes altered exclusively with aging (C1 and C2), as well as alterations found in aging that are even more pronounced in MDS patients (C5 and C6). For this reason we analyzed the potential involvement of these 4 clusters in transformation. To make more clear our reasoning for including C1, C2, C5 and C6 in the same analysis, we have modified the manuscript as follows:

Lines: 149-174: *“Among other relevant findings, some of which are detailed in the supplementary data (Fig. S2), these analyses identified alterations that could be related to the predisposition to the development of hematological malignancies. Interestingly, some of such alterations were detected in healthy elderly HSCs, suggesting that transcriptional lesions naturally occurring in aging could predispose HSCs for malignant transformation. For example, processes related to cell proliferation were mainly enriched in C2 but also in C6 (Fig. S2A, S2E, S2I), suggesting that an important loss of proliferation activity of HSCs takes place during aging, and it is further exacerbated in MDS, an event that has been previously associated with accumulation of DNA damage³. Furthermore, DNA sensing and repair processes were also enriched in C2 and C6 (Fig. S2A, S2E, S2I), suggesting a progressive loss of the ability to overcome DNA insults and an increased genomic instability of HSCs during aging and MDS, with a potential direct role in disease development. In order to further explore the potential role of transcriptional lesions occurring with age in the promotion of a myeloid disease/cancer susceptibility, we performed gene set enrichment analyses (GSEA) on clusters altered in aging, with*

or without exacerbation of expression in MDS (C1, C2, C5 and C6). These analyses demonstrated that genes contained in such clusters were enriched in cancer-related signatures (Fig. 2A); furthermore, among such genes, we identified factors with known roles in the development or maintenance of myeloid malignancies (Fig. 2B). Some examples included the upregulation of TRIB1, a transforming gene for myeloid cells⁴, and the matrix metalloproteinase MMP2, which presents high secretion levels in AML⁵; as well as the downregulation of genes such as adenosine deaminase ADA2, and the mini-chromosome maintenance proteins MCM7⁶ and MCM4, whose repression has been involved in myeloid leukemias (Fig. 2C). These results suggest, in line with a previous work performed in mice, that aging-derived transcriptional changes predispose HSCs for transformation. Such lesions may not be sufficient to promote disease development, and may need additional alterations to lead to an MDS phenotype, which may include enhanced alteration of the lesions found in aging (C5 and C6), genetic insults, or transcriptional alterations that occur exclusively in MDS HSCs.”

- We agree with the reviewer in that C3 and C4 are probably the most relevant clusters to examine closely. Therefore, the second point that we wanted to make is that genes exclusively altered in MDS (C3 and C4) may represent lesions with a potential direct role in MDS development. For these reason, as the review suggested, we examined more closely those clusters, performing GO and GSEA analyses and looking at specific genes. This is explained in the next paragraph of the revised manuscript:

Lines 176-188: “Thus, we next focused on clusters showing exclusive deregulation in MDS (C3, C4), as they represented lesions with a potential direct role in MDS development. GSEA and GO analyses (Fig. 2D, 2E) demonstrated diminished expression of genes controlling cell division and DNA repair (Fig. S2K), further suggesting an accumulation of genetic lesions in MDS-associated HSCs. Decreased levels of genes regulating cell-substrate adhesion and B-cell differentiation as well as increased expression of genes involved in processes such as miRNA processing, exocytosis, type I IFN production, or response to TGF- β were also detected in MDS-associated HSCs (Fig. S2L). Finally, increased expression of chromatin and transcriptional regulators, such as DDIT3, SERTAD2, RELA, EBF4 or ZNF219, among others (Fig. 2F), suggested an MDS-specific epigenetic and transcriptomic reprogramming that could potentially guide the aberrant phenotype of these cells. Collectively, these transcriptional changes are consistent with an aging-mediated predisposition towards myeloid transformation, and with MDS-specific alterations that may contribute to the development of the disease.”

I have some concern about the use of the MaSigPro R package for comparison of the healthy young, healthy old, and MDS groups (Fig. 1F, G). The package is specifically designed for time series data, which would imply that the young, old, and MDS samples would have to be harvested from the same group of individuals across time (this was not the case). The authors to explain their statistical reasoning that MaSigPro is appropriate for comparing 3 groups of distinct individuals as if it were a time series. Particularly concerning is the notion that the MDS group naturally comes after the Elderly group in time. Are the MDS patients actually older than the Elderly healthy controls? I understand why it seems logical that MDS comes after Elderly, based on what we know about MDS as a disease of the elderly, but it's a big assumption for a time series analysis.

As the reviewer indicated, the three sample series (healthy young, healthy elderly and MDS) used in this study were not longitudinal, and they showed a median age of 20, 65.5 and 70 years old, respectively (**Fig. 18**). Thus, MDS patients showed a very similar age than healthy elderly individuals. While we acknowledge that it is not strictly longitudinal data, we considered that the fact that MDS comes naturally with age, could support the type of approach carried out. Moreover, taking into account the limitations mentioned, we also considered that the type of analysis performed could really give novel insight into the molecular pathogenesis of the disease, uncovering the transcriptional alterations that take place in the

HSCs of MDS patients, in the context of aging. Furthermore, note that while MaSigPro was originally intended to model time-series data, the underlying statistical model can also be used to model data that is longitudinally ordered, as the one in the manuscript. Specifically, MaSigPro models the gene expression through second-order polynomial regression based on two variables: condition (z) and time (t). Note that under this model, time is not restricted to be purely time-series data of the same sample, as the variable t could as well capture the effect of the longitudinal position of the sample in longitudinal data. Thus, for all the reasons above mentioned and since the main purpose of the analyses was to cluster genes based on trajectories, we considered that the use of MaSigPro was appropriate to find clusters of genes following different dynamics of expression in aging and MDS.

Figure 18. Age of the three groups included in the study. Median +/- SD is represented.

A propos the above statements, the authors should show the GSEA enrichment plots for Elderly vs MDS as this would be more relevant to their thesis than comparing Young to MDS.

We believe the reviewer refers to figures 3E and 3F of the manuscript. In those figures, we generated the signatures *Young_vs_MDS_HSCs_Up* and *Young_vs_MDS_HSCs_Down* from the genes differentially expressed between those two groups (young and MDS HSCs). The selection of healthy young HSCs instead of healthy elderly samples was due to the fact that in the experiment performed in figure 3, *DDIT3* was overexpressed in healthy young HSCs due to limiting cell numbers of healthy elderly HSCs, and to the fact that *DDIT3* is expressed at low levels in both types of healthy samples. Thus, as overexpression was being induced in young HSCs, we considered that generating the gene signatures for GSEA analyses using young vs MDS HSCs was more appropriate.

Nevertheless, as the reviewer suggested, we also produced GSEA enrichment plots using gene signatures generated from the genes differentially expressed between healthy elderly and MDS HSCs (**Fig. 19**). We observed that genes altered upon *DDIT3* overexpression in HSCs were negatively enriched in genes downregulated in MDS samples compared to healthy elderly cells, whereas the enrichment in genes upregulated in such cells was positive but did not reach statistical significance.

Thus, producing the GSEA enrichment plots utilizing the gene signatures generated from the genes differentially expressed between healthy elderly and MDS HSCs also supported and “MDS-like” transcriptional profile promoted by *DDIT3*-overexpression in HSCs. Nevertheless, from our perspective, and for the reasons explained above, we believe that generating the gene signatures for GSEA analyses using young vs MDS HSCs is more appropriate for this particular experiment.

Fig. 19. GSEA plots depicting the enrichment upon *DDIT3* overexpression in gene signatures representing genes up- (A) and down-regulated (B) in HSCs from MDS compared to HSCs from healthy elderly individuals. The NES and adjusted p-adjusted values are indicated for each signature.

In Fig. 4, it is not clear what the expression of *DDIT3* in stages of erythroid differentiation means given it is overexpressed as a transgene. Does this mean that the transgene under an exogenous promoter is upregulated or are the authors looking specifically at endogenous *DDIT3*? As well, *DDIT3* is not seen to be expressed in the CD34 cells - it is possible that the transduction is mainly happening in erythroid progenitors hence the effect seen. In other words, are the authors sure that they are able to transduce HSC rather than progenitors, hence observing a lineage-specific effect rather than an effect that is actually reflective of what is happening in the HSC?

We apologize if this figure was not clear. Figure S4F depicts the total expression of *DDIT3* (endogenous and exogenous) in control and *DDIT3*-overexpressing cells at different stages of erythroid differentiation, detected by scRNA-seq. We meant to show that the overexpression of *DDIT3* is maintained throughout erythropoiesis in our system. It can also be observed that *DDIT3* expression rises physiologically as erythroid differentiation progresses, nevertheless, *DDIT3*-overexpressing cells maintain higher levels than control cells. CD34+ cells barely show any expression, which may be due to the fact that at that time point we barely detected any CD34+ cell (only 3 cells in this stage).

We agree with the reviewer that it is a key experiment to perform the differentiation experiments with HSCs in order to determine whether it produces the same effect than when CD34+ progenitor cells are transduced. As shown in **figure 5** (reponse to reviewer 1), we carried out liquid differentiation upon *DDIT3* overexpression in HSCs. *DDIT3* overexpression produced a delay in erythroid differentiation that was very similar to that observed for CD34+ cells, with a 30% decrease in the percentage of cells in stage III, and an accumulation of cells at stage II at 14 days of differentiation. Furthermore, at a later time point of 17 days, 40-45% of control cells had progressed to stage IV whereas only 12% of *DDIT3*-overexpressing cells reach such state.

As indicated in the response to reviewer 1, in our perspective, these results would indicate that it is unlikely that the observed results are due to a bias related to the use of CD34+ cells instead of an enriched population of HSCs.

In Fig. 4F it is not evident that there is "shorter, thinner and less dense streamlines" in the *DDIT3* populations. In which trajectories are they seeing this? The authors should quantify this effect and analyze it statistically.

We thank the reviewer for the comment and apologize if the data were not clear enough. Moreover, we agree that a quantification of the streamlines can help interpret the data. In the analysis performed using

velocity there is a unique trajectory, going from immature to more mature erythroid progenitors (from CFU to reticulocytes). It is possible to quantify the streamlines provided by velocity using two metrics:

- length of velocity, which represents the speed or rate of differentiation.
- confidence, which represents the coherence of the vector field (i.e: how a velocity vector correlates with its neighboring velocities).

We have represented the length of velocity for control and *DDIT3*-overexpressing cells, and it can be seen that the vector length is lost as erythropoiesis progresses, reaching very low scores from late basophilic to reticulocyte stages, compared to control cells, implying that the differentiation at later stages is delayed (**Fig. 20A, B**). Furthermore, the confidence for *DDIT3*-overexpressing cells at advance stages of differentiation is not as strong as that of control cells (**Fig. 20C**), also implying a weaker differentiation potential upon overexpression of the factor. Quantification for length and confidence can be found in **figure 20B and 20D** along with statistical significance for vector length and confidence.

We have included the length of velocity, which we believe is easier to interpret, and its quantification and statistics in figure 4 (Fig. 4G, 4H) of the revised manuscript, and have change the text in the manuscript to:

Lines 256-259: “*DDIT3*-upregulated cells showed a statistically significant decrease in RNA velocity from late basophilic to reticulocyte states, which could be clearly seen when length of velocity, which represents the speed or rate of differentiation was quantified (Fig. 4G, 4H). These results suggest an impairment of the correct differentiation of the later stages of erythroblasts.”

Figure 20. (A, C) UMAPs representing the length of velocity (A) or confidence (C) for control and *DDIT3*-overexpressing cells. (B, D) Box plots representing the length of velocity (B) or confidence (D) for each erythroid differentiation state for control and *DDIT3*-overexpressing cells. Statistically significant differences are indicated (**p<0.01; ***p<0.001; ns: non-significant).

In Fig 5F, the authors provide a list of differentially expressed factors on p. 10 of the text, but Not all the data are shown. They should either limit the list of factors or show data for all.

We thank the reviewer for the comment and apologize if the text was not clear. All the factors with differential activity can be seen in the heatmap in figure 5E, although some of them are shown in detail in figure 5F and S5F of the revised manuscript. We have now also referenced figure 5E, and indicated that details for some of them can be found in figures 5F and S5F, so it is easier for the reader to follow.

REFERENCES

1. Rossi DJ, Bryder D, Zahn JM, Ahlenius H, Sonu R, Wagers AJ, et al. Cell intrinsic alterations underlie hematopoietic stem cell aging. *Proc Natl Acad Sci U S A* 2005; 102I(26): 9194-9.
2. Adelman ER, Huang HT, Roisman A, Olsson A, Colaprico A, Qin T, et al. Aging Human Hematopoietic Stem Cells Manifest Profound Epigenetic Reprogramming of Enhancers That May Predispose to Leukemia. *Cancer Discov* 2019; 9I(8): 1080-1101.
3. Mian SA, Abarrategi A, Kong KL, Rouault-Pierre K, Wood H, Oedekoven CA, et al. Ectopic humanized mesenchymal niche in mice enables robust engraftment of myelodysplastic stem cells. *Blood Cancer Discov* 2021; 2I(2): 135-145.
4. Zhan D and Park CY. Stem Cells in the Myelodysplastic Syndromes. *Front Aging* 2021; 2I(7)19010.
5. Hayashi Y, Kawabata KC, Tanaka Y, Uehara Y, Mabuchi Y, Murakami K, et al. MDS cells impair osteolineage differentiation of MSCs via extracellular vesicles to suppress normal hematopoiesis. *Cell Rep* 2022; 39I(6): 110805.

REVIEWERS' COMMENTS

Reviewer #1 (Remarks to the Author):

I acknowledge the very significant amount of work that went into the review and rebuttal process, which addressed most of my comments. I note that a significant part of the additional work was not included in the revised manuscript. I am acutely aware of space limitations, but several major revisions were requested and performed, and they should be included in the manuscript in support of the conclusions, for the benefit of the readers.

The main points, in detail:

(1) The Authors should include validation of overexpression and knockdown approaches when they are first used, straight after the experimental diagrams. WB analysis of K562 cells is perfectly acceptable, but please note that it is confusing that K562 cells have no expression of DDIT3 in the control sample for overexpression, and yet this is present in the control samples for knockdown albeit at variable levels. This may be justified by the loading, but absence of DDIT3 in K562 cells is unexpected and the controls should be in agreement. Could this please be reassessed and included in the manuscript? Inclusion of the full blots will further support band specificity and the elegant use of immunofluorescence in primary cells. Inclusion of wider fields with more cells could be helpful, and analysis of knockdown cells should also be shown.

(2) The Authors present additional evidence for an erythroid differentiation delay upon DDIT3 overexpression. In the revised manuscript, this is particularly supported by the differentiation velocity of CD34+ cells with DDIT3 analysed by scRNA-seq. Although the data are in agreement with flow cytometry, it is not necessarily straightforward that RNA velocity-measured splicing variants cannot be influenced by, for example, cell divisional status. I acknowledge that the Authors have included data comparing cell cycle and apoptosis of control and DDIT3 cells, but it would have been more conclusive to perform the analyses within phenotypically-defined sub-populations to avoid averaging, as previously discussed. I am confused by the new panels G and H in Figure 4, as the values plotted do not seem to match the colour code. Can the Authors please clarify in the text and/or Figure Legend? Thank you. I would also recommend including the data on morphology and size of DDIT3-overexpressing BFU-E, as these data link with and strengthen the analysis of MDS samples and the conclusions drawn.

(3) The inclusion of additional MDS samples treated with DDIT3shRNA, as well as the quantification of knockdown strengthen the manuscript. This is particularly important since the presentation is descriptive for individual samples and there is variation between samples on the extent of knockdown effect in stage IV cells, and stage III abundances are contradictory between samples. To this end, I would also recommend including in the manuscript the overexpression of DDIT3 in the low-expressing MDS16 and MDS17, as they give support to the model of DDIT3-driven dyserythropoiesis in MDS.

Reviewer #2 (Remarks to the Author):

The authors have significantly improved the manuscript by providing additional mechanistic studies on the role of DDIT3 in dyserythropoiesis. Even if some of the raised questions remained unanswered (is there DNA replication stress upon DDIT3 manipulation in HSCs?; is DDIT3 a chromatin bound regulator of erythroid differentiation?) this Reviewer acknowledges the difficulties of performing such experiments with limited amount of HSC-enriched subsets and remains in favor of publication given the potential translational relevance of the main findings.

For statistical analyses: Line 743 and 753 (main text - merged file) - Missing name of the statistical test performed. Figure 7 A-B (rebuttal file) - Missing name of the statistical test performed. As a general comment, prior to publication I recommend to remove any statistical analysis from experiments for which n= biological replicates is equal or less than 3.

Reviewer #3 (Remarks to the Author):

I continue to have some reservations about how this data is presented. On the one hand time series is set up as young > elderly > MDS, and on the other the authors argue that to look for MDS-related rather than age-related gene expression changes that it is better to use young vs MDS, because they use cells from "young" donors. Their explanation is somewhat convoluted. And the lack of clarity in the writing makes their rationale difficult to follow. I would suggest that a reworking of key sections of the paper comparing young, old and MDS sample is necessary to be absolutely clear about rationale of their studies.

Fig 19 in the Rebuttal should be shown in the manuscript. The statement "the enrichment in genes upregulated in such cells was positive but did not reach statistical significance" is incorrect given the Padj-value of 0.94. The authors should limit their interpretation to what the data show, which is that there is no difference in the upregulated genes.

Figs 2C and F should have p-values provided for the comparisons.

RESPONSE TO REFEREES

Please, find below a detailed point-by-point answer to all the comments raised in the review process and an explanation on how we have addressed each of the criticisms and suggestions in the new version of the manuscript and supplemental material. Our responses are written in blue font, while changes in the manuscript are indicated in red font.

Reviewer #1 (Remarks to the Author):

I acknowledge the very significant amount of work that went into the review and rebuttal process, which addressed most of my comments. I note that a significant part of the additional work was not included in the revised manuscript. I am acutely aware of space limitations, but several major revisions were requested and performed, and they should be included in the manuscript in support of the conclusions, for the benefit of the readers.

The main points, in detail:

(1) The Authors should include validation of overexpression and knockdown approaches when they are first used, straight after the experimental diagrams. WB analysis of K562 cells is perfectly acceptable, but please note that it is confusing that K562 cells have no expression of DDIT3 in the control sample for overexpression, and yet this is present in the control samples for knockdown albeit at variable levels. This may be justified by the loading, but absence of DDIT3 in K562 cells is unexpected and the controls should be in agreement. Could this please be reassessed and included in the manuscript? Inclusion of the full blots will further support band specificity and the elegant use of immunofluorescence in primary cells. Inclusion of wider fields with more cells could be helpful, and analysis of knockdown cells should also be shown.

We thank the reviewer for the comment and agree that a demonstration of the efficacy of the overexpression and knockdown systems at the protein level is important for the work and for the reader.

Regarding the immunoblot, we apologize if the controls of the overexpression and the inhibition systems are confusing. We would like to explain that the lack of band in the control of the overexpression system does not imply that these cell line does not express *DDIT3* (as shown in the immunoblot demonstrating the knockdown system), but rather it is due to the high overexpression achieved in the cell line, and the relatively low levels in control cells in comparison.

Regarding immunofluorescence, although we acknowledge that the inclusion of wider fields from the immunofluorescence showing more cells could be helpful, there were technical limitations derived from the primary nature of the cells used. Firstly, the cell type used (CD34+ cells) represents a very low percentage of the bone marrow mononuclear cell fraction, and thus, the number of cells to start the whole process was low. These cells were isolated from MDS patients or healthy donors, transduced with lentiviruses and sorted based on GFP expression. As CD34+ cells are difficult to transduce, the number of cells we obtained to start with the immunofluorescence was even lower. Moreover, several steps of the immunofluorescence, including fixing cells

on a slide, incubations, and washes lead to cell loss and thus very few cells remained attached at the end of the process. However, besides the low number of cells, we believe that images obtained were clear enough to demonstrate the efficacy of the overexpression and knockdown systems utilized in this work. On the one hand, *DDIT3* overexpression observed by immunofluorescence was more physiological than that observed in the K562, and recapitulated the endogenous upregulation observed in MDS patients. On the other hand, *DDIT3* inhibition was sufficient to enhance erythroid differentiation in MDS patients.

As suggested, we have included the immunoblots performed in the K562 cell line and immunofluorescence done in primary cells for the overexpression (**Fig. S3c, s3d of the revised manuscript**), and knockdown system (**Fig. s6a, s6b of the revised manuscript**). We have also referenced these experiments in the text:

Lines 206-208: “*validation of the overexpression system at protein level by immunofluorescence in primary cells, and by immunoblot in cell lines can be found in Fig. S3c-d*”.

Lines 325-326: “*Validation of the knockdown system at the protein level was demonstrated by immunofluorescence in primary MDS cells, and by immunoblot in cell lines (Fig. S6a-b)*.”

(2) The Authors present additional evidence for an erythroid differentiation delay upon *DDIT3* overexpression. In the revised manuscript, this is particularly supported by the differentiation velocity of CD34+ cells with *DDIT3* analysed by scRNA-seq. Although the data are in agreement with flow cytometry, it is not necessarily straightforward that RNA velocity-measured splicing variants cannot be influenced by, for example, cell divisional status. I acknowledge that the Authors have included data comparing cell cycle and apoptosis of control and *DDIT3* cells, but it would have been more conclusive to perform the analyses within phenotypically-defined sub-populations to avoid averaging, as previously discussed. I am confused by the new panels G and H in Figure 4, as the values plotted do not seem to match the colour code. Can the Authors please clarify in the text and/or Figure Legend? Thank you. I would also recommend including the data on morphology and size of *DDIT3*-overexpressing BFU-E, as these data link with and strengthen the analysis of MDS samples and the conclusions drawn.

We thank the reviewer for the thoughtful comment. In order to interrogate the putative influence of cell divisional status on RNA-velocity results, we analyzed the cell cycle status of control and *DDIT3*-overexpressing cells for every state of erythroid differentiation detected by scRNAseq. As shown in figure 1, *DDIT3* overexpression did not promote differences in the cell cycle status at any state (differences are only detected for BFU state, but the number of cells is extremely low: 6 cells for control and 4 cells for *DDIT3*-overexpressing cells), further supporting our previous results (**Fig. S4d of the revised manuscript**) indicating that the dyserythropoiesis promoted by its overexpression is not associated with an alteration in the cell divisional status but by a delay in differentiation.

Figure 1. Proportion of cells in each phase of the cell cycle (G1, S, G2/M) for each erythroid differentiation state detected by scRNA-seq in control and *DDIT3*-overexpressing cells. The number of cells for each condition and differentiation state is indicated on the right.

We apologize if the color code of panels G and H of figure 4 was confusing. These panels represent UMAPs in which the velocity length was quantified and plotted as a heatmap. In order to avoid confusion of the readers, we have changed the color code in the revised version (**Fig. 4g and 4h of the revised manuscript**).

Regarding the data on morphology and size of *DDIT3*-overexpressing BFU-E, and following the suggestion from the reviewer, we have included the data in **figure S4a of the revised manuscript**, and in the text:

Lines 228-230: “*Erythroid colonies did not only present a lower total number upon overexpression of the factor, but also showed a modified morphology, being smaller and less compact than those formed by cells transduced with the control plasmid (Fig. S4a).*”

(3) The inclusion of additional MDS samples treated with *DDIT3*shRNA, as well as the quantification of knockdown strengthen the manuscript. This is particularly important since the presentation is descriptive for individual samples and there is variation between samples on the extent of knockdown effect in stage IV cells, and stage III abundances are contradictory between samples. To this end, I would also recommend including in the manuscript the overexpression of *DDIT3* in the low-expressing MDS16 and MDS17, as they give support to the model of *DDIT3*-driven dyserythropoiesis in MDS.

We thank the reviewer for the recommendation, and we completely agree with the reviewer that showing the effect of *DDIT3* inhibition and the overexpression in the same low-expressing MDS16 and MDS17 patients strengthens the conclusion of *DDIT3*-driven dyserythropoiesis. We have included the overexpression experiments in **figure S7c of the revised manuscript** and in the text:

Line 341-347: *“Interestingly, overexpression of DDIT3 in CD34+ cells from patients 16 and 17, which showed low basal levels of this factor, promoted a slight delay in erythroid differentiation, with a decrease of cells in stages III and IV, and a concomitant increase in stage I for MDS17, and stages I and II for specimen MDS16 (Fig. S7c). The opposite effects observed on erythroid differentiation upon knockdown and overexpression of DDIT3 in the same specimens further supported the model of DDIT3-driven dyserythropoiesis in MDS.”*

Reviewer #2 (Remarks to the Author):

The authors have significantly improved the manuscript by providing additional mechanistic studies on the role of DDIT3 in dyserythropoiesis. Even if some of the raised questions remained unanswered (is there DNA replication stress upon DDIT3 manipulation in HSCs?; is DDIT3 a chromatin bound regulator of erythroid differentiation?) this Reviewer acknowledges the difficulties of performing such experiments with limited amount of HSC-enriched subsets and remains in favor of publication given the potential translational relevance of the main findings. For statistical analyses: Line 743 and 753 (main text - merged file) - Missing name of the statistical test performed. Figure 7 A-B (rebuttal file) - Missing name of the statistical test performed. As a general comment, prior to publication I recommend to remove any statistical analysis from experiments for which n= biological replicates is equal or less than 3.

We thank the reviewer for the comment, and following his/her suggestion and the indications from the editor, we have eliminated all the statistics performed for experiments with less than 3 biological replicates. Furthermore, we have indicated the name of the statistical test used for every experiment.

Reviewer #3 (Remarks to the Author):

I continue to have some reservations about how this data is presented. On the one hand time series is set up as young > elderly > MDS, and on the other the authors argue that to look for MDS-related rather than age-related gene expression changes that it is better to use young vs MDS, because they use cells from “young” donors. Their explanation is somewhat convoluted. And the lack of clarity in the writing makes their rationale difficult to follow. I would suggest that a reworking of key sections of the paper comparing young, old and MDS sample is necessary to be absolutely clear about rationale of their studies.

We thank the reviewer for the comment and apologize if the explanations provided were not easy to follow. As the reviewer indicated, the time series we studied was set up as young healthy adults-older healthy adults-MDS. As explained in the manuscript, this analysis led to the identification of different transcriptional dynamisms in HSCs across aging and MDS, identifying, among others, gene expression changes that were characteristic of aging, and others that exclusively characterized MDS cells. Thus, to identify MDS-specific related alterations, we used both types of healthy samples.

Regarding the use of cells isolated from young instead of older healthy adults, we believe that reviewer refers to Figures 3e-f. In those figures, we generated the signatures Young_vs_MDS_HSCs_Up and Young_vs_MDS_HSCs_Down from the genes

differentially expressed between those two groups (young and MDS HSCs). The selection of healthy young HSCs instead of healthy elderly samples was due to the fact that in the experiment performed in figure 3, *DDIT3* was overexpressed in healthy young HSCs due to limiting cell numbers of healthy elderly HSCs. Thus, as overexpression was being induced in young HSCs, we considered that generating the gene signatures for GSEA analyses using young vs MDS HSCs was more appropriate. Nevertheless, as suggested by the reviewer, we have also included GSEAs generated from the gene signatures Older_healthy_adult_vs_MDS_HSCs_Up and Older_healthy_adult_vs_MDS_HSCs_Down, which were produced from the genes differentially expressed between those two groups (Older_healthy_adults and MDS HSCs) (**Fig. 3sh of the revised manuscript**).

Fig 19 in the Rebuttal should be shown in the manuscript. The statement “the enrichment in genes upregulated in such cells was positive but did not reach statistical significance” is incorrect given the Padj-value of 0.94. The authors should limit their interpretation to what the data show, which is that there is no difference in the upregulated genes.

We thank the reviewer for the comment and apologize if we mistakenly made an overstatement. We meant to indicate that although the NES of upregulated genes was positive (NES=0.79), it did not reach statistical significance. Fig 19 of the rebuttal has now been included as **fig. s3h of the revised manuscript**.

Figs 2C and F should have p-values provided for the comparisons.

Following this question, we would like to clarify that in the genes shown in figures 2C and 2F we did not aim to compare gene expression between two conditions, but rather represent examples of genes following specific and statistically significant trends of expression across aging and MDS, identified using MaSigPro (Fig. 1F of the revised manuscript). As suggested by the reviewer, we have now indicated the exact p-value obtained for the trend of expression of each gene for **figures 2c, 2f and supplementary figures s2b-d, s2f-h, s2j-l of the revised manuscript**.